# Supervised Graph Contrastive Learning for Gene Regulatory Networks

**Sho Oshima** [* 1]  **Yuji Okamoto** [* 1 2]  **Taisei Tosaki** [1 2]  **Ryosuke Kojima** [1 2]

## Abstract

Graph Contrastive Learning (GCL) is a powerful self-supervised learning framework that performs data augmentation through graph perturbations, with growing applications in the analysis of biological networks such as Gene Regulatory Networks (GRNs). The artificial perturbations commonly used in GCL, such as node dropping, induce structural changes that can diverge from biological reality. This concern has contributed to a broader trend in graph representation learning toward augmentation-free methods, which view such structural changes as problematic and should be avoided. However, this trend overlooks the fundamental insight that structural changes from biologically meaningful perturbations are not a problem to be avoided, but rather a rich source of information, thereby ignoring the valuable opportunity to leverage data from real biological experiments. Motivated by this insight, we propose SupGCL (Supervised Graph Contrastive Learning), a new GCL method for GRNs that directly incorporates biological perturbations from gene knockdown experiments as supervision. SupGCL is a probabilistic formulation that continuously generalizes conventional GCL, linking artificial augmentations with real perturbations measured in knockdown experiments, and using the latter as explicit supervision. On patient-derived GRNs from three cancer types, we train GRN representations with SupGCL and evaluate it in two regimes: (i) embedding space analysis, where it yields clearer disease-subtype structure and improves clustering, and (ii) task-specific fine-tuning, where it consistently outperforms strong graph representation learning baselines on 13 downstream tasks spanning gene-level functional annotation and patient-level prediction.

## 1. Introduction

Graph representation learning has recently attracted attention in various fields to learn a meaningful latent space to represent the connectivity and attributes in given graphs (Ju et al., 2024). Applications of graph representation learning are advancing in numerous areas where network data exists, such as analysis in social networks, knowledge graphs (Hu et al., 2023; Shen & Zhang, 2023), and biological network analysis in bioinformatics (Liu et al., 2023; Wu et al., 2021).

The application of graph representation learning to Gene Regulatory Networks (GRNs), which contain information about intracellular functions and processes, is particularly important in the fields of biology and drug discovery. It is expected to contribute to the identification of therapeutic targets and the elucidation of disease mechanisms. Representation learning for GRNs has been applied to tasks such as transcription factor inference (Yu et al., 2025) and predicting drug responses in cancer cell lines (Liu et al., 2022). Advances in gene expression measurement and analysis technologies have enabled the construction of patient-specific GRNs, highlighting gene regulation patterns that differ from the population as a whole (Nakazawa et al., 2021). Hereafter, this paper will refer to such individualized networks simply as GRNs.

Among the various graph representation learning methods, Graph Contrastive Learning (GCL) has emerged as a powerful self-supervised learning framework (You et al., 2020). GCL learns by maximizing the similarity between node representations across different views of the same graph generated via data augmentation, a process that assumes the preservation of essential features like graph topology (Zhu et al., 2021; Wang et al., 2024). While augmentation methods have been refined in the application of GCL to GRNs, there have been concerns that conventional artificial perturbations like node dropping can cause structural changes so significant—disrupting the function of networks including critical nodes like master regulators—that they hinder learning (Paull et al., 2021). In response to these challenges, Augmentation-Free approaches have been developed, which perturb model parameters instead of the graph structure itself, and have shown high performance (Thakoor et al., 2021; He et al., 2024).

However, this trend overlooks a fundamental insight: that

*Equal contribution [1]Department of Graduate School of Medicine, Kyoto University, Kyoto, Japan [2]RIKEN, Kobe, Japan. Correspondence to: Yuji Okamoto <yuji.0001@gmail.com>.

*Proceedings of the 43rd International Conference on Machine Learning*, Seoul, South Korea. PMLR 306, 2026. Copyright 2026 by the author(s).

the representational shifts caused by graph augmentation are not an obstacle to overcome but rather a rich source of information to be exploited. Consequently, it ignores the valuable opportunity to leverage data obtained from actual biological experiments, such as knockdown experiments, even though they have become readily accessible enabled by advances in high-throughput profiling technologies (Subramanian et al., 2017a).

To address these issues, we propose a novel supervised GCL method (SupGCL) that leverages gene knockdown perturbations within GRNs. Our method uses experimental data from actual gene knockdowns as supervision, enabling biologically faithful representation learning. In gene knockdown experiments, the suppression of specific genes causes biological perturbations, leading to the observation of a new altered GRN. By using these perturbations as supervision signals for GCL, we can perform data augmentation that retains biological characteristics. Moreover, since our method naturally extends traditional GCL models in the direction of supervised augmentation within a probabilistic framework, node-level GCL approaches emerge as special cases of our proposed model.

To evaluate the effectiveness of the proposed method, we benchmark SupGCL on patient-derived GRNs from three cancer types in both embedding space analysis and fine-tuning. Without any task-specific training, SupGCL produces markedly better disease-subtype structure in the embedding space, improving clustering quality. With fine-tuning, it consistently surpasses strong graph representation learning baselines—including previous GCL models—across 13 downstream tasks spanning gene-level functional annotation (Gene Ontology categories and cancer-gene classes) and patient-level tasks (survival risk and subtype classification). Our main contributions are as follows:

- **Redefining Graph Augmentation by Introducing Biological Perturbations:** We develop a new GCL method for GRNs that incorporates gene knockdown data as the supervision of graph perturbations to enhance biological plausibility.

- **Theoretical Extension of Node-level GCL:** We formulate supervised GCL, incorporating augmentation selection into a unified probabilistic modeling framework, and theoretically demonstrate that existing node-level GCL methods correspond to singular solutions under this formulation.

- **Empirical Validation:** We quantitatively demonstrate that SupGCL achieves higher clustering metrics (NMI/ARI) for disease subtypes without task-specific training. Furthermore, it consistently outperforms state-of-the-art baselines across 13 downstream tasks for three cancer types through fine-tuning.

Our implementation and all experimental codes and all GRNs datasets are available on `https://github.com/shobioinfo/SupGCL` and `https://zenodo.org/records/15496012`, respectively.

## 2. Related Works

**Augmentation-Free GCLs :** GCL learns representations by maximizing agreement between multiple augmented views of a graph. While early node-level methods such as GRACE achieved strong performance (Zhu et al., 2020), graph-level methods like GraphCL often suffered from losing key node features under augmentation (You et al., 2021; Sun et al., 2025). These methods remained a common limitation that is the heavy dependence on augmentation choices. This limitation motivated augmentation-free GCL, initiated by BGRL (Thakoor et al., 2021), which replaces heuristic (and potentially structure-damaging) augmentations with a bootstrapping mechanism, inspiring follow-up work such as simGRACE (Xia et al., 2022a) and AFGRL (Lee et al., 2022). SGRL (He et al., 2024) further incorporates principles such as feature uniformity to mitigate representation collapse observed in methods like BGRL, yielding stable and high-performing self-supervised learning. However, these approaches treat graph perturbations primarily as nuisances to avoid, and thus miss opportunities to exploit real-world supervisory data by structural changes.

**Augmentation-Adjustment GCLs :** To reduce reliance on manually designed augmentations, another line of work learns augmentations by directly optimizing graph perturbations. In this setting, AD-GCL (Suresh et al., 2021) and ArieL (Feng et al., 2024) generate robust views via adversarial training, while AutoGCL (Yin et al., 2022) learns diverse augmentation policies that preserve the labels of contrastive pairs. By endogenizing augmentation rather than treating it as a fixed noise source, these frameworks support robust learning across diverse graph structures. However, since their perturbations are driven solely by optimization objectives, they do not effectively expand data with variations that reflect real-world phenomena.

**Statistical Formulations of Contrastive Learning :** The theoretical formulation of contrastive learning has been actively evolving. The I-Con framework (Alshammari et al., 2025) presents a unified statistical formulation of representation learning by adopting a variational-inference perspective to organize contrastive learning methods as the minimization of KL divergences between an ideal reference distribution and a model-induced one. Within this framework, SupCon (Khosla et al., 2020), proposed as a supervised contrastive learning model, constructs the reference distribution using label data to concentrate features of the same class. However, such methods fundamentally rely on one-to-one

supervised categorical labels for each sample and remain restricted to probabilistic modeling at the sample level.

**Representation Learning for GRNs :** Applying Graph Neural Networks to known GRNs is a growing area of research for downstream tasks such as gene function classification and patient survival prediction(Zohari & Chehreghani, 2025). The prevailing approach in this field has been supervised learning without pre-training (Liu et al., 2022; Yu et al., 2025). While link prediction addresses the important task of inferring the unknown structure of GRNs (Huynh-Thu et al., 2010; Yu et al., 2025), our work focuses on learning representations from known GRN structures for downstream applications.

Recently, self-supervised methods have been introduced to this domain. Initial efforts focused on applying general augmentation-free methods, such as the Graph Autoencoder, which learns representations by reconstructing the graph's structure (Jung et al., 2024). More recently, the focus has begun to shift towards specialized GCL frameworks tailored for GRNs, although such approaches are still uncommon. A notable example is MuSe-GNN (Liu et al., 2023), which leverages different data modalities (e.g., scRNA-seq and scATAC-seq) as natural "views," avoiding the need for artificial augmentations by using biologically diverse information. In contrast to these methods, our work, SupGCL, uniquely incorporates real-world perturbations from within a single modality as supervisory signals for GCL.

## 3. Preliminaries

### 3.1. Background of Graph Contrastive Learning

Although there are various definitions of contrastive learning, it can be expressed using a probabilistic model based on KL divergence over pairs of augmentations or node instances (Alshammari et al., 2025). Let $\mathcal{X}$ denote a set of entities and let $(x, y) \in \mathcal{X} \times \mathcal{X}$ be a pair from that set. The contrastive loss is formulated as follows:

$$\text{Loss}_{\text{I-Con}} \triangleq \frac{1}{|\mathcal{X}|} \sum_{x \in \mathcal{X}} D_{\text{KL}}(p_\theta(y|x)|q_\phi(y|x)). \quad (1)$$

Here, $q_\phi(y|x)$ is the probability distribution of the learned model with parameter $\phi$, and $p_\theta(y|x)$ is a reference distribution. To avoid trivial solutions when training both $p_\theta$ and $q_\phi$ simultaneously, the reference distribution $p_\theta$ is almost fixed. The reference model $p_\theta(y|x)$ is often designed as a probability that assigns a non-zero constant to positive pairs $(x, y)$ and zero to negative pairs $(x, y)$.

Graph Contrastive Learning (GCL) handles the learned model $q_\phi(j|i)$ corresponding to a pair of nodes $(i, j)$. Consider graph operations for augmentation, order them, and represent the index of these operations by $a$. Let $z_i^a \in \mathbb{R}^d$

be the graph embedding of the $i$-th node obtained from the Graph Neural Network under the $a$-th augmentation operation. For two augmentation operations $(a, b)$, the pair of probability models $(p, q_\phi^{a,b})$ used in GCL is defined by

$$p(j|i) \triangleq \delta_{ij}, \quad q_\phi^{a,b}(j|i) \triangleq \frac{\exp(\text{sim}(z_i^a, z_j^b)/\tau_{\text{n}})}{\sum_{k \in \mathcal{V}} \exp(\text{sim}(z_i^a, z_k^b)/\tau_{\text{n}})}.$$

Here, $\mathcal{V}$ is the set of nodes in the given graph, $\delta_{ij}$ is the Kronecker delta, $\tau_{\text{n}} > 0$ is a temperature parameter and $\text{sim}(\cdot, \cdot)$ denotes cosine similarity. This setting is often extended so that the definitions of $(p, q_\phi^{a,b})$ vary according to how positive and negative pairs are sampled. Note that the target model $q_\phi$ depends on the sampling method of augmentation operators, so the probability model also depends on $(a, b)$.

GCL trains the model using the following loss function on the pair of probability models $(p, q_\phi^{a,b})$ induced by augmentation operations $(a, b)$, according to the formulation of contrastive learning loss (1).

$$\text{Loss}_{\text{Node}}^{a,b} \triangleq \frac{1}{|\mathcal{V}|} \sum_{i \in \mathcal{V}} D_{\text{KL}}(p(j|i)|q_\phi^{a,b}(j|i)). \quad (2)$$

This encourages the embeddings at the node level $z_i^a$ and $z_i^b$ of the same node under different augmentation operations to be close to each other. Typically, augmentation operations $a, b$ are chosen by uniform sampling from a set of candidates $\mathcal{A}$. Hence, in practice, the expected value is minimized under the uniform distribution $\text{U}_\mathcal{A}$ over $\mathcal{A}$:

$$\text{Loss}_{\text{Node}} \triangleq \mathbb{E}_{a,b \sim \text{U}_\mathcal{A}}[\text{Loss}_{\text{Node}}^{a,b}]. \quad (3)$$

While GCL achieves node-level representation learning via the procedure described above, in many cases the augmentation operations themselves rely on artificial perturbations such as randomly adding and/or deleting nodes and/or edges. In this study, we introduce gene knockdown – a biological perturbation – as a supervision for these augmentation operations.

### 3.2. Notation and Problem Definition

In this study, we describe a GRN as a directed graph $\mathcal{G} \triangleq (\mathcal{V}, \mathcal{E}, X^\mathcal{V}, X^\mathcal{E})$ that contains information on nodes and edges. Here, $\mathcal{V}$, and $\mathcal{E}$ are the sets of nodes and edges, respectively, and each node represents a gene. $X_{:,i}^\mathcal{V}$ is the feature of the $i$-th gene, and $X_{:,i}^\mathcal{E}$ is the feature of the $i$-th edge in the network. The augmentation operation corresponding to the knockdown of the $i$-th gene is modeled by setting the feature of the $i$-th gene to zero and also setting the features of all edges connected to the $i$-th gene to zero.

We associate the $a$-th augmentation operation with the knockdown of the $a$-th gene. In what follows, we denote by

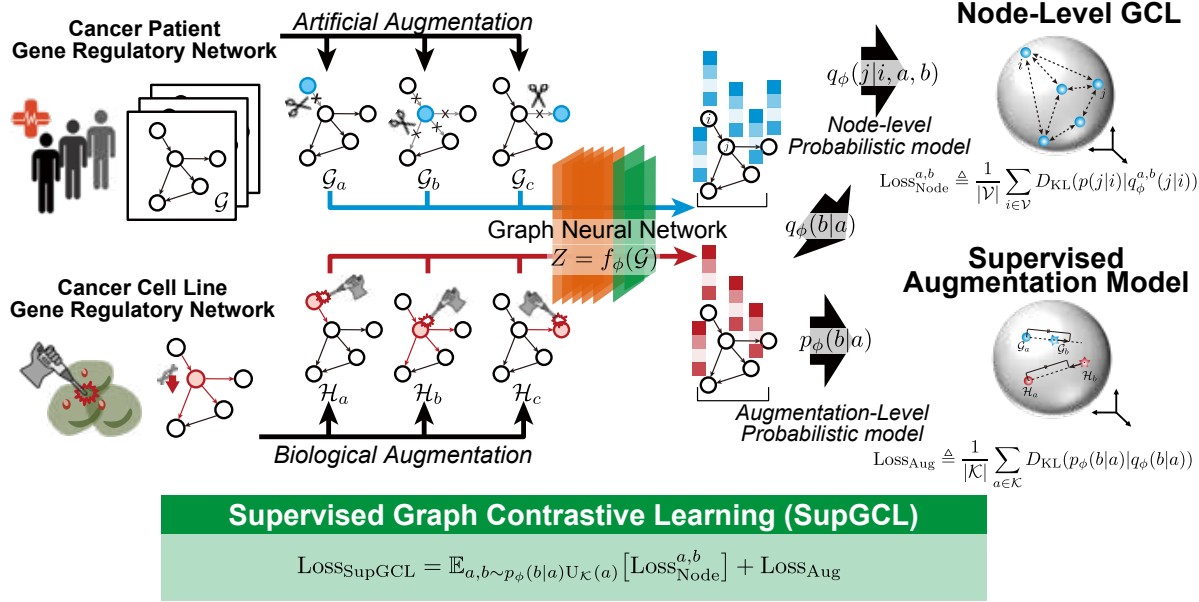

*Figure 1.* **Schematic overview of SupGCL.** Artificial augmentations are generated by simulating gene knockdowns in a patient GRN, while the teacher GRNs for supervision are derived from real-world knockdown experiments. Embeddings are extracted using a shared GNN, and both node-level and augmentation-level contrastive losses are computed via KL divergence.

$\mathcal{G}_a$ the graph obtained by applying the $a$-th augmentation operation to $\mathcal{G}$. Moreover, in this study, let $\mathcal{H}_a$ be the teacher GRN for the knockdown of the $a$-th gene , and let $\mathcal{K}$ be the set of all augmentation operations for which such teacher GRNs exist. In other words, $\mathcal{H}_a$ is a GRN that serves as a teacher for artificial augmentation for the $a$-th gene.

Our goal is to use the original GRN $\mathcal{G}$ and its teacher GRNs $\{\mathcal{H}_a\}_{a\in\mathcal{K}}$ to train a Graph Neural Network (GNN) $f_\phi$. Defining embedded representations through the GNN $f_\phi$ as

$$\boldsymbol{Z}^a \triangleq f_\phi(\mathcal{G}_a) \in \mathbb{R}^{|\mathcal{V}|\times d}, \quad \boldsymbol{Y}^a \triangleq f_\phi(\mathcal{H}_a) \in \mathbb{R}^{|\mathcal{V}|\times d},$$

where $\boldsymbol{z}_i^a$ and $\boldsymbol{y}_i^a$ denote the embedding vectors of the $i$-th node in $\boldsymbol{Z}^a$ and $\boldsymbol{Y}^a$, respectively, and $d$ is the embedding dimension. Note that the same GNN $f_\phi$ is used to produce both $\boldsymbol{Z}^a$ and $\boldsymbol{Y}^a$.

In this work, we train the neural network $f_\phi$ using the set of pairs $\{(\boldsymbol{Z}^a, \boldsymbol{Y}^a)\}_{a\in\mathcal{K}}$, where $(\boldsymbol{Z}^a, \boldsymbol{Y}^a)$ corresponds to the graph embedding obtained by the GRN augmentation operation and the embedding of the teacher GRN for the corresponding gene knockdown.

## 4. Method

For the set of embedded representations $\{(\boldsymbol{Z}^a, \boldsymbol{Y}^a)\}_{a\in\mathcal{K}}$, we consider the pair of augmentation operations $(a, b)$ and

the pair of nodes $(i, j)$ according to the contrastive learning scheme. First, for the pair of augmentation operations $(a, b)$, we clarify the supervised learning problem for augmentation operations using KL divergence and then propose SupGCL using a distribution over pairs of combinations of nodes and extension operations. A sketch of the proposed method is shown in Figure 1.

The probability distribution of augmentation operations is naturally introduced by using similarities in the entire graph embedding space $\mathbb{R}^{|\mathcal{V}|\times d}$ (rather than per node). By introducing the Frobenius inner product as the similarity in the matrix space, we define the probability models for the augmentation operations as:

$$p_\phi(b|a) \triangleq \frac{\exp\left(\mathrm{sim}_F(\boldsymbol{Y}^a, \boldsymbol{Y}^b)/\tau_a\right)}{\sum_{c\in\mathcal{K}} \exp\left(\mathrm{sim}_F(\boldsymbol{Y}^a, \boldsymbol{Y}^c)/\tau_a\right)},$$

$$q_\phi(b|a) \triangleq \frac{\exp\left(\mathrm{sim}_F(\boldsymbol{Z}^a, \boldsymbol{Z}^b)/\tau_a\right)}{\sum_{c\in\mathcal{K}} \exp\left(\mathrm{sim}_F(\boldsymbol{Z}^a, \boldsymbol{Z}^c)/\tau_a\right)},$$

where $\mathrm{sim}_F(\cdot, \cdot)$ denotes the cosine similarity via the Frobenius inner product, and $\tau_a > 0$ is a temperature parameter. Unlike node-level learning, $p_\phi(b|a)$ is not a fixed constant but rather a reference distribution based on the supervised embeddings $\{\boldsymbol{Y}^a\}_{a\in\mathcal{K}}$. Both probability models $p_\phi$ and $q_\phi$ are parameterized by the same GNN $f_\phi$.

Using these probability distributions, substituting the ref-

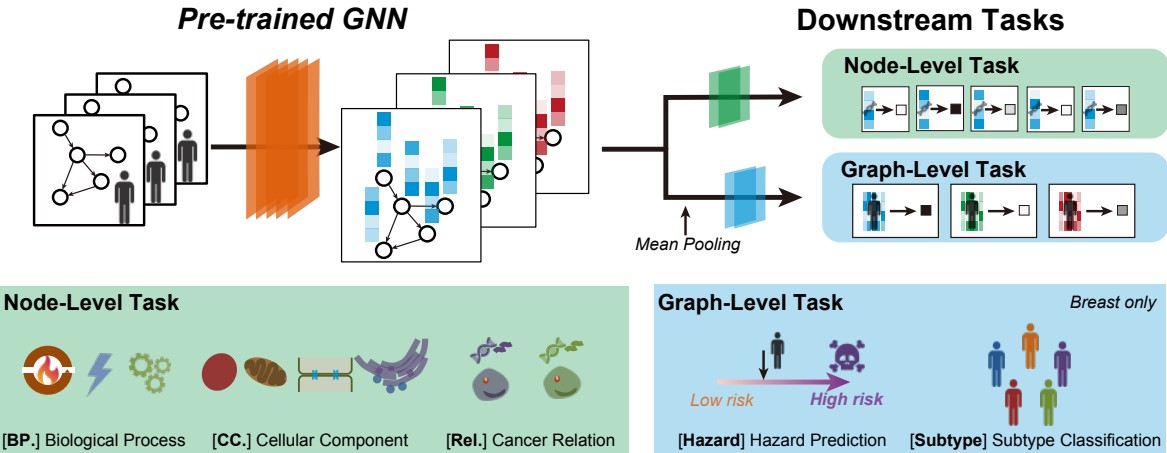

*Figure 2.* **Overview of downstream tasks.** Node-level tasks involve gene classification into Biological Process [**BP.**], Cellular Component [**CC.**], and cancer relevance [**Rel.**]. Graph-level tasks include patient survival prediction [**Hazard**] and breast cancer subtyping [**Subtype**]. Mean pooling provides graph-level representations.

erence model $p_\phi(b|a)$ and the learned model $q_\phi(b|a)$ into the formulation of contrastive learning in (1) yields the loss function for augmentation operations:

$$\text{Loss}_{\text{Aug}} \triangleq \frac{1}{|\mathcal{K}|} \sum_{a \in \mathcal{K}} D_{\text{KL}}(p_\phi(b|a)|q_\phi(b|a)). \quad (4)$$

Minimizing this loss reduces the discrepancy in embedding distributions between artificially augmented graphs and biologically grounded knockdown graphs. However, if both $p_\phi$ and $q_\phi$ are optimized simultaneously, the model may converge to a trivial solution. For example, if the GNN outputs constant embeddings, both distributions become uniform, and $\text{Loss}_{\text{Aug}} = 0$. Thus, minimizing $\text{Loss}_{\text{Aug}}$ alone is insufficient for learning meaningful graph representations.

To address this issue, here we first introduce a reference model $p_\phi(j, b|i, a)$ and a learned model $q_\phi(j, b|i, a)$ that use conditional probabilities for each pair of nodes and augmentation $(i, a), (j, b) \in \mathcal{V} \times \mathcal{K}$. By substituting these into the contrastive learning formulation in (1), we derive the loss function of Supervised Graph Contrastive Learning:

$$\text{Loss}_{\text{SupGCL}}$$
$$\triangleq \frac{1}{|\mathcal{V}||\mathcal{K}|} \sum_{i \in \mathcal{V}, a \in \mathcal{K}} D_{\text{KL}}(p_\phi(j, b|i, a)|q_\phi(j, b|i, a)). \quad (5)$$

Since data augmentation affects the entire graph, we assume that the graph-level teacher distribution $p_\phi(b|a)$, which evaluates the similarity between augmentation operations, and the node-level teacher distribution $p(j|i)$, which evaluates

node identity, are independent. This leads to the following theorem.

**Theorem 4.1.** *Assuming* $p_\phi(i, j, a, b) = p(i, j)p_\phi(a, b)$, *then*

$$\text{Loss}_{\text{SupGCL}}$$
$$= \mathbb{E}_{a,b \sim p_\phi(b|a)U_\mathcal{K}(a)}\left[\text{Loss}_{\text{Node}}^{a,b}\right] + \text{Loss}_{\text{Aug}}. \quad (6)$$

*Proof.* This follows directly from the standard decomposition of KL divergence: $D_{\text{KL}}(p(x, y)|q(x, y)) = \mathbb{E}_{x \sim p(x)}[D_{\text{KL}}(p(y|x)|q(y|x))] + D_{\text{KL}}(p(x)|q(x))$. See Appendix A for more details. □

The first term in Theorem 4.1 corresponds to the expectation of the node-level GCL loss $\text{Loss}_{\text{Node}}^{a,b}$ (as defined in (2)) with respect to the supervised augmentation distribution $p_\phi(b|a)$. Importantly, since the theorem is independent of the specific choice of the node-level model $(p, q_\phi^{a,b})$, any contrastive loss described by KL divergence can be used in practice. Meanwhile, the second term reduces the distributional difference between the artificially generated augmentation-based GRN and the teacher GRN. Minimizing these two terms enable robust graph-controlled learning that naturally avoids selecting genes that could dramatically alter the structure of gene regulatory networks (GRNs).

Moreover, the performance of the node-level representation learning and the biological validity following teacher data for the augmentation operations can be controlled by the temperature parameters $\tau_\text{n}$ and $\tau_\text{a}$ of each probability model. In particular, when the temperature parameter $\tau_\text{a}$ involved in

the augmentation operation is sufficiently large, the augmentation operation becomes independent of the teacher GRNs $\{\boldsymbol{Y}_a\}_{a\in\mathcal{K}}$, and coincides with the conventional node-level GCL loss function.

**Corollary 4.2.** $\lim_{\tau_{\mathrm{a}}\to\infty} \mathrm{Loss}_{\mathrm{SupGCL}} = \mathrm{Loss}_{\mathrm{Node}}$.

*Proof.* As $\tau_{\mathrm{a}} \to \infty$, we have $p_\phi(b|a) \to \mathrm{U}_\mathcal{K}(b)$ and $q_\phi(b|a) \to \mathrm{U}_\mathcal{K}(b)$. Therefore, the expectation term becomes $\lim_{\tau_{\mathrm{a}}\to\infty} \mathbb{E}_{a,b\sim p_\phi(b|a)\mathrm{U}_\mathcal{K}(a)}\left[\mathrm{Loss}_{\mathrm{Node}}^{a,b}\right] = \mathbb{E}_{a,b\sim\mathrm{U}_\mathcal{K}}[\mathrm{Loss}_{\mathrm{Node}}^{a,b}], \lim_{\tau_{\mathrm{a}}\to\infty} D_{\mathrm{KL}}(p_\phi(b|a)|q_\phi(b|a)) = 0$ , thus proving the corollary. $\square$

In this study, we train the GNN using standard gradient-based optimization applied to the loss function defined in Theorem 4.1. The corresponding pseudocode is provided in Appendix B.

*Remark* 4.3. When supervision is available for graph augmentations, one may consider learning the augmentation function itself as a possible strategy. However, such an approach results in an ill-posed formulation within the contrastive learning framework admitting trivial solutions. An analysis of the validity of SupGCL in these augmentation settings, together with a quantitative comparison across different augmentation schemes, is provided in the Appendix L.

*Remark* 4.4. The independence assumption $p_\phi(j,i,b,a) = p_\phi(b,a)p(j,i)$ implies that the vertex set of the teacher graph does not need to match that of the target graph. In general, if the vertex set $V(H_*)$ of the teacher graph differs from the vertex set $V(G)$ of the target graph, the KL divergence between $p_\phi(j,b \mid i,a)$ and the target distribution $q_\phi(j,b \mid i,a)$ is not defined because their domains are different. In contrast, under this assumption, the proposed method defines the loss function even when the node-level domains of the teacher and target distributions are different. Therefore, the proposed formulation can be applied to GRNs with different gene sizes and to GRN datasets composed of diverse cancer types.

# 5. Experiments

## 5.1. Benchmark of Gene Regulatory Networks

**Evaluation Protocol:** We evaluated the proposed method through the following procedure. First, we constructed patient-specific GRNs from cancer patient gene expression data and teacher GRNs from gene knockdown experiment data. Second, we pre-trained the proposed model on patient-specific and teacher GRNs, and evaluated the embedding space using NMI and ARI with respect to disease subtype clustering. Finally, we evaluated 13 fine-tuning downstream task performance (e.g., classification accuracy and regression performance) using the pre-trained models. The compared five baselines listed below.

**w/o-pretrain:** Directly performs classification or regression for downstream tasks.

**GAE(Kipf & Welling, 2016):** Reconstruction based graph representation learning method.

**GraphCL(You et al., 2021):** Graph contrastive learning method using positive pairs between graphs.

**GRACE(Zhu et al., 2020):** Node-level graph contrastive learning method.

**SGRL (He et al., 2024):** State-of-the-art (SOTA) augmentation-free GCL method.

These five models consist of representative methods for four types of graph representation learning model, together with a SOTA GCL model. Other comparison results (augmentation-free GCLs, augmentation-adjustment GCLs, and biological foundation models) are presented in Appendix K.

**Datasets:** We benchmarked on real-world datasets, constructing patient-specific GRNs from cancer samples in The Cancer Genome Atlas (TCGA) (Weinstein et al., 2013) and teacher GRNs from gene knockdown experiments in cancer cell lines from the Library of Integrated Network-based Cellular Signatures (LINCS) (Subramanian et al., 2017a). The TCGA and the LINCS are both large-scale and widely-used public platforms providing gene expression dataset from cancer patients and cell lines, respectively.

We considered three cancer types available in both resources: breast, lung, and colorectal cancer. To ensure a consistent node set across TCGA- and LINCS-derived networks, we restricted all GRNs to the intersection of genes measured in TCGA and the LINCS L1000 landmark set, resulting in 975 genes. The same 975-gene set was used for SupGCL and all baseline methods to avoid confounding the comparison by differences in the available input genes. This restriction also keeps GRN estimation computationally tractable while retaining a LINCS-defined representative landmark-gene space. TCGA included N=1092 (breast), 1011 (lung), and 288 (colorectal) patient samples. For LINCS, the total numbers of knockdown experiments were 8793 (breast), 11843 (lung), and 15926 (colorectal); among the 975 common genes, the numbers of unique knockdown targets were 768, 948, and 948, respectively.

Beyond expression profiles, we used additional annotations for downstream evaluation. TCGA provides survival status and disease subtype labels for each patient sample. At the gene level, we assigned multi-label Gene Ontology (GO) annotations—Biological Process (BP; metabolism, signaling, cell organization; 3 classes) and Cellular Component (CC; nucleus, mitochondria, ER, membrane; 4 classes)—as well as binary cancer-relevance labels based on the OncoKB cancer-related gene list (Rel). These labels were used for downstream tasks. (see Appendices C and E in detail.)

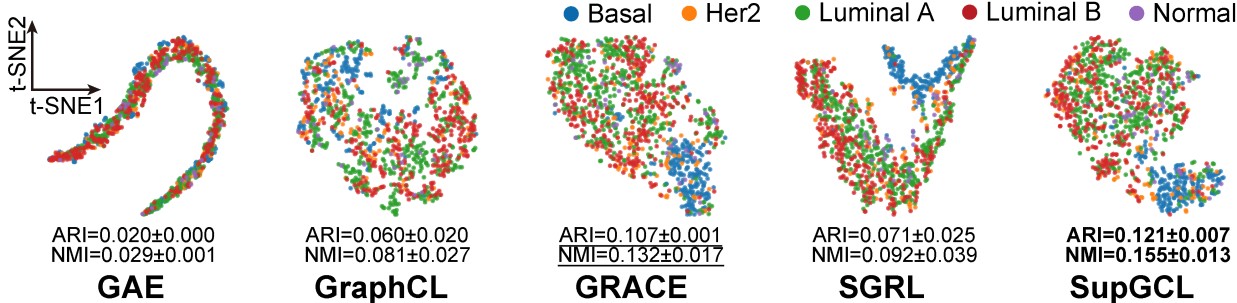

*Figure 3.* t-SNE visualization of pre-trained graph-level embeddings on breast cancer GRNs. Each point represents the readout feature of an individual patient's network. NMI and ARI scores indicate quantitative clustering metrics of the embeddings across 5 experimental runs.

*Table 1.* Fine-tuning results on downstream tasks. **BP**, **CC**, and **Rel** are node-level tasks, while **Hazard** and **Subtype** are graph-level tasks. **Breast**, **Lung**, and **Colorectal** denote cancer types.

| TASK | W/O-PRETRAIN | GAE | GRAPHCL | GRACE | SGRL | SUPGCL |
|---|---|---|---|---|---|---|
| **BP.** [SUBSET ACCURACY ↑] | | | | | | |
| BREAST | 0.232±0.031 | 0.230±0.029 | 0.167±0.042 | 0.230±0.051 | 0.220±0.052 | **0.243±0.052** |
| LUNG | 0.259±0.056 | 0.247±0.038 | 0.115±0.024 | 0.259±0.063 | 0.233±0.027 | **0.282±0.037** |
| COLORECTAL | 0.231±0.062 | 0.245±0.023 | 0.207±0.058 | 0.249±0.050 | 0.146±0.029 | **0.262±0.030** |
| **CC.** [SUBSET ACCURACY ↑] | | | | | | |
| BREAST | 0.264±0.042 | 0.250±0.034 | 0.131±0.050 | 0.236±0.026 | 0.249±0.030 | **0.291±0.026** |
| LUNG | 0.267±0.041 | 0.245±0.033 | 0.069±0.041 | 0.255±0.043 | 0.248±0.037 | **0.274±0.044** |
| COLORECTAL | 0.278±0.098 | 0.256±0.042 | 0.190±0.062 | 0.265±0.030 | 0.133±0.081 | **0.279±0.052** |
| **REL.** [ACCURACY ↑] | | | | | | |
| BREAST | 0.573±0.033 | 0.561±0.059 | 0.553±0.051 | 0.575±0.035 | 0.580±0.055 | **0.600±0.057** |
| LUNG | 0.575±0.053 | 0.568±0.029 | 0.555±0.036 | 0.592±0.038 | 0.593±0.034 | **0.604±0.053** |
| COLORECTAL | 0.563±0.071 | 0.574±0.049 | 0.535±0.056 | 0.576±0.071 | 0.580±0.042 | **0.594±0.039** |
| **HAZARD** [C-INDEX ↑] | | | | | | |
| BREAST | 0.601±0.035 | 0.625±0.035 | 0.638±0.049 | 0.642±0.064 | 0.640±0.077 | **0.650±0.059** |
| LUNG | 0.611±0.052 | 0.619±0.062 | 0.616±0.049 | 0.609±0.055 | 0.611±0.060 | **0.627±0.051** |
| COLORECTAL | 0.621±0.070 | 0.631±0.091 | 0.657±0.071 | 0.647±0.059 | 0.616±0.123 | **0.698±0.085** |
| **SUBTYPE** [ACCURACY ↑] | | | | | | |
| BREAST | 0.804±0.031 | 0.834±0.028 | 0.719±0.077 | 0.841±0.026 | 0.829±0.030 | **0.847±0.036** |

**Pre-processing:** To estimate the network structure of each GRN from gene expression data, we used a Bayesian network structure learning algorithm based on B-spline regression (Imoto et al., 2002). For each experiment, gene expression values were used as node features, while edge features were defined as the linear sum of estimated regression coefficients and the parent node's gene expression (Tanaka et al., 2020). This structure estimation was performed per cancer type per dataset using the above algorithm. Further details are provided in Appendix D.

**Model Setting:** For all methods including the SupGCL and comparative models, we used the same 5-layer Graph Transformer architecture (Shi et al., 2021). Hyperparameters for pre-training were tuned with Optuna (Akiba et al., 2019), and the model was optimized using the AdamW optimizer (Loshchilov & Hutter, 2019). All training runs were

performed on a single NVIDIA H100 SXM5 GPU. Additional experimental details are provided in Appendix E, and empirical runtime and computational complexity analyses are reported in Appendix F.

### 5.2. Result 1: Embedding Space Analysis

We visualized the graph-level embedding spaces from the breast cancer type (Figure 3). The embeddings, colored by disease subtype, show that GAE and GraphCL fail to separate subtypes, while GRACE and SGRL exhibit moderate separation. SupGCL significantly outperforming other methods on both the Normalized Mutual Information (NMI) and the Adjusted Rand Index (ARI), affirming that its embeddings capture subtype-specific structure. This advantage is statistically supported by quantitative metrics across 5 pre-training runs.

Table 2. Ablation study of SupGCL on the augmentation temperature $\tau_a$ for breast cancer.

| $\tau_a$ | BP. ↑ | CC. ↑ | REL. ↑ | HAZARD ↑ | SUBTYPE ↑ |
|---|---|---|---|---|---|
| 0.10 | $\mathbf{0.262 \pm 0.035}$ | $0.289 \pm 0.049$ | $0.586 \pm 0.047$ | $\mathbf{0.670 \pm 0.078}$ | $0.837 \pm 0.029$ |
| 0.25 (DEFAULT) | $0.243 \pm 0.052$ | $\mathbf{0.291 \pm 0.026}$ | $0.600 \pm 0.057$ | $0.650 \pm 0.059$ | $\mathbf{0.847 \pm 0.036}$ |
| 0.50 | $0.261 \pm 0.034$ | $0.284 \pm 0.061$ | $\mathbf{0.606 \pm 0.039}$ | $0.648 \pm 0.053$ | $0.846 \pm 0.032$ |
| 1.00 | $0.244 \pm 0.042$ | $0.280 \pm 0.032$ | $0.596 \pm 0.048$ | $0.640 \pm 0.056$ | $0.835 \pm 0.031$ |
| 2.00 | $0.237 \pm 0.024$ | $0.277 \pm 0.058$ | $0.590 \pm 0.044$ | $0.656 \pm 0.060$ | $0.842 \pm 0.031$ |
| $\infty$ (W/O SUPERVISED GCL) | $0.233 \pm 0.055$ | $0.259 \pm 0.054$ | $0.573 \pm 0.070$ | $0.633 \pm 0.051$ | $0.832 \pm 0.034$ |
| REFERENCE: GRACE | $0.230 \pm 0.051$ | $0.236 \pm 0.026$ | $0.575 \pm 0.035$ | $0.642 \pm 0.064$ | $0.841 \pm 0.026$ |

Furthermore, we evaluated the biological validity of the node-level embeddings performing enrichment analysis (Appendix J). These analyses demonstrated that SupGCL captured the biological context more effectively than existing methods.

## 5.3. Result 2: Evaluation by Downstream Task

In this experiment, pre-training of the proposed and conventional methods was conducted using patient-specific GRNs from TCGA and teacher GRNs from LINCS. Subsequently, fine-tuning was performed on the pre-trained models using patient GRNs, and downstream task performance was evaluated (see Figure 2). During fine-tuning, two additional fully connected layers were appended to the node-level representations and graph-level representations (obtained via mean pooling), and downstream tasks were performed.

Graph-level tasks (hazard prediction, subtype classification) used survival and subtype labels from TCGA. Note that subtype classification was conducted only for breast cancer. Node-level tasks (BP., CC., and Rel.) used gene-level annotations from Gene Ontology and OncoKB. Each downstream task was evaluated using 10-fold cross-validation. For cancer gene classification, due to label imbalance, we performed undersampling over 10 random seeds. The results are reported as mean ± standard deviation.

Table 1 reports the results of node-level and graph-level downstream tasks, with the best and second-best performances indicated in bold and underlined, respectively. SupGCL achieves the highest performance across all datasets and tasks compared to other pre-training methods. This performance advantage extends to a broader range of methods, including augmentation-free and augmentation-adjustment GCL methods and biological foundation models, as detailed in Appendix K.

Note that for the node-level task, existing graph representation learning methods exhibit little difference compared to the results without pretraining. This indicates that graph representation learning methods without a teacher GRN fail to learn meaningful representations that capture gene-level characteristics relevant to this downstream task. In contrast,

our proposed method leverages a teacher GRN, enabling the learning of gene representations associated with biological perturbations, which leads to improved performance.

In addition, we evaluated the generalization ability and robustness of SupGCL (see Appendices H and I, respectively). SupGCL demonstrated that pre-training remained effective even when the cancer types of the teacher and patient GRNs differed. Furthermore, our model exhibited strong robustness under limited availability of patient or teacher GRNs, as well as in the presence of noise in the estimated GRNs.

## 5.4. Result 3: Effectiveness of Supervised Learning

In this section, to verify the effectiveness of the reference model $p_\phi$, we conducted additional experiments by varying the augmentation-level temperature parameter $\tau_a$, which controls its degree of influence. This experiment also serves to empirically validate the theoretical relationship established in Corollary 4.2, which states that, as $\tau_a$ increases, the SupGCL objective asymptotically approaches the objective of unsupervised node-level graph contrastive learning (GCL). The experimental results on the breast cancer dataset, summarized in Table 2, demonstrate the performance of SupGCL under different values of $\tau_a$. As $\tau_a$ becomes larger—corresponding to a weaker influence of the knockdown-derived supervised information—the performance gradually deteriorates and converges to that of the model without supervised information ($\tau_a = \infty$). This observation is consistent with the theoretical result in Corollary 4.2. Furthermore, when $\tau_a = \infty$, the performance converges to a level comparable to that of GRACE, an unsupervised node-level GCL choising different probablistic distribution $q_\phi(j \mid i)$. These results indicate that the knockdown-derived supervised information contributes substantially to improving contrastive learning performance, while its influence is effectively controlled by $\tau_a$. The same tendency is consistently observed for the Lung and Colorectal datasets as well (see Appendix G.6).

## 6. Conclusion

In this study, we proposed SupGCL, a supervised graph contrastive learning method for Gene Regulatory Networks (GRNs) that incorporates biological perturbations from real-world gene knockdown experiments as supervision for graph augmentation. This proposed method formulates supervised graph augmentation within a unified probabilistic framework and theoretically derived that existing node-level GCL methods can be derived as special cases of our formulation.

In our empirical evaluation, an analysis of the learned embedding space first confirmed that SupGCL significantly improves clustering metrics (NMI/ARI) for cancer subtypes, capturing latent structures with high biological plausibility. Furthermore, evaluating the pre-trained model on downstream tasks with fine-tuning, SupGCL consistently outperformed state-of-the-art baselines across 13 tasks, including gene function classification and patient survival analysis.

A limitation of this study is that the empirical validation of the proposed method does not cover larger-scale graphs, such as those with tens or hundreds of thousands of nodes. This is because the dimensionality of the target graphs is constrained by biological factors, such as the accuracy of gene regulatory network inference and the biological importance of the genes themselves. Consequently, we have not investigated how the performance of SupGCL is affected as the size of the target graph increases, particularly with respect to the extent to which supervisory information contributes to performance improvements in larger-scale graphs.

Another limitation of our study is the challenge of cross-domain generalization caused by distributional shifts across different cancer types. However, preliminary results presented in the Appendix H suggest that pan-cancer learning—integrating multiple cancer types—is effective, enriching representations without compromising performance. Future work will focus on expanding the range of cancer types and evaluating SupGCL on larger-scale GRNs, paving the way for the development of large-scale, general-purpose biological foundation models.

## Acknowledgments

This work was supported by JST Moonshot R&D (JPMJMS2021, JPMJMS2024), JST CREST(No. JP-MJCR22D3), JST Research and Development Program for Next-generation Edge AI Semiconductors (JPM-JES2511), JSPS KAKENHI (JP25K00148, JP25H02626, JP26K14994), a project (JPNP14004) commissioned by the New Energy and Industrial Technology Development Organization (NEDO), and RIKEN TRIP initiative (AGIS). This work used computational resources of the supercomputer Fugaku provided by RIKEN through the HPCI System Research Project (Project IDs: hp150272, ra000018). Taisei Tosaki received financial support from RIKEN Jr. Research-associated Programs.

## Impact Statement

This work proposes a pretraining framework based on Supervised Graph Contrastive Learning (SupGCL) that learns representations of patient-derived gene regulatory networks (GRNs) using experimentally measured perturbations from gene knockdown experiments as supervision. By aligning synthetic graph perturbations with biologically meaningful perturbations, the proposed approach can better elucidate disease subtype structure in patient GRNs for breast, lung, and colorectal cancers, and is expected to improve performance on multiple downstream tasks such as gene-level functional inference and patient-level prediction. This may facilitate hypothesis generation in cancer research, deepen mechanistic understanding, and improve prioritization of candidate genes and pathways, thereby supporting exploration from basic research to translational studies. All data used in this study are obtained from public resources and do not involve additional collection of personal information. However, the learned representations are statistical in nature and do not directly establish causal effects or guarantee clinical utility.

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

## Appendix Contents

## A. Detail Proof of Theorem 4.1

Since the loss function of contrastive learning (1) is formulated as the KL divergence between the joint distributions $p(x, y)$ and $q(x, y)$, under the assumption that $p(x)$ and $q(x)$ follow uniform distributions. Then, the following derived for $\text{Loss}_{\text{I}-\text{Con}}$:

$$\text{Loss}_{\text{SupGCL}} = \frac{1}{|V||\mathcal{K}|} \sum_{i \in V, a \in \mathcal{K}} D_{\text{KL}}(p_\phi(j, b|i, a)|q_\phi(j, b|i, a)).$$

$$= D_{\text{KL}}(p_\phi(i, j, a, b)|q_\phi(i, j, a, b))$$

$$= \mathbb{E}_{(a,b) \sim p_\phi(a,b)} \Big[ D_{\text{KL}}(p_\phi(i, j|a, b)|q_\phi(i, j|a, b)) \Big] + D_{\text{KL}}(p_\phi(a, b)|q_\phi(a, b))$$

$$= \mathbb{E}_{(a,b) \sim p_\phi(a,b)} \Big[ D_{\text{KL}}(p(i, j)|q_\phi(i, j|a, b)) \Big] + D_{\text{KL}}(p_\phi(a, b)|q_\phi(a, b))$$

$$= \mathbb{E}_{a,b \sim p_\phi(b|a) \mathrm{U}_\mathcal{K}(a)} \Big[ \text{Loss}_{\text{Node}}^{a,b} \Big] + \text{Loss}_{\text{Aug}}$$

The derivation from the second to the third line utilizes the basic decomposition of KL divergence: $D_{\text{KL}}(p(x, y)|q(x, y)) = \mathbb{E}_{x \sim p(x)}[D_{\text{KL}}(p(y|x)|q(y|x))] + D_{\text{KL}}(p(x)|q(x))$.

## B. Algorithm

The learning algorithm of this research is presented in Algorithm 1. We train the Graph Neural Network (GNN) $f_\phi$ using the target GRN dataset for all patients, $\mathcal{G}_{\text{all}} = \{\mathcal{G}^{(i)}\}_{i=1}^N$, and the teacher GRN dataset, $\mathcal{H}_{\text{all}} = \{\mathcal{H}_a^{(i)}\}_{i \in \mathcal{I}_a, a \in \mathcal{K}}$. Here, $\mathcal{I}_a$ is the set of indices for teacher GRNs, $\{\mathcal{H}_a^{(i)}\}_{i \in \mathcal{I}_a}$, which correspond to data augmentations for the $a$-th node.

Our algorithm follows a standard training loop, consisting of the calculation of $\text{Loss}_{\text{SupGCL}}$ and the optimization of $f_\phi$ using AdamW. Furthermore, to reduce computational costs, we employ sampling-based estimation of the normalization constant for the calculation of softmax functions $p_\phi(b|a)$ and $q_\phi(b|a)$, and use importance sampling for the calculation of $\text{Loss}_{\text{node}}^{a,b}$ and $\text{Loss}_{\text{Aug}}$. For computational performance reasons, we adopted an Augmentation sampling size of 8.

We discuss the computational order and sampling of SupGCL in Appendix F.3.2 and Appendix F.4.2.

---

**Algorithm 1** Training loop of SupGCL:

---

**Input:** Graph Neural Network $f_\phi$; patient GRNs $\mathcal{G}_{\text{all}} = \{\mathcal{G}^{(i)}\}_{i=1}^N$; teacher GRNs $\mathcal{H}_{\text{all}} = \{\mathcal{H}_a^{(i)}\}_{i \in \mathcal{I}_a, a \in \mathcal{K}}$
**for** each $\mathcal{G} \subset \mathcal{G}_{\text{all}}$ **do**
    Sample $a, b \sim \mathrm{U}_\mathcal{K}$
    Obtain augmented graphs $\mathcal{G}_a, \mathcal{G}_b$ from $\mathcal{G}$
    Select teacher graphs $\mathcal{H}_a, \mathcal{H}_b$ from $\mathcal{H}_{\text{all}}$
    $\boldsymbol{Z}^a, \boldsymbol{Z}^b \leftarrow f_\phi(\mathcal{G}_a), f_\phi(\mathcal{G}_b)$ // *Embed target GRNs*
    $\boldsymbol{Y}^a, \boldsymbol{Y}^b \leftarrow f_\phi(\mathcal{H}_a), f_\phi(\mathcal{H}_b)$ // *Embed teacher GRNs*
    $q_\phi(b|a) \leftarrow \text{softmax}\left(\left[\frac{\langle \boldsymbol{Z}^a, \boldsymbol{Z}^* \rangle_F}{\tau_a}\right]\right)[b]$ // *Target augmentation distribution*
    $p_\phi(b|a) \leftarrow \text{softmax}\left(\left[\frac{\langle \boldsymbol{Y}^a, \boldsymbol{Y}^* \rangle_F}{\tau_a}\right]\right)[b]$ // *Teacher augmentation distribution*
    $\text{Loss}_{\text{Aug}} \leftarrow |\mathcal{K}| \, p_\phi(b|a) \big( \log p_\phi(b|a) - \log q_\phi(b|a) \big)$ // *Importance-sampled augmentation loss*
    $q_\phi(j|i, a, b) \leftarrow \text{softmax}\left(\left[\frac{\langle \boldsymbol{z}_i^a, \boldsymbol{z}_*^b \rangle}{\tau_n}\right]\right)[j]$ // *Node-level target distribution*
    $\text{Loss}_{\text{Node}}^{a,b} \leftarrow \frac{1}{|\mathcal{V}|} \sum_{i \in \mathcal{V}} \log q_\phi(i|i, a, b)$
    $\text{Loss}_{\text{SupGCL}} \leftarrow |\mathcal{K}| \, p_\phi(b|a) \, \text{Loss}_{\text{Node}}^{a,b} + \text{Loss}_{\text{Aug}}$ // *Importance-sampled SupGCL loss*
    Update $f_\phi$ using AdamW optimizer
**end for**
**Output:** Trained Graph Neural Network $f_\phi$

---

## C. Details of Experimental Datasets

This section provides details on data acquisition and preprocessing procedures.

## C.1. Details on the TCGA Dataset

In this study, for gene expression data, we used normalized count data from the TCGA TARGET GTEx study provided by UCSC Xena (UCSC Xena, 2016) for the TCGA dataset. For the LINCS dataset, we used normalized gene expression data from the LINCS L1000 GEO dataset (GSE92742) (Subramanian, 2017).

The TCGA platform contains four datasets: the gene expression dataset "dataset: gene expression RNAseq – RSEM norm_count," the dataset for cancer type attributes of individual patients "dataset: phenotype – TCGA TARGET GTEx selected phenotypes," the patient prognosis dataset "dataset: phenotype – TCGA survival data," and the patient phenotype data "dataset: phenotype - Phenotypes."

In this study, based on the dataset of patient cancer type attributes, we extracted patient IDs corresponding to the TCGA cohorts listed in Table 3, and subsequently obtained the associated gene expression data. Notably, the study population was limited to patients whose target cancer was a primary tumor. Additionally, overall survival time (OS.time) and survival status (OS: alive = 0 / deceased = 1) were retrieved from the patient prognosis dataset. For breast cancer specifically, disease subtype labels based on the PAM50 classification were acquired from the patient phenotype dataset. PAM50 classification using RNA expression data was feasible for all samples, assigning all 1,092 breast cancer patients to one of five subtypes: Luminal A (N = 438), Luminal B (N = 311), Basal (N = 196), Her2 (N = 111), and Normal (N = 36). Patient samples with missing values were excluded during the data extraction process. The final sample sizes and mortality rates for each cancer type after processing are summarized in Table 3.

*Table 3.* Sample extraction conditions and survival data statistics for each cancer type

| Cancer Type | TCGA Cohort Name | Number of samples with valid survival time data | Number of deaths | Mortality rate (%) |
| --- | --- | --- | --- | --- |
| Breast cancer | Breast Invasive Carcinoma | 1,090 | 151 | 13.9 |
| Lung cancer | Lung Adenocarcinoma + Lung Squamous Cell Carcinoma | 996 | 394 | 39.6 |
| Colon cancer | Colon Adenocarcinoma | 286 | 69 | 24.1 |

## C.2. Details on the LINCS Dataset

### C.2.1. PREPARING LINCS DATASET

In this study, we used the Level 3 normalized gene expression data (filename on LINCS datasets: "GSE92742_Broad_LINCS_Level3_INF_mlr12k_n1319138x12328.gctx.gz") provided from the GEO dataset (GSE92742) of the LINCS L1000 project. Additionally, by referring to the concurrently provided experimental metadata ("GSE92742_Broad_LINCS_inst_info.txt.gz" indicating cell lines and treatment conditions, and "GSE92742_Broad_LINCS_pert_info.txt.gz" indicating drug and gene knockdown information), we extracted only the shRNA-mediated knockdown experiment groups. The cell lines and treatment durations were limited to samples from: MCF7 breast cancer cell line treated for 96 hours, A549 lung cancer cell line treated for 96 hours, and HT29 colon cancer cell line treated for 96 hours. The expression data was limited to 978 landmark genes.

### C.2.2. DISCUSSION FOR THE QUALITY OF LINCS DATASET

It is well-documented that LINCS datasets contain significant noise, particularly off-target effects (Subramanian et al., 2017b). While aggregation methods such as Consensus Gene Signatures (CGS) have been proposed to isolate consistent on-target effects (Subramanian et al., 2017b), they inherently reduce the effective sample size—a trade-off we sought to avoid in this study. Despite the use of instance-level data without such aggregation, SupGCL demonstrated superior predictive performance on downstream tasks, especially graph-level classification, compared to existing methods. Furthermore, our enrichment analysis(Appendix J) reveals that SupGCL successfully acquired latent representations with high biological interpretability. These findings suggest that SupGCL is capable of robust pretraining even when learning from noisy teacher data.

## C.3. Extraction of Gene Label Data for BP/CC Tasks using Gene Ontology

In this study, for Biological Process (BP) classification and Cellular Component (CC) classification in downstream tasks, we performed multi-label annotation for each of the 975 genes constituting the GRN, based on terms obtained from the GO database. For BP labels, three categories whose importance is known were used: 'Metabolism', 'Signal Transduction', and 'Cellular Organization' (Paolacci et al., 2009). Similarly, for CC labels, four categories were used: 'Nucleus', 'Mitochondrion', 'Endoplasmic Reticulum', and 'Plasma Membrane' (Costa et al., 2010). These categories were selected to cover the major functions of the GRN.

To extract these multi-labels, the following steps were performed:

1. Batch retrieve terms associated with each gene using the MyGene.info API (Xin et al., 2015) and the OBO file (available at https://geneontology.org/docs/download-ontology/).

2. Aggregate multi-labels for the target genes using the GOATOOLS library (Ashburner et al., 2000).

Genes that could not be labeled into any of the BP or CC categories were excluded from the downstream tasks in this study. The number and percentage of genes included in each category are shown in Table 4.

*Table 4.* Gene label distribution for high-level BP/CC categories (n = 975)

| Category | Number of genes | Percentage of category (%) |
|---|---|---|
| BP: Metabolism | 533 | 54.67 |
| BP: Signal Transduction | 261 | 26.77 |
| BP: Cellular Organization | 369 | 37.85 |
| BP: Not Applicable | 204 | 20.92 |
| CC: Nucleus | 388 | 39.80 |
| CC: Mitochondrion | 151 | 12.49 |
| CC: Endoplasmic Reticulum | 148 | 15.18 |
| CC: Plasma Membrane | 231 | 23.69 |
| CC: Not Applicable | 257 | 26.36 |

## C.4. Annotation by OncoKB

For cancer-related gene classification in downstream tasks, cancer-related genes were obtained from the OncoKB™ Cancer Gene List provided by OncoKB (Oncology Knowledge Base) (Chakravarty et al., 2017). Using the list of 1188 cancer-related genes provided as of April 30, 2025, positive labels were assigned to the 975 genes used in this study. As a result, 106 genes were labeled as cancer-related genes.

# D. Estimating Gene Regulatory Networks

## D.1. Taxonomy of Gene Regulatory Network Estimation

Estimating accurate gene regulatory networks (GRNs) is crucial for elucidating cellular processes and disease mechanisms. Methods for computationally estimating GRNs from gene expression data can be categorized into correlation-based (Langfelder & Horvath, 2008), mutual information-based (Margolin et al., 2006), probabilistic graphical models-based such as Bayesian networks (Friedman et al., 2000), and deep learning-based approaches (Shu et al., 2021). The number of experimentally validated GRNs is limited. Therefore, GRN estimation methods need the ability to account for measurement errors and appropriately capture non-linear and multimodal interactions between genes while mitigating overfitting. Thus, we adopted a method combining Bayesian network estimation using multiple sampling and non-parametric regression (Imoto et al., 2002) to identify patient- or sample-specific GRNs (Nakazawa et al., 2021).

## D.2. Details of Estimating GRNs

To construct patient-specific GRNs, we estimate Bayesian Networks using B-spline regression. First, we estimate the conditional probability density functions between genes using the entire gene expression dataset. Then, using these learned

parameters, we construct patient- or sample-specific GRNs.

Let $x_1, x_2, ..., x_n$ be random variables for $n$ nodes, and let $\mathrm{pa}(i)$ be the set of parent nodes of the $i$-th node. In this case, a Bayesian network using B-spline curves (Imoto et al., 2002) is defined as a probabilistic model decomposed into conditional distributions with parent nodes:

$$p(x_1, \ldots, x_n) = \prod_{i=1}^{n} p(x_i | x_{\mathrm{pa}(i)})$$
$$= \prod_{i=1}^{n} \mathcal{N}\left(x_i \,\middle|\, \sum_{j \in \mathrm{pa}(i)} m_{ij}(x_j), \sigma^2\right).$$

Here, $\mathcal{N}$ is a Gaussian distribution, and $m_{ij}$ is a B-spline curve defined by B-spline basis functions $b_s : \mathbb{R} \to \mathbb{R}$ as

$$m_{ij}(x_j) = \sum_{s=1}^{M} w_{i,j}^s b_s(x_j) \tag{7}$$

The Bayesian network is estimated by learning the relationships with parent nodes based on the model described above and the parameters of the conditional distributions. In this study, the score function used for searching the structure of the Bayesian Network can be analytically derived using Laplace approximation (Imoto et al., 2002). Since the problem of finding a Directed Acyclic Graph (DAG) that maximizes this score is NP-hard, we performed structure search using a heuristic structure estimation algorithm, the greedy hill-climbing (HC) algorithm (Imoto et al., 2003). Furthermore, to ensure the reliability of the estimation results by the HC algorithm, we performed multiple sampling runs.

We extracted edges that appeared more frequently than a predefined threshold relative to the number of sampling runs in the estimated networks. Finally, for each edge in the obtained network structure, the conditional probabilities were relearned using all input data.

Using the network and conditional probabilities learned here, we derive patient-specific GRNs (Tanaka et al., 2020). The node set and edge set of the graph for a patient-specific GRN are defined by the network learned with gene expression levels as random variables $x_1, \ldots, x_n$. Furthermore, the feature of the $i$-th node $X_i^{\mathcal{V}}$ is the gene expression level of each sample, and the feature of an edge from $i$ to $j$ (designated as the $k$-th edge) is the realization of the learned B-spline curve $X_k^{\mathcal{E}} = m_{ij}(x_j)$. Patient-specific networks using such edge features have led to the discovery of subtypes that correlate more strongly with prognosis than existing subtypes (Nakazawa et al., 2021). Additionally, using differences in edge features between patients to extract patient-specific networks has been reported to contribute to the identification of novel diagnostic and therapeutic marker candidates in diseases such as idiopathic pulmonary fibrosis (Tomoto et al., 2024) and chronic nonbacterial osteomyelitis (Yahara et al., 2022)

For GRN estimation, we used INGOR (version 0.19.0), a software that estimates Bayesian Networks based on B-spline regression, and executed it on the supercomputer Fugaku. INGOR is based on SiGN-BN (Tamada et al., 2011), a software that similarly estimates Bayesian Networks using B-spline regression, and achieves faster estimation by optimizing parallel computation on Fugaku. In all GRN estimations, the number of sampling runs was set to 1000, and the threshold for adopting edges was set to 0.05. Since network estimation with a large amount of sample data can lead to Out of Memory errors, the upper limit of gene expression data used for network estimation was set to 3000 (especially for LINCS data). All other hyperparameters related to network estimation used the default settings of SiGN-BN. The number of Fugaku nodes and the required execution times are shown in Table 5. In addition, the number of parallel threads was set to 4 for estimations using TCGA data and 2 for LINCS data.

*Table 5.* Estimated network times in supercomputer Fugaku

| | Breast cancer | | Lung cancer | | Colorectal cancer | |
|---|---|---|---|---|---|---|
| | TCGA | LINCS | TCGA | LINCS | TCGA | LINCS |
| **Number of Fugaku Nodes** | 288 | 288 | 288 | 528 | 288 | 528 |
| **Estimated Time** [hh:mm:ss] | 00:54:49 | 08:35:12 | 00:38:24 | 13:28:07 | 00:05:40 | 21:01:38 |

### D.3. Results of Estimating GRNs

The statistics of the estimated networks are shown in Table 6. It should be noted that network estimation methods using sampling can extract highly reliable edges, but they may occasionally extract structures containing cyclic edges. However, all networks estimated in this study maintained a DAG structure.

*Table 6.* Estimated network statistics

| | Breast cancer | | Lung cancer | | Colorectal cancer | |
|---|---|---|---|---|---|---|
| | TCGA | LINCS | TCGA | LINCS | TCGA | LINCS |
| **Number of Nodes** | 975 | 975 | 975 | 975 | 975 | 975 |
| **Number of Edges** | 13170 | 10498 | 13322 | 13968 | 13686 | 12541 |
| **Average Degree** | 13.5077 | 10.7671 | 13.6636 | 14.3262 | 14.0369 | 12.8626 |

## E. Experimental Setting

### E.1. Validity of GNN Encoder

In this study, we consistently employed the Graph Transformer as the GNN encoder architecture. The choice of encoder is a critical factor that directly affects model performance, and its validity must be carefully assessed for our method. Our decision to adopt the Graph Transformer was based on prior work on GRNs (Liu et al., 2023) (Appendix E.1), where it achieved the best performance among various architectures. That study evaluated multiple GNN encoders—including GCN, GAT, TransformConv, SURGL, GPS, and GRACE—in the context of learning gene representations by integrating heterogeneous modalities such as scRNA-seq, scATAC-seq, and spatial transcriptomics. Their results showed that the Graph Transformer consistently outperformed other models, whereas GCN- and GAT-based models failed to sufficiently capture gene functional similarity across datasets. It should be noted, however, that this prior work focused on heterogeneous modalities, and is therefore not directly comparable to our setting, which relies exclusively on GRNs.

Motivated by these findings, we chose the Graph Transformer over GCN or GAT in our framework. Both GAT and Graph Transformer share the ability to incorporate real-valued edge features into node updates. However, in our experiments (using GRACE as an example), replacing the Graph Transformer with GAT sometimes led to degraded performance. The results are reported in Table 7.

*Table 7.* Comparison of downstream task performance with different encoders in GRACE (Breast Cancer GRN).

| Model Setting | Hazard (c-index) | BP (Subset Accuracy) |
|---|---|---|
| GRACE (Graph Transformer, in paper) | $0.642 \pm 0.064$ | $0.230 \pm 0.051$ |
| GRACE (GAT) | $0.632 \pm 0.038$ | $0.241 \pm 0.040$ |

### E.2. Hyperparameters Settings

In this study, we used Optuna (Akiba et al., 2019), a Bayesian optimization tool, for hyperparameter search in pre-training. The search space for all models included the AdamW learning rate $\mathrm{lr} \in [10^{-5}, 10^{-3}]$, batch size $\mathrm{batch\_size} \in \{4, 8\}$, and model-specific hyperparameters. Model-specific hyperparameters were the temperature parameter $\tau \in \{0.25, 0.5, 0.75, 1.0\}$ for GraphCL, GRACE, and SupGCL, and the global-hop parameter $k \in \{1, 2, 3\}$ for SGRL. The graph embedding dimension was unified to 64 for all models. See Appendix G for a discussion on embedding dimension. For training the pre-trained models, the data was split into training and validation sets at an $8 : 2$ ratio, and the validation loss was used as the metric for various decisions. The optimal hyperparameters determined by actual hyperparameter tuning are shown in Table 8.

### E.3. Fine-tuning Settings

This section described the fine-tuning setting of downstream-task. The overview is in Table 9.

*Table 8.* Hyperparameter settings for each cancer type

| Cancer Type | Model | learning rate | batch size | temperature | global hop |
|---|---|---|---|---|---|
| Breast cancer | GAE | $5.74 \times 10^{-4}$ | 4 | — | — |
| | GRACE | $9.71 \times 10^{-4}$ | 4 | 0.25 | — |
| | GraphCL | $1.23 \times 10^{-4}$ | 4 | 0.25 | — |
| | SGRL | $2.39 \times 10^{-4}$ | 4 | — | 3 |
| | SupGCL | $2.37 \times 10^{-4}$ | 4 | 0.25 | — |
| Lung cancer | GAE | $2.16 \times 10^{-4}$ | 4 | — | — |
| | GRACE | $4.44 \times 10^{-5}$ | 8 | 0.25 | — |
| | GraphCL | $6.24 \times 10^{-5}$ | 4 | 0.25 | — |
| | SGRL | $8.18 \times 10^{-5}$ | 8 | — | 2 |
| | SupGCL | $1.89 \times 10^{-4}$ | 4 | 0.25 | — |
| Colorectal cancer | GAE | $2.26 \times 10^{-4}$ | 4 | — | — |
| | GRACE | $4.03 \times 10^{-5}$ | 8 | 0.25 | — |
| | GraphCL | $8.83 \times 10^{-5}$ | 4 | 0.25 | — |
| | SGRL | $7.10 \times 10^{-4}$ | 4 | — | 3 |
| | SupGCL | $3.32 \times 10^{-4}$ | 4 | 0.25 | — |

*Table 9.* Description of the downstream tasks

| Task | Task Type | Metrics |
|---|---|---|
| **Node-Level Task** | | |
| [**BP.**]: Biological process classification | Multi-label binary classification (with 3 labels) | Subset accuracy |
| [**CC.**]: Cellular component classification | Multi-label binary classification (with 4 labels) | Subset accuracy |
| [**Rel.**]: Cancer relation | Classification (binary) | Accuracy |
| **Graph-Level Task** | | |
| [**Hazard**]: Hazard prediction | Survival analysis (1-dim risk score) | C-index |
| [**Subtype**]: Disease subtype prediction | Classification (5 groups) | Accuracy |

### E.3.1. GRAPH-LEVEL TASK

For fine-tuning graph-level tasks (Hazard Prediction, Subtype Classification), training was performed on a per-patient basis. The latent states embedded by the Graph Neural Network were transformed into graph-level embeddings using mean-pooling, and then fed through a 2-layer MLP to train task-specific models.

We employed 10-fold cross-validation across patients to generate training/test datasets. Fine-tuning was performed using AdamW with a learning rate of $1 \times 10^{-3}$. For evaluation, we reported the mean and standard deviation of the scores across all folds.

**Hazard Prediction :**    For the hazard prediction task, we adopted the classic Cox proportional hazards model. In the Cox model, the hazard function for a patient at time $t$ is defined as

$$h(t \mid \boldsymbol{x}) = h_0(t) \exp\!\big(\boldsymbol{\beta}^\top \boldsymbol{x}\big)$$

Here, $h_0(t)$ is the baseline hazard, $\boldsymbol{x}$ is the input variable, and $\boldsymbol{\beta}$ is the regression coefficient to be learned. The prognosis estimation is performed by connecting this input variable $\boldsymbol{x}$ to the graph NN and its head.

This study performed training using partial likelihood maximization based on patient prognosis information and evaluated performance using the C-index.

**Subtype Classification:**    For the subtype classification task, we created a classification model using a 5-class softmax function and trained it using multi-class cross-entropy. Furthermore, performance was evaluated using Accuracy and Macro F1-score. The F1-score results are shown in Appendix G.

### E.3.2. NODE-LEVEL TASK

For fine-tuning node-level tasks (BP/CC Classification, Cancer Rel. Classification), tasks were solved on a per-gene basis. For the latent state of each node embedded by the Graph Neural Network, task-specific models were learned through a 2-layer MLP.

In BP/CC Classification, performance was evaluated using gene-wise 10-fold cross-validation. For Cancer Rel. Classification, it is necessary to mitigate class imbalance in positive and negative label data. To achieve this, we prepared a dataset by undersampling the negative label data, split it into training and test data at an 8:2 ratio, and performed fine-tuning and accuracy evaluation. This undersampling and data splitting process was repeated 10 times with different seeds to evaluate the performance on this task.

For optimization, AdamW was used with a batch size of 8 and a learning rate of $1 \times 10^{-3}$. For evaluation, we reported the mean and standard deviation of the scores for each fold.

**BP./CC. Classification :**   In Biological Process (BP) classification, three categories for each gene—"metabolism," "signal transduction," and "cellular organization"—are predicted as a multi-hot vector. In Cellular Component (CC) classification, four categories—"nucleus," "mitochondria," "endoplasmic reticulum," and "plasma membrane"—are predicted as a multi-hot vector. The model was structured using a sigmoid function for each category, and training was performed using binary cross-entropy for each respective category. Performance was evaluated using Subset Accuracy, Macro F1-score, and Jaccard Index as evaluation metrics. The Macro F1-score and Jaccard Index results are shown in Appendix G.

**Cancer Rel. Classification**   In cancer-related gene classification, 106 genes defined as positive by OncoKB were labeled as "positive," and all other genes were labeled as "negative" for binary classification. A model was created to estimate negative and positive cases using a sigmoid function, and training was performed using binary cross-entropy. Performance was evaluated using Accuracy and F1-score as evaluation metrics. The F1-score results are shown in Appendix G.

## F. Empirical Runtime and Computational Complexity of SupGCL

In this section, we first report the empirical runtime of the proposed method and the comparative methods, and then discuss the computational complexity of the proposed and conventional GCL methods.

### F.1. Computational Environment and Computation Time

All experiments were conducted on an NVIDIA H100 SXM5 (95.83 GiB), and the computational run time for each model is shown in Table 10. Please note that although the number of training steps is based on 3000 epochs, the actual training time varies due to early stopping using the validation data.

### F.2. Computational Complexity of SupGCL and Reduction Methods

The computational complexity of contrastive learning depends on the complexity of calculating the representation matrix, the dot product calculations in the softmax function, and the frequency with which each occurs. In this chapter, we present the forward pass computational complexity of the SupGCL loss function and compare it with the complexity of node-level Graph Contrastive Learning (GCL), which serves as the baseline for this study. Next, we demonstrate methods to reduce SupGCL complexity via sampling and analyze the resulting complexity.

The model variables related to sampling used hereafter are shown in Table 11, and the order of complexity for each is shown in Table 12. In the following, please note that the mapping destination by the Graph Neural Network $f_\phi$ lies on the hypersphere feature space $\mathbb{S}_d$. Consequently, the cosine similarity of node features $\langle z_i | z_j \rangle$ and the graph-level cosine similarity $\mathrm{sim}_F(Z^a, Z^b)$ can be described as a simple dot product $\langle z_i | z_j \rangle$ and the Frobenius inner product $\mathrm{sim}_F(Z^a, Z^b)$, respectively, without loss of generality.

*Table 10.* Pre-training computational run time for each cancer type

| Cancer Type | Model | Computation Time | Epochs |
|---|---|---|---|
| | GAE | 3.354 hr | 3000 |
| | GRACE | 10.77 hr | 3000 |
| Breast cancer | GraphCL | 8.306 hr | 2000 |
| | SGRL | 5.859 hr | 2000 |
| | SupGCL | 21.40 hr | 1500 |
| | GAE | 1.875 hr | 1800 |
| | GRACE | 9.960 hr | 3000 |
| Lung cancer | GraphCL | 3.303 hr | 1300 |
| | SGRL | 7.485 hr | 3000 |
| | SupGCL | 19.67 hr | 1500 |
| | GAE | 45.23 min | 2500 |
| | GRACE | 2.839 hr | 3000 |
| Colorectal cancer | GraphCL | 1.476 hr | 2000 |
| | SGRL | 53.14 min | 1100 |
| | SupGCL | 9.551 hr | 2500 |

*Table 11.* Summary of Parameters and Notation

| Variable | Description |
|---|---|
| $d$ | Dimension of the feature vector for each node |
| $\mathcal{V}$ | Set of nodes |
| $\mathcal{K}$ | Set of augmentation methods. Basically, $\mathcal{K} \approx \mathcal{V}$. |
| $\mathbb{S}_d$ | $d$-dimensional spherical set, i.e., $\mathbb{S}_d \triangleq \{x \in \mathbb{R}^d \mid \|x\|^2 = 1\}$ |
| $z_i^a \in \mathbb{S}_d$ | Feature vector of node $i$ under graph augmentation $a$ |
| $Z^a, Y^a \in \mathbb{S}_d^{|\mathcal{V}|}$ | Feature matrix of the graph under graph augmentation $a$ |
| $\mathcal{K}_\mathcal{B} \subset \mathcal{K}$ | Batch set of augmentation methods |
| $\mathcal{V}_\mathcal{B} \subset \mathcal{V}$ | Batch set of vertices (nodes) |

## F.3. Computational Complexity of Graph Contrastive Learning

### F.3.1. NODE-LEVEL GCL

In node-level contrastive learning, the loss function is calculated using combinations of inner products of node features. Since the computational complexity of the node feature inner product $\langle z_i^a | z_j^b \rangle$ is $O(d)$, the loss function for each graph augmentation pair $a, b \in \mathcal{K}$ is described as follows:

$$
\begin{aligned}
\text{Loss}_{\text{Node}}^{a,b} &\triangleq \frac{1}{|\mathcal{V}|} \sum_{i \in \mathcal{V}} D_{\text{KL}}(\delta_{*i} | q_\phi(*|i, b, a)) = -\frac{1}{|\mathcal{V}|} \sum_{i \in \mathcal{V}} \log q_\phi(i|i, b, a) \\
&= \frac{1}{|\mathcal{V}|} \sum_{i \in \mathcal{V}} \left( \log \sum_{j \in \mathcal{V}} e^{\langle z_i^a | z_k^b \rangle / \tau_{\text{n}}} + \square \right)
\end{aligned}
\tag{8}
$$

Here, overlapping parts of the inner product are denoted by "$\square$" and are omitted. The loss function $\text{Loss}_{\text{Node}}^{a,b}$ involves two GNN computations and $|\mathcal{V}|^2$ inner product operations. Therefore, assuming the complexity of calculating the representation matrix $Z^* = f_\phi(\mathcal{G}_*)$ using a GNN is $O(N_{\text{GNN}})$, the complexity of $\text{Loss}_{\text{Node}}^{a,b}$ is $O(N_{\text{GNN}} + |\mathcal{V}|^2)$.

For the node-level loss function $\text{Loss}_{\text{Node}}^{a,b}$ with respect to graph augmentations $a, b \in \mathcal{K}$, the expected loss function is

*Table 12.* Summary of Forward Pass Complexity

| Target | Forward Complexity | Description |
|--------|--------------------|-------------|
| $Z^a = f_\phi(G^a)$ | $O(N_{\text{GNN}})$ | Forward pass of GNN |
| $\langle z_i^a \vert z_j^b \rangle$ | $O(N_{\text{GNN}} + d)$ | Node-level inner product |
| $\text{sim}_F(Z^a, Z^b)$ | $O(N_{\text{GNN}} + \vert\mathcal{V}\vert d)$ | Graph-level inner product |
| $\widehat{\text{sim}}_F(Z^a, Z^b)$ | $O(N_{\text{GNN}} + \vert\mathcal{V}_\mathcal{B}\vert d)$ | Batched graph-level inner product |
| $\text{Loss}_{\text{Node}}$ | $O(\vert\mathcal{K}\vert N_{\text{GNN}} + \vert\mathcal{K}\vert^2\vert\mathcal{V}\vert^2 d)$ | Node-level loss function |
| $\text{Loss}_{\text{SupGCL}}$ | $O(\vert\mathcal{K}\vert N_{\text{GNN}} + \vert\mathcal{K}\vert^2\vert\mathcal{V}\vert^2 d)$ | SupGCL loss function |
| $\widehat{\text{Loss}}_{\text{Node}}$ | $O(\vert\mathcal{K}_\mathcal{B}\vert N_{\text{GNN}} + \vert\mathcal{K}_\mathcal{B}\vert^2\vert\mathcal{V}\vert^2 d)$ | Node-level loss function with sampled augmentation methods (used for standard node-level GCL training like GRACE) |
| $\widehat{\text{Loss}}_{\text{SupGCL}}$ | $O(\vert\mathcal{K}_\mathcal{B}\vert N_{\text{GNN}} + \vert\mathcal{K}_\mathcal{B}\vert^2\vert\mathcal{V}\vert^2 d)$ | SupGCL loss function with sampled augmentation methods (used for result training) |
| $\widehat{\widehat{\text{Loss}}}_{\text{Node}}$ | $O(\vert\mathcal{K}_\mathcal{B}\vert N_{\text{GNN}} + \vert\mathcal{K}_\mathcal{B}\vert^2\vert\mathcal{V}_\mathcal{B}\vert^2 d)$ | Node-level loss function with sampled augmentation methods and vertices (used for large-scale graph representation learning) |
| $\widehat{\widehat{\text{Loss}}}_{\text{SupGCL}}$ | $O(\vert\mathcal{K}_\mathcal{B}\vert N_{\text{GNN}} + \vert\mathcal{K}_\mathcal{B}\vert^2\vert\mathcal{V}_\mathcal{B}\vert^2 d)$ | SupGCL loss function with sampled augmentation methods and vertices |

defined as:

$$
\text{Loss}_{\text{Node}} \triangleq \mathbb{E}_{a,b \sim \text{U}_\mathcal{A}}\left[\text{Loss}_{\text{Node}}^{a,b}\right] = \frac{1}{\vert\mathcal{K}\vert^2} \sum_{a,b \in \mathcal{K}} \text{Loss}_{\text{Node}}^{a,b}
$$

$$
= \frac{1}{\vert\mathcal{V}\vert\vert\mathcal{K}\vert^2} \sum_{a,b \in \mathcal{K}} \sum_{i \in \mathcal{V}} \left( \log \sum_{j \in \mathcal{V}} e^{\langle z_i^a \vert z_k^b \rangle / \tau_{\text{n}}} + \square \right)
$$

(9)

Since $\text{Loss}_{\text{Node}}$ involves $\vert\mathcal{K}\vert$ GNN computations and $O(\vert\mathcal{K}\vert^2\vert\mathcal{V}\vert^2)$ inner product operations, the complexity is $O(\vert\mathcal{K}\vert N_{\text{GNN}} + \vert\mathcal{K}\vert^2\vert\mathcal{V}\vert^2 d)$. Note that due to the symmetry of the inner product $\langle z_i^a \vert z_j^b \rangle = \langle z_j^b \vert z_i^a \rangle$, the actual number of operations is less than $\vert\mathcal{K}\vert^2\vert\mathcal{V}\vert^2$.

### F.3.2. COMPUTATIONAL COMPLEXITY OF SUPGCL

The calculation of the SupGCL loss function uses the Frobenius inner product $\text{sim}_F(Z^a, Z^b)$, which is a graph-level inner product operation. While the complexity of this operation is $O(\vert\mathcal{V}\vert d)$, please note that due to its relationship with node-level inner product operations:

$$
\text{sim}_F(Z^a, Z^b) = \frac{1}{\vert\mathcal{V}\vert} \sum_{i \in \mathcal{V}} \langle z_i^a \vert z_i^b \rangle
$$

(10)

it is not necessary to calculate the Frobenius inner product separately if node-level inner products have already been computed.

Now, the SupGCL loss function is defined by a combination of augmentation-level loss and node-level loss. First, consider the complexity of the augmentation-level loss function $\text{Loss}_{\text{Aug}}$:

$$
\text{Loss}_{\text{Aug}} \triangleq \frac{1}{\vert\mathcal{K}\vert} \sum_{a \in \mathcal{K}} D_{\text{KL}}\big(p_\phi(*\vert a)\big\vert q_\phi(*\vert a)\big) = \frac{1}{\vert\mathcal{K}\vert} \sum_{a,b \in \mathcal{K}} p_\phi(b\vert a)\Big(\log p_\phi(b\vert a) - \log q_\phi(b\vert a)\Big)
$$

$$
= \frac{1}{\vert\mathcal{K}\vert} \sum_{a,b \in \mathcal{K}} \square\Big(\log \sum_{c \in \mathcal{K}} e^{\text{sim}_F(Z^a, Z^c)\tau_{\text{a}}} - \log \sum_{c \in \mathcal{K}} e^{\text{sim}_F(Y^a, Y^c)/\tau_{\text{a}}} + \square\Big)
$$

(11)

The calculation of the augmentation-level loss function $\text{Loss}_{\text{Aug}}$ involves $2|\mathcal{K}|$ GNN computations and $O(|\mathcal{K}|^2)$ Frobenius inner product $\langle * | * \rangle_F$ operations, resulting in a complexity of $O(|\mathcal{K}|N_{\text{GNN}} + |\mathcal{K}|^2|\mathcal{V}|d)$. The reason the complexity is reduced compared to the node-level loss function $\text{Loss}_{\text{Node}}$ is that it does not perform inner products between all features across all nodes; thus, the number of node-level inner product summations is reduced from $O(|\mathcal{V}|^2)$ to $O(|\mathcal{V}|)$.

Finally, we present the complexity of SupGCL. The SupGCL loss function is:

$$
\begin{aligned}
\text{Loss}_{\text{SupGCL}} &= \mathbb{E}_{a,b\sim p_\phi(b|a)\text{U}_{\mathcal{K}}(a)}\left[\text{Loss}_{\text{Node}}^{a,b}\right] + \text{Loss}_{\text{Aug}} \\
&= \mathbb{E}_{a,b\sim p_\phi(b|a)\mathcal{U}(a)}\left[-\frac{1}{|\mathcal{V}|}\sum_{i\in\mathcal{V}}\log q_\phi(i|i,b,a) + \log p_\phi(b|a) - \log q_\phi(b|a)\right] \\
&= \sum_{a,b\in\mathcal{K}}\square\left[\frac{1}{|\mathcal{V}|}\sum_{i\in\mathcal{V}}\left(\log\sum_{k\in\mathcal{V}}e^{\langle z_i^a|z_k^b\rangle/\tau_{\text{n}}} + \square\right) - \log\sum_{c\in\mathcal{K}}e^{\text{sim}_F(Y^a,Y^c)/\tau_{\text{a}}} + \square\right]
\end{aligned}
\tag{12}
$$

Consequently, $2|\mathcal{K}|$ GNN computations are performed. For $Z$, $O(|\mathcal{K}|^2|\mathcal{V}|^2)$ inner products are computed, and for $Y$, $O(|\mathcal{K}|^2)$ Frobenius inner products $\langle * | * \rangle_F$ are computed. Thus, the total complexity is $O(2|\mathcal{K}|N_{\text{GNN}} + |\mathcal{K}|^2|\mathcal{V}|^2d + |\mathcal{K}|^2|\mathcal{V}|d) = O(|\mathcal{K}|N_{\text{GNN}} + |\mathcal{K}|^2|\mathcal{V}|^2d)$. Therefore, it can be confirmed that the order of complexity is no different from that of node-level GCL.

### F.4. Reduction of Computational Complexity Using Sampling

Accurate calculation of the loss functions for the proposed method and conventional node-level GCL requires $O(|\mathcal{K}|N_{\text{GNN}} + |\mathcal{K}|^2|\mathcal{V}|^2d)$ complexity, which depends heavily on the number of graph augmentations $|\mathcal{K}|$ and the number of graph nodes $|\mathcal{V}|$. Therefore, it has been conventional to perform sampling on augmentation methods and vertices to approximate the loss function, thereby enabling representation learning on large-scale graphs. This study similarly introduces sampling methods and calculates their computational complexity.

#### F.4.1. SAMPLING IN NODE-LEVEL GCL

Generally, in contrastive learning, training is performed by randomly sampling augmentation methods and calculating the loss function for them. Let $\mathcal{K}_{\mathcal{B}} \subset \mathcal{K}$ be a batch randomly selected from the augmentation set. The sampled loss function used for gradient calculation is described as follows:

$$
\widehat{\text{Loss}}_{\text{Node}} \triangleq \frac{1}{|\mathcal{K}_{\mathcal{B}}|^2}\sum_{a,b\in\mathcal{K}_{\mathcal{B}}^2}\text{Loss}_{\text{Node}}^{a,b}
\tag{13}
$$

Since the complexity of the loss function $\text{Loss}_{\text{Node}}^{a,b}$ for each augmentation method is $O(N_{\text{GNN}} + |\mathcal{V}|^2d)$, the complexity of the sampled loss function $\widehat{\text{Loss}}_{\text{Node}}$ becomes $O(|\mathcal{K}_{\mathcal{B}}|N_{\text{GNN}} + |\mathcal{K}_{\mathcal{B}}|^2|\mathcal{V}|^2d)$. Although we consider only pairs within the batch $\mathcal{K}_{\mathcal{B}}$ to simplify the analysis, this is generally extended to sample each independently.

In large-scale graphs, sampling is performed not only in the batch direction but also in the node direction. While there are sampling methods that sample the graph itself and use subgraphs, below we consider a method that approximates the normalization term of the softmax function $q(j|i,b,a)$ using sampling.

$$
\hat{q}(j|i,b,a) \triangleq \frac{1}{\hat{D}_{\text{node}}^i}e^{\langle z_i^a|z_j^b\rangle/\tau_{\text{n}}}, \quad \hat{D}_{\text{node}}^i = \sum_{k\in\mathcal{V}_{\mathcal{B}}}e^{\langle z_i^a|z_k^b\rangle/\tau_{\text{n}}}
\tag{14}
$$

Here, $\mathcal{V}_{\mathcal{B}} \subset \mathcal{V}$ is a batch of vertices randomly selected from the vertex set, and its size is defined as $|\mathcal{V}_{\mathcal{B}}|$.

Using this, the loss function considering node-direction sampling is defined as follows:

$$
\widehat{\widehat{\text{Loss}}}_{\text{Node}} \triangleq \frac{1}{|\mathcal{K}_{\mathcal{B}}^2|}\sum_{a,b\in\mathcal{K}_{\mathcal{B}}^2}\widehat{\text{Loss}}_{\text{Node}}^{a,b}, \quad \widehat{\text{Loss}}_{\text{Node}}^{a,b} = -\frac{1}{|\mathcal{V}_{\mathcal{B}}|}\sum_{i\in\mathcal{V}_{\mathcal{B}}}\log\hat{q}(i|i,b,a)
\tag{15}
$$

This reduces the computational complexity to $O(|\mathcal{K}_{\mathcal{B}}|N_{\text{GNN}} + |\mathcal{K}_{\mathcal{B}}|^2|\mathcal{V}_{\mathcal{B}}|^2d)$.

### F.4.2. SAMPLING IN SUPGCL

SupGCL, like standard node-level GCL, also performs batching in the augmentation direction and the node direction. The probability distributions $\hat{p}, \hat{q}$ on the batch regarding augmentation methods are defined as:

$$\hat{p}_\phi(b|a) \triangleq \frac{1}{\hat{C}^a_{\text{Aug}}} e^{\text{sim}_F(Y^a, Y^b)/\tau_a}, \quad \hat{C}^a_{\text{Aug}} \triangleq \sum_{c \in \mathcal{K}_\mathcal{B}} e^{\text{sim}_F(Y^a, Y^c)/\tau_a},$$

$$\hat{q}_\phi(b|a) \triangleq \frac{1}{\hat{D}^a_{\text{Aug}}} e^{\text{sim}_F(Z^a, Z^b)/\tau_a}, \quad \hat{D}^a_{\text{Aug}} \triangleq \sum_{c \in \mathcal{K}_\mathcal{B}} e^{\text{sim}_F(Z^a, Z^c)/\tau_a} \tag{16}$$

Accordingly, the loss function regarding augmentation becomes:

$$\widehat{\text{Loss}}_{\text{Aug}} \triangleq \frac{1}{|\mathcal{K}_\mathcal{B}|} \sum_{a \in \mathcal{K}_\mathcal{B}} D(\hat{p}(*|a)|\hat{q}(*|a)) \tag{17}$$

and its complexity is reduced to $O(|\mathcal{K}_\mathcal{B}|N_{\text{GNN}} + |\mathcal{K}_\mathcal{B}|^2|\mathcal{V}|d)$. Similarly, the SupGCL loss function is naturally derived as follows:

$$\widehat{\text{Loss}}_{\text{SupGCL}} = \mathbb{E}_{a,b \sim \hat{p}_\phi(b|a)U_{\mathcal{K}_\mathcal{B}}(a)}\left[\text{Loss}^{a,b}_{\text{Node}}\right] + \widehat{\text{Loss}}_{\text{Aug}}$$

$$= \frac{1}{|\mathcal{K}_\mathcal{B}|^2} \sum_{a,b \in \mathcal{K}^2_\mathcal{B}} \hat{p}_\phi(b|a)\left(-\frac{1}{\mathcal{V}}\sum_{i \in \mathcal{V}} q_\phi(i|i, b, a) + \log \hat{p}_\phi(b|a) - \log \hat{q}_\phi(b|a)\right) \tag{18}$$

Therefore, the complexity of $\widehat{\text{Loss}}_{\text{SupGCL}}$ is reduced to $O(|\mathcal{K}_\mathcal{B}|N_{\text{GNN}} + |\mathcal{K}_\mathcal{B}|^2|\mathcal{V}|^2d)$.

Although the computational complexity is stated to depend on the square of $|\mathcal{K}_\mathcal{B}|$, the number of samples for the softmax normalization term and the batch size for the loss calculation do not need to be identical. From the perspective of unbiasedness discussed in the next section, increasing the batch size specifically for the normalization terms $\hat{C}^a_{\text{Aug}}$ and $\hat{D}^a_{\text{Aug}}$ leads to a reduction in estimation bias. This study adopts this perspective and employs independent batch sizes for the normalization term and the loss function calculation.

Similarly, node-level batching is also possible. Since the graph-level inner product can be described as the expected value over a uniform distribution in the node direction:

$$\text{sim}_F(Z^a, Z^b) = \sum_{i \in \mathcal{V}} \langle z^a_i | z^b_i \rangle = |\mathcal{V}|\mathbb{E}_{i \in \mathcal{U}_\mathcal{V}}\left[\langle z^a_i | z^b_i \rangle\right] \tag{19}$$

the batched graph-level inner product becomes:

$$\widehat{\text{sim}}_F(Z^a, Z^b) \triangleq \frac{1}{|\mathcal{V}_\mathcal{B}|} \sum_{i \in \mathcal{V}_\mathcal{B}} \langle z^a_i | z^b_i \rangle \tag{20}$$

Using this, the probability distributions $\hat{\hat{p}}, \hat{\hat{q}}$ on the batch concerning augmentation methods and node direction are defined as:

$$\hat{\hat{p}}_\phi(b|a) \triangleq \frac{1}{\hat{\hat{C}}^a_{\text{Aug}}} e^{\widehat{\text{sim}}_F(Y^a, Y^b)/\tau_a}, \quad \hat{\hat{C}}^a_{\text{Aug}} \triangleq \sum_{c \in \mathcal{K}_\mathcal{B}} e^{\widehat{\text{sim}}_F(Y^a, Y^c)/\tau_a}$$

$$\hat{\hat{q}}_\phi(b|a) \triangleq \frac{1}{\hat{\hat{D}}^a_{\text{Aug}}} e^{\widehat{\text{sim}}_F(Z^a, Z^b)/\tau_a}, \quad \hat{\hat{D}}^a_{\text{Aug}} \triangleq \sum_{c \in \mathcal{K}_\mathcal{B}} e^{\widehat{\text{sim}}_F(Z^a, Z^c)/\tau_a} \tag{21}$$

and the loss function regarding augmentation becomes:

$$\widehat{\widehat{\text{Loss}}}_{\text{Aug}} \triangleq \frac{1}{|\mathcal{K}_\mathcal{B}|} \sum_{a \in \mathcal{K}_\mathcal{B}} D(\hat{\hat{p}}(*|a)|\hat{\hat{q}}(*|a)) \tag{22}$$

Thus, the batched SupGCL loss function regarding augmentation methods and node direction is:

$$\widehat{\widehat{\text{Loss}}}_{\text{SupGCL}} = \mathbb{E}_{a,b \sim \hat{p}_\phi(b|a)U_{\mathcal{K}_\mathcal{B}}(a)}\left[\widehat{\text{Loss}}^{a,b}_{\text{Node}}\right] + \widehat{\widehat{\text{Loss}}}_{\text{Aug}} \tag{23}$$

and the complexity is reduced to $O(|\mathcal{K}_\mathcal{B}|N_{\text{GNN}} + |\mathcal{K}_\mathcal{B}|^2|\mathcal{V}_\mathcal{B}|^2 d)$.

These results imply that SupGCL allows for extensions similar to Node-Level batching, and the order of its computational complexity remains unchanged. Therefore, SupGCL can be extended to large-scale graph representation learning and various augmentation methods. In the next section, we will present the statistical characteristics of this batching and compare SupGCL with node-level contrastive learning.

### F.5. Robustness to Sampling

The batch-based computational reduction employed in this study relies on rigorous importance sampling. Consequently, the convergence of each statistic is guaranteed as $\mathcal{K}_\mathcal{B}$ and $\mathcal{V}_\mathcal{B}$ increase. However, in terms of unbiasedness, the behavior differs from that of the node-level loss function.

While the node-level loss function satisfies

$$|\text{Loss}_{\text{Node}} - \text{E}_{\mathcal{K}_\mathcal{B}\sim\mathcal{K}}[\widehat{\text{Loss}_{\text{Node}}}]| = 0, \tag{24}$$

the SupGCL loss function adheres to the following proposition:

**Proposition F.1** (Sampling Error of SupGCL).

$$\left|\text{Loss}_{\text{SupGCL}} - \mathbb{E}_{\mathcal{K}_\mathcal{B}\sim\mathcal{K}}[\widehat{\text{Loss}_{\text{SupGCL}}}]\right| \approx O\left(\frac{|\mathcal{K}| - |\mathcal{K}_\mathcal{B}|}{|\mathcal{K}_\mathcal{B}||\mathcal{K}|} \cdot \frac{1}{|\mathcal{V}|} \cdot \frac{1}{\tau_{\text{a}}^2}\right)$$

See Section F.6 for the proof.

The relationship above indicates that while the node-level loss function is unbiased with respect to sampling, the SupGCL loss function is not. This discrepancy arises from the difference between the augmentation-wise probability distributions $p_\phi(b|a), q_\phi(b|a)$ and their sampled counterparts $\hat{p}_\phi(b|a), \hat{q}_\phi(b|a)$. Consequently, the error converges as the batch size approaches the full set ($\mathcal{K}_\mathcal{B} \to \mathcal{K}$) and as $\tau_{\text{a}}$ increases. Furthermore, the augmentation-wise probability distribution employs the graph-level similarity described in Eq. (10). Since graph-level similarity is the sample mean of node-level similarities, the batching error depends on $1/|\mathcal{V}|$. Additionally, the convergence with respect to $\tau_{\text{a}}$ is linked to the relationship between the proposed method and the conventional $\text{Loss}_{\text{Node}}$:

$$\lim_{\tau_{\text{a}}\to\infty} \text{Loss}_{\text{SupGCL}} = \text{Loss}_{\text{Node}}.$$

On the other hand, Graph Contrastive Learning (GCL) has traditionally employed approximations in the node or sample dimensions.

**Proposition F.2** (Error due to Node Sampling).

$$\left|\widehat{\text{Loss}_{\text{Node}}} - \mathbb{E}_{\mathcal{V}_\mathcal{B}\sim\mathcal{V}}[\widehat{\widehat{\text{Loss}_{\text{Node}}}}]\right| \approx O\left(\frac{|\mathcal{V}| - |\mathcal{V}_\mathcal{B}|}{|\mathcal{V}_\mathcal{B}||\mathcal{V}|} \cdot \frac{1}{\tau_{\text{n}}^2}\right)$$

See Section F.6 for the proof.

The above relationship implies that the loss function derived from the probability distribution $\hat{q}_\phi(j|i, b, a)$, sampled in the node dimension, is not an unbiased estimator. Furthermore, as previously established, the discrepancy due to sampling diminishes as the temperature parameter $\tau_{\text{n}}$ increases (Wang & Liu, 2021).

These two propositions reveal that the accuracy degradation due to sampling in standard GCL (node-level) and the proposed augmentation-wise sampling differs by an order proportional to $1/|\mathcal{V}|$. Thus, the degradation in contrastive learning accuracy caused by our sampling strategy is estimated to be approximately $1/|\mathcal{V}|$ of that observed in node-level sampling. Although exact estimation is impossible solely via sampling error due to dataset dependency, this relationship allows us to interpret the convergence impact of sampling in contrast to node-level instances.

Note that while Propositions 1 and 2 focus on the deviation of the error, the gradients also exhibit the same order of magnitude. This stems from Lemma 2 used in the proofs, which establishes that the deviation of the gradient of error function decomposes into a term dependent on batch-related constants (temperature $\tau$, dimensionality $N$, and batch size $m$) and a term reflecting the GNN influence associated with the gradient method.

In practice, examining the sampling ratios in existing contrastive learning methods for image datasets reveals that the samples referenced in each update step constitute only a fraction of the total data. For instance, in pre-training on ImageNet-1K (approx. $1,281,167$ training images) using SimCLR (Chen et al., 2020), a batch size of 4096 is standard, meaning the number of samples used per step is approximately $0.32\%$ of the total data. Even with a larger batch size of 8192, this ratio remains around $8,192/1,281,167 \approx 0.64\%$. Meanwhile, MoCo v2 (He et al., 2020)) and related methods typically use a batch size of around 256 on ImageNet-1K, yielding a sampling ratio of merely $256/1,281,167 \approx 0.02\%$. Similarly, for SupCon (Khosla et al., 2020) and subsequent SimCLR-based methods on CIFAR-10 (50,000 training images), batch sizes of 256–512 are widely adopted, resulting in a ratio of at most $0.5$–$1\%$.

The same applies to node-level Graph Contrastive Learning (GCL). For small-scale graphs like Cora or Citeseer, methods such as GRACE utilize all nodes as negative samples, effectively performing $100\%$ sampling. However, for large-scale graphs like Reddit (232,965 nodes), sampling is performed in mini-batch units (e.g., 256 nodes) following GraphSAGE (Hamilton et al., 2017), restricting the ratio to approximately $0.1\%$ ((Xia et al., 2022b). Comparing these trends, the augmentation-wise sampling ratio in SupGCL (8 batch size / $768 \sim 948$ node $= 0.84 \sim 1.04\%$) falls within or exceeds the range of general CL methods, indicating that it is at a practically reasonable level from a contrastive learning perspective.

Calculating the order of the error upper bound $\frac{|\mathcal{V}|-|\mathcal{V}_\mathcal{B}|}{|\mathcal{V}_\mathcal{B}||\mathcal{V}|}$ based on Proposition F.2 yields the comparisons in Table 13. As shown, the order of the error upper bound for the proposed SupGCL is comparable to that of SimCLR with a batch size of $8,192$ and is sufficiently lower than that of other methods. In other words, in terms of the error upper bound order, the batch size in the augmentation direction employed in this study can be interpreted as sufficiently robust for practical use.

*Table 13.* Comparison of Sampling Errors Across Methods

| Method | No. of Nodes or Samples: $\lvert\mathcal{V}\rvert$ $(\lvert\mathcal{K}\rvert)$ | Batch Size (Node/Sample-level): $\lvert\mathcal{V}_\mathcal{B}\rvert$ or $\lvert\mathcal{K}_\mathcal{B}\rvert$ | Sampling Ratio | Order of Sampling deviation $\frac{\lvert\mathcal{V}\rvert-\lvert\mathcal{V}_\mathcal{B}\rvert}{\lvert\mathcal{V}_\mathcal{B}\rvert\lvert\mathcal{V}\rvert}$ or $\frac{\lvert\mathcal{K}\rvert-\lvert\mathcal{K}_\mathcal{B}\rvert}{\lvert\mathcal{K}_\mathcal{B}\rvert\lvert\mathcal{K}\rvert} \cdot \frac{1}{\lvert\mathcal{V}\rvert}$ |
|---|---|---|---|---|
| SimCLR(Chen et al., 2020) | $1,281,167$ sample | $4096 \sim 8192$ sample | $0.31 \sim 0.63\%$ | $(1.21 \sim 2.43) \times 10^{-4}$ |
| Moco v2(He et al., 2020) | $1,281,167$ sample | 256 sample | $0.01\%$ | $3.91 \times 10^{-3}$ |
| SupCon(Khosla et al., 2020) | $1,281,167$ sample | $256 \sim 512$ sample | $0.01 \sim 0.03\%$ | $(1.95 \sim 3.91) \times 10^{-3}$ |
| ProGCL(Xia et al., 2022b) | $232,965$ node | 256 node | $0.10\%$ | $3.90 \times 10^{-3}$ |
| **SupGCL (Ours)** | $975(769 \sim 946)$ node | 8 node | $0.84 \sim 1.04\%$ | $(1.26 \sim 1.27) \times 10^{-4}$ |

Additionally, regarding the temperature parameter, the settings for SupGCL remain within the scale of existing CL literature. In SimCLR, a temperature of $\tau \approx 0.5$ is standard, and recent reports suggest that lowering it to $\tau \approx 0.25$ can improve performance (Chen et al., 2020). Similarly, MoCo v2 (He et al., 2020) widely adopts $\tau \approx 0.2$. In contrast, the augmentation-wise temperature $\tau_\mathrm{a}$ used in SupGCL is set in the range of $0.1$ to $2.0$, which is of the same order as existing methods. Therefore, from the perspective of the temperature parameter, the sampling error of SupGCL is expected to exhibit behavior similar to that of conventional CL.

While the practicality of the error upper bound was theoretically confirmed, we further analyzed the impact of increasing the augmentation-wise batch size in additional experiments. Table 14 compares the performance on Hazard and BP prediction tasks for breast cancer patients. We observed an improvement in accuracy as the batch size (more precisely, the sample size for the normalization terms $\hat{C}^*, \hat{D}^*$) increased. However, due to computational resource constraints, we were unable to perform this calculation across all tasks in this study.

*Table 14.* Downstream task performance vs. Augmentation Batch size

| | Hazard (Breast) | BP (Breast) |
|---|---|---|
| $\lvert\mathcal{K}_\mathcal{B}\rvert = 8$ (Setting in paper) | $0.650 \pm 0.059$ | $0.243 \pm 0.052$ |
| $\lvert\mathcal{K}_\mathcal{B}\rvert = 16$ | $0.654 \pm 0.065$ | $0.243 \pm 0.031$ |
| $\lvert\mathcal{K}_\mathcal{B}\rvert = 32$ | $0.664 \pm 0.052$ | $0.241 \pm 0.043$ |
| $\lvert\mathcal{K}_\mathcal{B}\rvert = 40$ | $0.663 \pm 0.048$ | $0.251 \pm 0.039$ |

### F.6. Proof of Bias Order

The two propositions regarding the error upper bounds presented above are derived from the following lemma on the approximate convergence of KL divergence.

**Lemma F.3.** *Let the conditional probability distributions $p(j|i)$ and $q(j|i)$ be defined using the Softmax function over a set $V$ as follows:*

$$p(j|i) = \frac{e^{\beta a_{ij}}}{A_i}, \quad A_i = \sum_{k \in V} e^{\beta a_{ik}}, \tag{25}$$

$$q(j|i) = \frac{e^{\beta b_{ij}}}{B_i}, \quad B_i = \sum_{k \in V} e^{\beta b_{ik}}. \tag{26}$$

*Define the loss function $L$ as the average Kullback-Leibler (KL) divergence:*

$$L = \frac{1}{|V|} \sum_{i \in V} D_{\mathrm{KL}}(p(\cdot|i) \,\|\, q(\cdot|i)). \tag{27}$$

*If $\beta \ll 1$, the loss function can be approximated as:*

$$L \approx \frac{\beta^2}{2|V|} \sum_{i \in V} \mathrm{Var}_j(a_{ij} - b_{ij}), \tag{28}$$

*where $\mathrm{Var}_j(\cdot)$ denotes the variance with respect to the uniform distribution over $j \in V$.*

*Proof.* We analyze the KL divergence term for a fixed index $i$, denoted as $D_i = D_{\mathrm{KL}}(p(\cdot|i) \,\|\, q(\cdot|i))$. By definition:

$$
\begin{aligned}
D_i &= \sum_{j \in V} p(j|i) \ln \frac{p(j|i)}{q(j|i)} \\
&= \sum_{j \in V} p(j|i) \left( (\beta a_{ij} - \ln A_i) - (\beta b_{ij} - \ln B_i) \right) \\
&= \beta \sum_{j \in V} p(j|i)(a_{ij} - b_{ij}) - (\ln A_i - \ln B_i).
\end{aligned} \tag{29}
$$

Since $\beta \ll 1$, we employ the Taylor expansion for the exponential terms and the logarithm of the partition function. Let $N = |V|$. The partition function $A_i$ is expanded as:

$$
\begin{aligned}
A_i &= \sum_{j \in V} \left( 1 + \beta a_{ij} + \frac{\beta^2}{2} a_{ij}^2 + \mathcal{O}(\beta^3) \right) \\
&= N \left( 1 + \beta \mu_{a_i} + \frac{\beta^2}{2} m_{a_i}^{(2)} + \mathcal{O}(\beta^3) \right),
\end{aligned} \tag{30}
$$

where $\mu_{a_i} = \frac{1}{N} \sum_j a_{ij}$ and $m_{a_i}^{(2)} = \frac{1}{N} \sum_j a_{ij}^2$ are the mean and second moment with respect to the uniform distribution. Using the approximation $\ln(1 + x) \approx x - \frac{x^2}{2}$, the log-partition function becomes:

$$
\begin{aligned}
\ln A_i &\approx \ln N + \left( \beta \mu_{a_i} + \frac{\beta^2}{2} m_{a_i}^{(2)} \right) - \frac{1}{2}(\beta \mu_{a_i})^2 \\
&= \ln N + \beta \mu_{a_i} + \frac{\beta^2}{2} \underbrace{(m_{a_i}^{(2)} - \mu_{a_i}^2)}_{\sigma_{a_i}^2},
\end{aligned} \tag{31}
$$

where $\sigma_{a_i}^2 = \mathrm{Var}_j(a_{ij})$ under the uniform distribution. Similarly, for $B_i$, we have:

$$\ln B_i \approx \ln N + \beta \mu_{b_i} + \frac{\beta^2}{2} \sigma_{b_i}^2. \tag{32}$$

Next, we approximate the expectation term $\sum_j p(j|i)(a_{ij} - b_{ij})$. First, approximate $p(j|i)$ up to $\mathcal{O}(\beta)$:

$$p(j|i) = \frac{e^{\beta a_{ij}}}{A_i} \approx \frac{1 + \beta a_{ij}}{N(1 + \beta \mu_{a_i})} \approx \frac{1}{N}(1 + \beta a_{ij})(1 - \beta \mu_{a_i})$$

$$\approx \frac{1}{N}\left(1 + \beta(a_{ij} - \mu_{a_i})\right). \tag{33}$$

Let $\Delta_{ij} = a_{ij} - b_{ij}$. The expectation is:

$$\sum_{j \in V} p(j|i)\Delta_{ij} \approx \frac{1}{N}\sum_{j \in V}(1 + \beta(a_{ij} - \mu_{a_i}))\Delta_{ij}$$

$$= \frac{1}{N}\sum_{j \in V}\Delta_{ij} + \beta\left(\frac{1}{N}\sum_{j \in V}a_{ij}\Delta_{ij} - \mu_{a_i}\frac{1}{N}\sum_{j \in V}\Delta_{ij}\right)$$

$$= (\mu_{a_i} - \mu_{b_i}) + \beta\mathrm{Cov}_j(a_{ij}, \Delta_{ij}), \tag{34}$$

where $\mathrm{Cov}_j$ denotes covariance under the uniform distribution. Note that $\mathrm{Cov}_j(a_{ij}, a_{ij} - b_{ij}) = \sigma_{a_i}^2 - \mathrm{Cov}_j(a_{ij}, b_{ij})$.

Substituting equation 31, equation 32, and equation 34 back into equation 29:

$$D_i \approx \beta\left[(\mu_{a_i} - \mu_{b_i}) + \beta(\sigma_{a_i}^2 - \mathrm{Cov}_j(a_{ij}, b_{ij}))\right]$$

$$- \left[(\ln N + \beta\mu_{a_i} + \frac{\beta^2}{2}\sigma_{a_i}^2) - (\ln N + \beta\mu_{b_i} + \frac{\beta^2}{2}\sigma_{b_i}^2)\right]. \tag{35}$$

The first-order terms $\beta(\mu_{a_i} - \mu_{b_i})$ cancel out. Collecting the $\beta^2$ terms:

$$D_i \approx \beta^2\left(\sigma_{a_i}^2 - \mathrm{Cov}_j(a_{ij}, b_{ij}) - \frac{1}{2}\sigma_{a_i}^2 + \frac{1}{2}\sigma_{b_i}^2\right)$$

$$= \frac{\beta^2}{2}\left(\sigma_{a_i}^2 - 2\mathrm{Cov}_j(a_{ij}, b_{ij}) + \sigma_{b_i}^2\right)$$

$$= \frac{\beta^2}{2}\mathrm{Var}_j(a_{ij} - b_{ij}). \tag{36}$$

Finally, averaging over all $i \in V$:

$$L = \frac{1}{|V|}\sum_{i \in V}D_i \approx \frac{\beta^2}{2|V|}\sum_{i \in V}\mathrm{Var}_j(a_{ij} - b_{ij}). \tag{37}$$

$\square$

**Lemma F.4.** *Define the KL divergence within a batch $U \subset V$ as:*

$$\hat{p}(j|i) = \frac{1}{\hat{A}_i}e^{\beta a_{ij}}, \quad \hat{A}_i = \sum_{j \in U}e^{\beta a_{ij}}$$

$$\hat{q}(j|i) = \frac{1}{\hat{B}_i}e^{\beta b_{ij}}, \quad \hat{B}_i = \sum_{j \in U}e^{\beta b_{ij}}$$

$$\hat{L} = \frac{1}{|U|}\sum_{i \in U}D_{\mathrm{KL}}(\hat{p}(j|i)|\hat{q}(j|i))$$

*Then, the following approximation holds:*

$$L - \mathbb{E}_B[\hat{L}] \approx \frac{|V| - |U|}{(|V| - 1)|U|} \cdot \frac{\beta^2}{2} \cdot \frac{1}{|V|}\sum_{i \in V}\mathrm{Var}_j(a_{ij} - b_{ij}). \tag{38}$$

*Proof.* Let $N = |V|$ and $M = |U|$. By applying the result of the previous Lemma to the batch $U$, the batch loss $\hat{L}$ for a specific batch $U$ can be approximated as:

$$\hat{L} \approx \frac{\beta^2}{2M} \sum_{i \in U} \text{Var}_{j \in U}(\Delta_{ij}), \tag{39}$$

where $\Delta_{ij} = a_{ij} - b_{ij}$, and $\text{Var}_{j \in U}(\cdot)$ denotes the variance over the set $U$ (sample variance). We consider the expectation of $\hat{L}$ with respect to the random choice of the batch $U$ (sampled uniformly without replacement from $V$). By the linearity of expectation:

$$\mathbb{E}_U[\hat{L}] \approx \frac{\beta^2}{2} \mathbb{E}_U \left[ \frac{1}{M} \sum_{i \in U} \hat{\sigma}_i^2(U) \right], \tag{40}$$

where $\hat{\sigma}_i^2(U) = \text{Var}_{j \in U}(\Delta_{ij})$. We can rewrite the expectation using an indicator variable $\mathbb{I}(i \in U)$:

$$\mathbb{E}_U \left[ \frac{1}{M} \sum_{i \in V} \mathbb{I}(i \in U) \hat{\sigma}_i^2(U) \right] = \frac{1}{M} \sum_{i \in V} P(i \in U) \mathbb{E}[\hat{\sigma}_i^2(U) \mid i \in U]$$

$$= \frac{1}{M} \sum_{i \in V} \frac{M}{N} \mathbb{E}[\hat{\sigma}_i^2(U) \mid i \in U]$$

$$= \frac{1}{N} \sum_{i \in V} \mathbb{E}[\hat{\sigma}_i^2(U) \mid i \in U].$$

Here, $\mathbb{E}[\hat{\sigma}_i^2(U) \mid i \in U]$ represents the expected variance of the values $\Delta_{ij}$, $j \in U$ when the batch $U$ is a random subset of size $M$ containing $i$. Since the variance is computed over $j \in U$, and $U$ is drawn from $V$, we invoke the relationship between the expected sample variance (biased estimator) and the population variance. Let $\sigma_i^2 = \text{Var}_{j \in V}(\Delta_{ij})$ be the population variance. The expectation of the sample variance defined with denominator $M$ is given by:

$$\mathbb{E}[\hat{\sigma}_i^2(U)] = \frac{M-1}{M} \frac{N}{N-1} \sigma_i^2. \tag{41}$$

Substituting this back into the expression for $\mathbb{E}_U[\hat{L}]$:

$$\mathbb{E}_U[\hat{L}] \approx \frac{\beta^2}{2N} \sum_{i \in V} \left( \frac{M-1}{M} \frac{N}{N-1} \sigma_i^2 \right)$$

$$= \frac{M-1}{M} \frac{N}{N-1} \left( \frac{\beta^2}{2N} \sum_{i \in V} \sigma_i^2 \right)$$

Finally, we compute the difference $L - \mathbb{E}_U[\hat{L}]$:

$$L - \mathbb{E}_U[\hat{L}] \approx \left( 1 - \frac{N(M-1)}{M(N-1)} \right) \left( \frac{\beta^2}{2N} \sum_{i \in V} \sigma_i^2 \right)$$

$$= \frac{N-M}{M(N-1)} \left( \frac{\beta^2}{2N} \sum_{i \in V} \sigma_i^2 \right)$$

This completes the proof. $\square$

The proof above similarly holds for the case where $p(j|i) = \delta_{ij}$, resulting in an error order of $O(\beta^2 \cdot \frac{|V|-|U|}{|V||U|})$. This result supports Proposition F.2 regarding the error order due to node sampling.

Crucially, for the error upper bound of SupGCL, the term $\text{Var}_j(a_{ij} - b_{ij})$ plays a significant role. In the node-level probability density function, $a_{ij}, b_{ij}$ correspond to the similarity between nodes $\langle z_i^* | z_j^* \rangle$, whereas in the augmentation-level

probability density function, $a_{ij}, b_{ij}$ represent the similarity between graphs $\text{sim}_F(Z^a, Z^b)$. Assuming that the similarity of each node is sampled from an identical distribution, i.e.,

$$\langle z_i^a | z_i^b \rangle \sim P^{a,b}, \tag{42}$$

the graph-level similarity can be regarded as the sample mean of the node-level similarities. Therefore, the variance of the graph-level similarity is given by:

$$\text{Var}(\text{sim}_F(Y^a, Y^b) - \text{sim}_F(Z^a, Z^b)) = \text{Var}\left( \frac{1}{|V|} \sum_{i \in V} \langle y_i^a, y_i^b \rangle - \frac{1}{|V|} \sum_{i \in V} \langle z_i^a, z_i^b \rangle \right)$$
$$= \frac{1}{|V|} \text{Var}(\langle y^a, y^b \rangle - \langle z^a, z^b \rangle),$$

where $z^*, y^*$ are random variables representing node-level features. Consequently, the order of the error upper bound for SupGCL satisfies $O(\beta^2 \cdot \frac{|V|-|U|}{|V||U|} \cdot \frac{1}{|V|})$, confirming the validity of Proposition F.1.

In this proof, we adopt the i.i.d. assumption as shown in Eq. 42. This constitutes a strong assumption that the inner products of node-level features follow a probability distribution that is independent of the nodes. However, by employing Hoeffding's inequality, we can relax this assumption due to the boundedness of the feature inner products. Consequently, the relaxed assumption requires only the independence of the node-level distributions, which is the condition for Hoeffding's inequality.

## G. Additional Results

As additional experimental results, we performed the following six analyses:

1. Performance evaluation of the proposed and existing methods using various evaluation metrics.

2. Visualization of the latent states of pre-trained models.

3. Performance comparison with varying embedding dimensions.

4. Statistical significance testing for the performance metrics across downstream tasks.

5. Evaluation on Link Prediction.

6. Comparison of Data Augmentation Strategies

Furthermore, an analysis of the robustness of SupGCL against dataset changes is provided in Appendix I.

### G.1. Additional Evaluation Metrics

In the main paper, we presented results using only Accuracy (or subset accuracy). Below, we report the results using other metrics for the same tasks.

Tables 15 and 16 show the Macro F1-score and Jaccard index results for node-level tasks. Please note that for the cancer-related classification task, we evaluate only the F1-score because it involves binary data. Additionally, Table 17 shows the Macro F1-score for subtype classification in breast cancer. While our proposed method, SupGCL, did not individually achieve state-of-the-art results across all tasks and metrics, it demonstrated the most balanced performance overall.

### G.2. Additional Latent Space Analysis

**Node-level latent Space:** Previously, in Sec 5.2, we visualized the graph-level embedding space generated by pre-trained models (Figure 3). Node-level results for the breast, lung, and colorectal cancer datasets are presented in Figure 4. Since subtype data were unavailable for the lung and colorectal cancer datasets, only node-level latent space visualizations are presented for these cancers.

These results confirm that both GRACE and our proposed method, SupGCL, yield stable latent representations for these cancers, with no observed latent space collapse. A more detailed analysis on latent space collapse is provided in the next section. Furthermore, a more detailed analysis of the latent space from a biological perspective is provided in Appendix J

*Table 15.* Node-level downstream task: macro F1-score

| Task | w/o-pretrain | GAE | GraphCL | GRACE | SGRL | SupGCL |
|---|---|---|---|---|---|---|
| **BP.** | | | | | | |
| Breast | 0.553±0.024 | 0.551±0.034 | 0.540±0.045 | 0.558±0.022 | 0.543±0.022 | **0.571±0.025** |
| Lung | 0.538±0.039 | 0.546±0.021 | **0.584±0.065** | 0.555±0.026 | 0.549±0.023 | 0.546±0.031 |
| Colorectal | 0.514±0.053 | 0.550±0.025 | 0.516±0.033 | **0.560±0.042** | 0.560±0.040 | 0.547±0.038 |
| **CC.** | | | | | | |
| Breast | 0.404±0.036 | 0.378±0.021 | 0.336±0.018 | 0.362±0.040 | 0.384±0.037 | **0.418±0.024** |
| Lung | 0.349±0.086 | **0.395±0.023** | 0.376±0.072 | 0.393±0.026 | 0.385±0.026 | 0.387±0.028 |
| Colorectal | 0.288±0.060 | **0.403±0.032** | 0.265±0.029 | 0.372±0.047 | 0.401±0.049 | 0.397±0.030 |
| **Rel.** | | | | | | |
| Breast | 0.523±0.094 | 0.571±0.048 | 0.593±0.072 | 0.591±0.038 | 0.578±0.067 | **0.610±0.070** |
| Lung | 0.507±0.117 | 0.559±0.045 | 0.538±0.236 | 0.535±0.139 | 0.575±0.061 | **0.592±0.067** |
| Colorectal | 0.474±0.242 | 0.582±0.081 | 0.556±0.124 | 0.547±0.197 | 0.569±0.145 | **0.596±0.060** |

*Table 16.* Node-level downstream task: Jaccard index

| Task | w/o-pretrain | GAE | GraphCL | GRACE | SGRL | SupGCL |
|---|---|---|---|---|---|---|
| **BP.** | | | | | | |
| Breast | 0.490±0.017 | 0.487±0.028 | 0.454±0.046 | 0.478±0.037 | 0.468±0.028 | **0.500±0.035** |
| Lung | **0.539±0.030** | 0.494±0.034 | 0.484±0.030 | 0.510±0.051 | 0.479±0.019 | 0.518±0.027 |
| Colorectal | **0.537±0.031** | 0.506±0.019 | 0.500±0.036 | 0.514±0.024 | 0.469±0.030 | 0.502±0.022 |
| **CC.** | | | | | | |
| Breast | 0.402±0.052 | 0.378±0.021 | 0.303±0.028 | 0.359±0.028 | 0.377±0.029 | **0.422±0.028** |
| Lung | 0.387±0.040 | 0.382±0.035 | 0.321±0.062 | 0.384±0.036 | 0.376±0.031 | **0.392±0.034** |
| Colorectal | 0.377±0.055 | 0.379±0.036 | 0.308±0.067 | 0.388±0.036 | 0.360±0.053 | **0.395±0.033** |

**Analysis of Latent Space Collapse:** To further investigate the characteristics of the node-level latent spaces presented in Sec 5.2, we employed Principal Component Analysis (PCA). Figure 5 displays the PCA-projected latent spaces from pre-trained models on the breast cancer dataset, along with their corresponding explained variance ratios. For GraphCL, which previously exhibited tendencies towards latent space collapse, this analysis confirmed that its PCA explained variance ratio was overwhelmingly concentrated in the first principal component (PC1), accounting for 98.3%.

### G.3. Performance Evaluation across Different Embedding Dimensions

Finally, we investigated the effect of varying embedding dimensions on performance. Figure 6 presents the performance metrics and their corresponding standard deviations across 13 tasks for embedding dimensions of $\{8, 16, 32, 64\}$. Excluding GraphCL, which exhibited instability in generating stable latent spaces, the other five methods showed only marginal performance gains when the embedding dimension was increased from 32 to 64. Furthermore, the proposed method consistently achieved high performance across all tasks and embedding dimensions, experimentally demonstrating its superiority over existing representation learning approaches for biological downstream tasks.

### G.4. Statistical Significance Testing

We further conducted statistical significance testing for the performance metrics across downstream tasks. Specifically, we computed Bonferroni-corrected p-values for pairwise comparisons between our proposed method SupGCL and five baseline methods, using Student's t-test with a significance level of 5%. While most comparisons did not reveal statistically significant differences, we observed significant improvements over GraphCL and SGRL in a subset of node-level tasks. Table 18 reports the tasks where significant differences were found, along with the corresponding p-values.

### G.5. Evaluation on Link Prediction

In addition to node-level and graph-level downstream tasks, we also evaluated performance on an edge-level task, namely link prediction. While the primary goal of SupGCL is to perform representation learning on already constructed GRNs rather than GRN inference itself, it is nevertheless possible to blind edges during training and use the learned representations

*Table 17.* Macro F1-score for subtype classification

| Task | w/o-pretrain | GAE | GraphCL | GRACE | SGRL | SupGCL |
|---|---|---|---|---|---|---|
| **Subtype**
Breast | 0.626 ± 0.070 | 0.720 ± 0.057 | 0.552 ± 0.089 | 0.761 ± 0.063 | 0.715 ± 0.064 | **0.785 ± 0.056** |

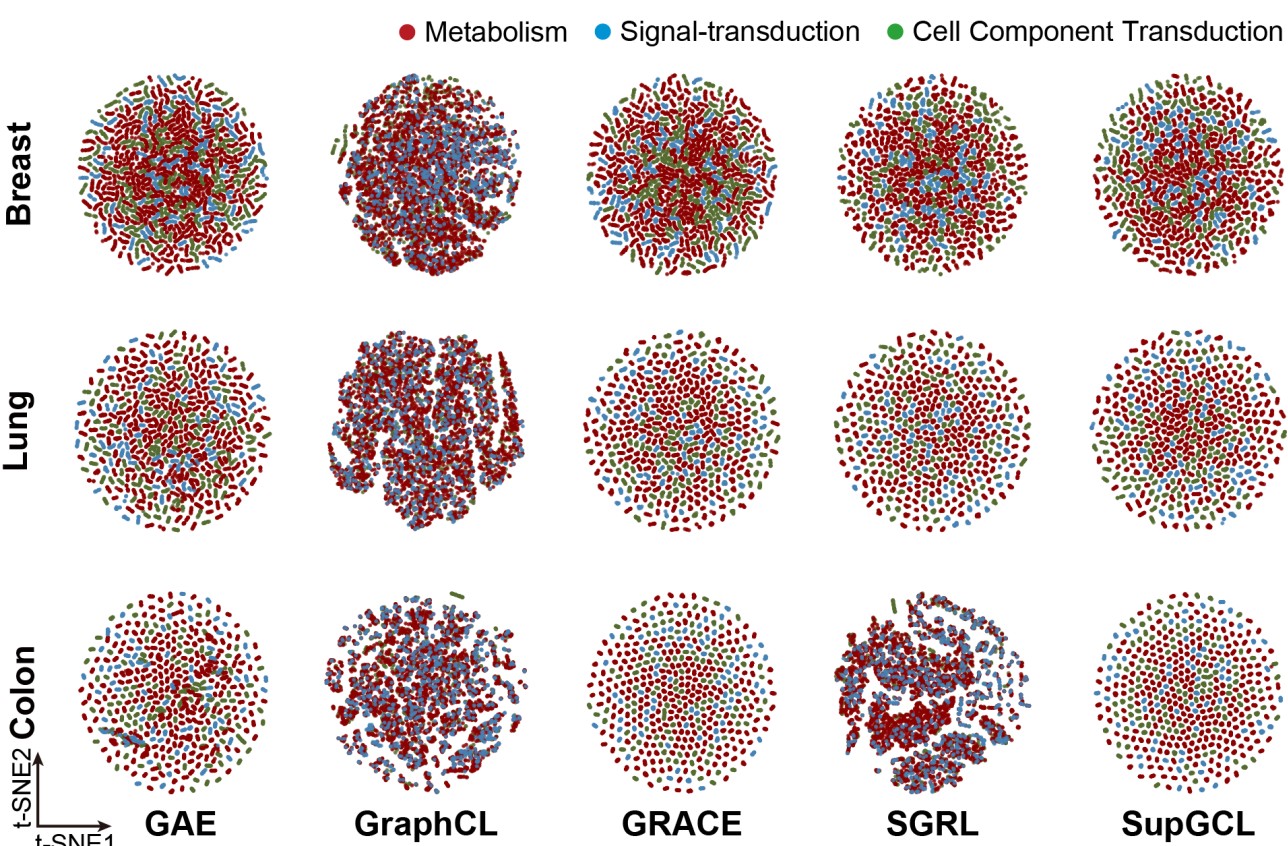

*Figure 4.* t-SNE visualization of pre-trained embeddings on breast, lung, and colorectal cancer GRNs.

to predict the existence of unseen edges. This provides a useful auxiliary evaluation of the learned embeddings.

We conducted experiments on the breast cancer GRN. Edges present in the GRN were treated as positive examples, and absent edges were treated as negatives. From these, 500 positive and 500 negative samples were drawn to form 1000 test cases, which were then split into train:test = 8:2 with a fixed seed. This procedure was repeated across 10 different seeds (0–9). Node embeddings were fed into a two-layer MLP head, and the inner product between resulting vectors was passed through a sigmoid function to predict edge existence. Binary cross-entropy (BCE) was used as the loss function, with target edges masked during encoding to simulate inference of unseen edges. Optimization settings were aligned with those of other tasks and kept consistent across all methods. We report Accuracy and F1 score as evaluation metrics, averaged over the 10 seeds with standard deviations.

Recent studies have also proposed supervised GNN-based approaches specifically designed for GRN inference. A representative example is GCLink (Yu et al., 2025), which directly learns from ground-truth transcriptional networks. In contrast, our study focuses on GRNs estimated from gene expression via Bayesian network inference, which encode statistical causal relationships and are not restricted to transcription factor–regulon interactions. For reference, we additionally report results of applying a GCLink-style model to our setting. Note that the GCLink results in Table 19 correspond not to the original setup (supervised by ground-truth transcriptional networks), but to a modified variant adapted to our estimated GRNs.

The results (Table 19) show that SupGCL outperforms existing baselines on edge-level GRN inference tasks as well. In particular, SupGCL achieved the highest Accuracy and F1 score compared to all other models, including GRACE.

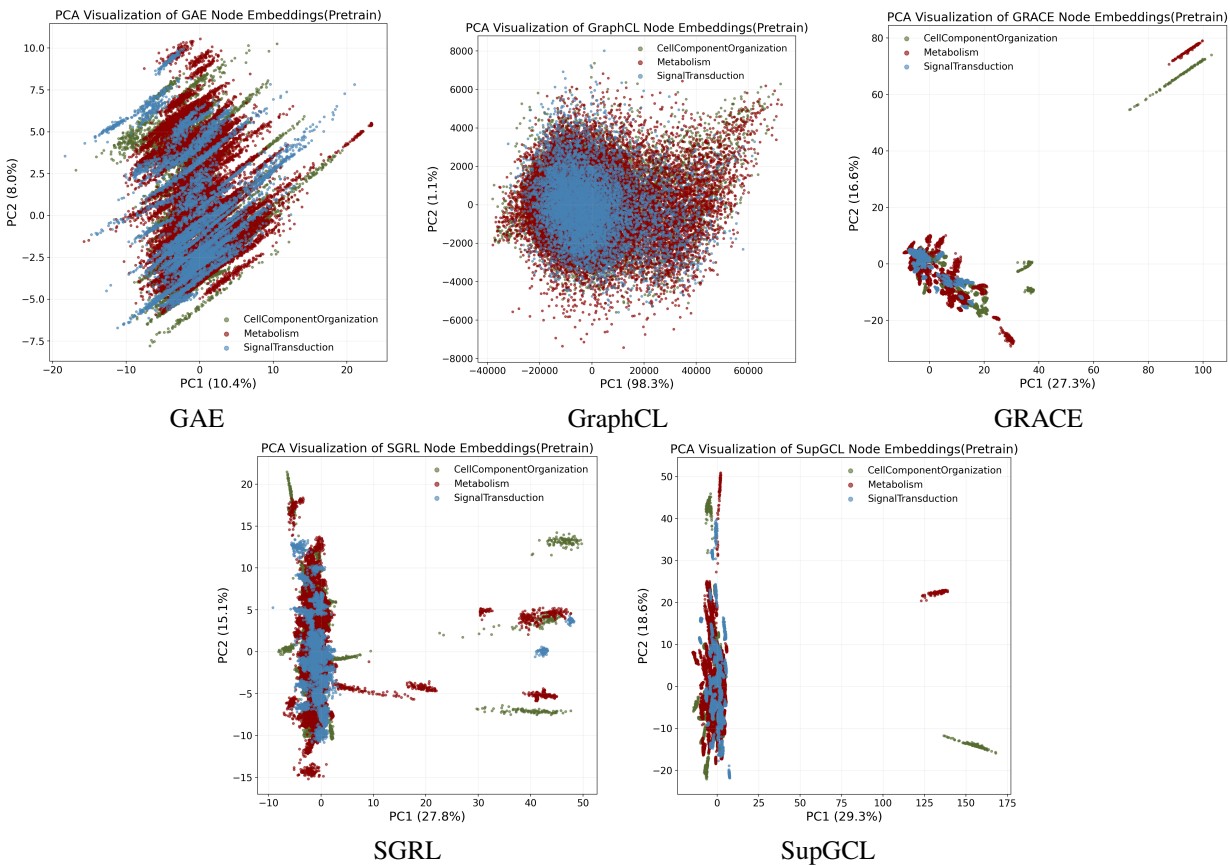

*Figure 5.* PCA analysis of the latent spaces of pre-trained models.

*Table 18.* Comparison of SupGCL with other methods using Bonferroni-corrected p-values.

| Task | Cancer Subtype | Compared Method | p-value |
|------|----------------|-----------------|---------|
| BP | Lung | GraphCL | $1.88 \times 10^{-3}$ |
| BP | Colorectal | SGRL | $1.65 \times 10^{-2}$ |
| CC | Breast | GraphCL | $1.43 \times 10^{-2}$ |
| CC | Lung | GraphCL | $4.01 \times 10^{-3}$ |

Performance was also comparable to or slightly better than GCLink. These findings suggest that although SupGCL is primarily designed for representation learning, it remains competitive for GRN inference tasks such as link prediction.

*Table 19.* Performance on link prediction (Breast Cancer GRN).

| Method | Accuracy | F1 score |
|--------|----------|----------|
| w/o pretrain | $0.730 \pm 0.021$ | $0.843 \pm 0.014$ |
| GAE | $0.743 \pm 0.024$ | $0.846 \pm 0.014$ |
| GraphCL | $0.714 \pm 0.031$ | $0.824 \pm 0.022$ |
| GRACE | $0.757 \pm 0.031$ | $0.848 \pm 0.016$ |
| SGRL | $0.741 \pm 0.032$ | $0.835 \pm 0.023$ |
| GCLink | $0.756 \pm 0.031$ | $0.853 \pm 0.018$ |
| SupGCL (Ours) | $\mathbf{0.763 \pm 0.028}$ | $\mathbf{0.855 \pm 0.018}$ |

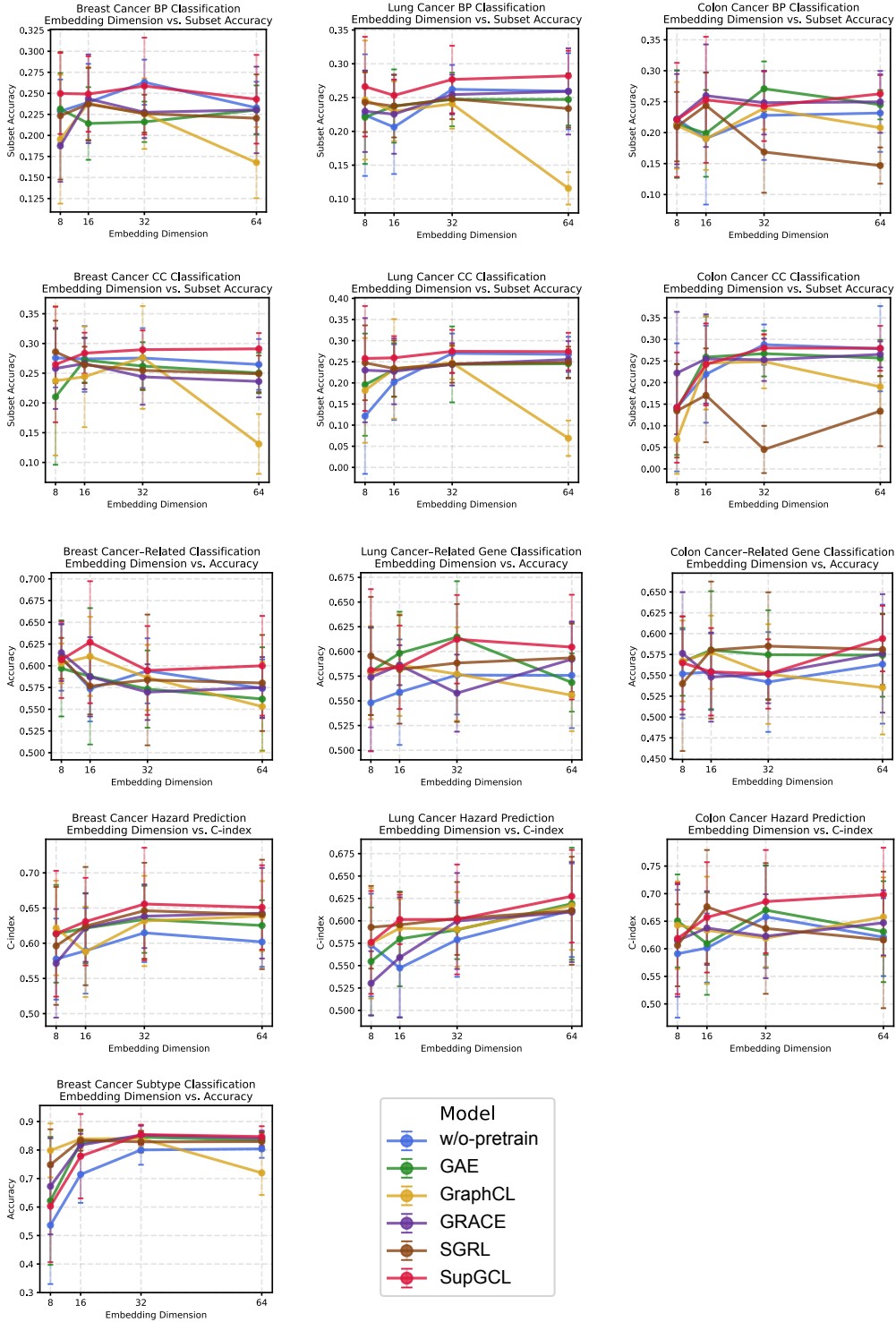

*Figure 6.* Embedding dimension analysis. This figure shows the performance changes across 13 tasks as the embedding dimension varies. The left column shows the results for breast cancer, the center column for lung cancer, and the right column for colorectal cancer. The first row presents the subset accuracy of BP classification, the second row shows the subset accuracy of CC classification, the third row displays the accuracy of cancer-related gene classification, the fourth row indicates the C-index for hazard prediction, and the fifth row shows the results of subtype classification.

## G.6. Additional Ablation Studies

To complement the breast cancer results reported in Table 2 of the main text, we report additional ablation results on lung and colon cancer datasets under different values of the augmentation-level temperature parameter $\tau_a$. The parameter $\tau_a$ controls the strength of the supervised augmentation information induced by the reference model. As $\tau_a$ increases, the influence of the supervised information is weakened. The limiting case $\tau_a = \infty$ corresponds to removing augmentation labels from the supervised contrastive objective. The results in Tables 20 and 21 show that the performance at $\tau_a = \infty$ is comparable to that of GRACE. This trend is consistent with the theoretical observation that SupGCL approaches an unsupervised node-level graph contrastive learning objective as $\tau_a$ increases.

*Table 20.* Lung Cancer Performance comparison across BP, CC, REL, and HAZARD.

| Method | BP | CC | REL | HAZARD |
|---|---|---|---|---|
| $\tau = 0.1$ | $0.262 \pm 0.043$ | $0.271 \pm 0.053$ | $0.603 \pm 0.051$ | $0.625 \pm 0.066$ |
| $\tau = 0.25$ | $0.282 \pm 0.037$ | $0.274 \pm 0.044$ | $0.604 \pm 0.053$ | $0.627 \pm 0.051$ |
| $\tau = 0.5$ | $0.263 \pm 0.036$ | $0.279 \pm 0.067$ | $0.624 \pm 0.069$ | $0.611 \pm 0.033$ |
| $\tau = 1.0$ | $0.273 \pm 0.051$ | $0.278 \pm 0.050$ | $0.611 \pm 0.033$ | $0.614 \pm 0.041$ |
| $\tau = 2.0$ | $0.267 \pm 0.067$ | $0.267 \pm 0.034$ | $0.603 \pm 0.043$ | $0.607 \pm 0.040$ |
| $\tau = \infty$ | $0.253 \pm 0.052$ | $0.254 \pm 0.027$ | $0.589 \pm 0.023$ | $0.596 \pm 0.041$ |
| GRACE | $0.259 \pm 0.063$ | $0.255 \pm 0.043$ | $0.592 \pm 0.038$ | $0.609 \pm 0.055$ |

*Table 21.* Colon Cancer Performance comparison across BP, CC, REL, and HAZARD.

| Method | BP | CC | REL | HAZARD |
|---|---|---|---|---|
| $\tau = 0.1$ | $0.267 \pm 0.041$ | $0.262 \pm 0.042$ | $0.594 \pm 0.050$ | $0.682 \pm 0.074$ |
| $\tau = 0.25$ | $0.262 \pm 0.030$ | $0.279 \pm 0.052$ | $0.594 \pm 0.039$ | $0.698 \pm 0.085$ |
| $\tau = 0.5$ | $0.278 \pm 0.037$ | $0.278 \pm 0.040$ | $0.604 \pm 0.067$ | $0.728 \pm 0.084$ |
| $\tau = 1.0$ | $0.265 \pm 0.033$ | $0.270 \pm 0.044$ | $0.588 \pm 0.077$ | $0.681 \pm 0.116$ |
| $\tau = 2.0$ | $0.276 \pm 0.033$ | $0.262 \pm 0.034$ | $0.585 \pm 0.065$ | $0.673 \pm 0.046$ |
| $\tau = \infty$ | $0.250 \pm 0.041$ | $0.252 \pm 0.035$ | $0.571 \pm 0.061$ | $0.652 \pm 0.043$ |
| GRACE | $0.249 \pm 0.050$ | $0.265 \pm 0.030$ | $0.576 \pm 0.071$ | $0.647 \pm 0.059$ |

# H. Cross-Domain Training

## H.1. Cross-Domain Evaluation Results

Our proposed framework ultimately aims to enable cancer-type–agnostic representation learning of GRNs. SupGCL is scale-free with respect to both the gene set and the set of knockdown perturbations, and can in principle be applied to diverse GRNs derived from different cancer types. Accordingly, pre-training remains feasible even when the patient graphs $\mathcal{G}$ and the supervision graphs $\mathcal{H}_a$ originate from different cancer types. Downstream tasks can also be learned when the cancer types used for pre-training and fine-tuning differ, and it is further possible to apply a model fine-tuned on one cancer type directly to outcome prediction in another. However, the degree to which SupGCL maintains predictive accuracy under such cross-domain conditions requires careful empirical evaluation.

To investigate cancer-type dependency, we conducted three types of cross-domain experiments:

(i) **Pre-training phase**: a setting in which knockdown GRNs from a different cancer type than the patient GRNs are used as supervision.

(ii) **Fine-tuning phase**: a setting in which a model pre-trained on one cancer type is fine-tuned on data from another cancer type.

(iii) **OOD setting**: a setting in which a model pre-trained and fine-tuned on one cancer type is directly applied to patient GRNs from an unseen cancer type.

The configurations and results for SupGCL are summarized in Table 22.

In addition, to examine whether this behavior is specific to SupGCL or common to GCL methods more broadly, we performed the same cross-domain analysis with GRACE, a representative general-purpose GCL model. For a fair comparison, we aligned the settings for (ii) cross-domain fine-tuning and (iii) OOD application with those used for SupGCL and evaluated both methods under identical conditions. The corresponding results for GRACE are reported in Table 23.

For SupGCL, experiment (i) shows that using breast knockdown data as supervision for lung or colorectal patient GRNs yields performance comparable to the in-domain setting, indicating that SupGCL can generalize across cancer types at the pre-training stage when only the teacher GRN domain is changed. In experiment (ii), performance is largely preserved when using lung as the pre-training source and breast as the target for fine-tuning, whereas using colorectal as the source leads to a noticeable degradation. A similar trend is observed in experiment (iii), where directly applying a breast-trained model to lung or colorectal patients results in a substantial drop in performance.

GRACE exhibits similar degradation patterns under the same cross-domain fine-tuning and OOD configurations (Table 23). In particular, cross-domain fine-tuning (Breast→Breast vs. Lung→Breast or Colorectal→Breast) results in only moderate changes in performance, whereas OOD application (Breast / Lung, Breast / Colorectal) leads to pronounced decreases in both hazard prediction and BP classification accuracy. This suggests that the observed behavior is not a limitation unique to SupGCL, but rather reflects a broader challenge for GCL methods on GRNs: models fine-tuned on a single cancer type are not robust when naively transferred to unseen cancer types.

Taken together, these results indicate that SupGCL can maintain reasonable generalization when using knockdown GRNs from different cancer types during pre-training, but—much like GRACE—suffers from performance deterioration when the cancer type changes at the fine-tuning and inference stages. In particular, models fine-tuned on a single cancer type lack robustness when directly applied to unseen cancer types, highlighting the need for learning strategies that more explicitly promote cross-cancer generalization.

*Table 22.* Cross-domain performance of SupGCL across three evaluation settings. (i) **Pre-training phase**: "original" indicates settings in which the cancer types of the patient GRNs and the teacher knockdown GRNs coincide (e.g., Lung / Lung KD, Colorectal / Colorectal KD). "cross domain 1, 2" denote settings in which the cancer types of the patient and teacher GRNs differ, namely Lung / Breast KD and Colorectal / Breast KD, respectively. (ii) **Fine-tuning phase**: "original" indicates a model pre-trained on breast cancer and fine-tuned on breast patient GRNs (Breast→Breast). "cross domain 1, 2" indicate models pre-trained on lung or colorectal GRNs and then fine-tuned on breast patient GRNs (Lung→Breast and Colorectal→Breast). (iii) **OOD setting**: "original" indicates that the cancer types for training (pre-training and fine-tuning) and prediction coincide (e.g., Lung / Lung, Colorectal / Colorectal). "cross domain 1, 2" indicate settings in which a model trained on breast cancer is directly applied to lung or colorectal patients (Breast / Lung, Breast / Colorectal).

| SETTING | CONFIGURATION | HAZARD (C-INDEX) | BP (SUBSET ACC.) |
|---|---|---|---|
| **(I) CROSS DOMAIN AT PRE-TRAINING PHASE: PATIENT / CELL LINE (TEACHER GRAPH)** | | | |
| | ORIGINAL: LUNG / LUNG KD | $0.627 \pm 0.051$ | $0.282 \pm 0.037$ |
| | CROSS DOMAIN 1: LUNG / BREAST KD | $0.633 \pm 0.069$ | $0.270 \pm 0.060$ |
| | ORIGINAL: COLORECTAL / COLORECTAL KD | $0.698 \pm 0.085$ | $0.262 \pm 0.030$ |
| | CROSS DOMAIN 2: COLORECTAL / BREAST KD | $0.687 \pm 0.092$ | $0.257 \pm 0.044$ |
| **(II) CROSS DOMAIN AT FINE-TUNING PHASE: PRE-TRAINED MODEL → DOWNSTREAM TASK** | | | |
| | ORIGINAL: MODEL BREAST→BREAST | $0.650 \pm 0.059$ | $0.243 \pm 0.052$ |
| | CROSS DOMAIN 1: MODEL LUNG→BREAST | $0.654 \pm 0.075$ | $0.248 \pm 0.043$ |
| | CROSS DOMAIN 2: MODEL COLORECTAL→BREAST | $0.632 \pm 0.072$ | $0.249 \pm 0.030$ |
| **(III) OOD: CANCER TYPE OF FINE-TUNING / PREDICTION** | | | |
| | ORIGINAL: LUNG / LUNG | $0.627 \pm 0.051$ | $0.282 \pm 0.037$ |
| | CROSS DOMAIN 1: BREAST / LUNG | $0.603$ | $0.163$ |
| | ORIGINAL: COLORECTAL / COLORECTAL | $0.698 \pm 0.085$ | $0.262 \pm 0.030$ |
| | CROSS DOMAIN 2: BREAST / COLORECTAL | $0.583$ | $0.162$ |

*Table 23.* Cross-domain performance of GRACE

| SETTING | CONFIGURATION | HAZARD (C-INDEX) | BP (SUBSET ACC.) |
|---|---|---|---|
| **(II) CROSS DOMAIN AT FINE-TUNING PHASE: PRE-TRAINED MODEL → DOWNSTREAM TASK** | | | |
| | ORIGINAL: MODEL BREAST→BREAST | $0.642 \pm 0.064$ | $0.230 \pm 0.051$ |
| | CROSS DOMAIN 1: MODEL LUNG→BREAST | $0.610 \pm 0.041$ | $0.225 \pm 0.033$ |
| | CROSS DOMAIN 2: MODEL COLORECTAL→BREAST | $0.615 \pm 0.063$ | $0.232 \pm 0.032$ |
| **(III) OOD: CANCER TYPE OF FINE-TUNING / PREDICTION** | | | |
| | ORIGINAL: LUNG / LUNG | $0.609 \pm 0.055$ | $0.259 \pm 0.063$ |
| | CROSS DOMAIN 1: BREAST / LUNG | $0.534$ | $0.148$ |
| | ORIGINAL: COLORECTAL / COLORECTAL | $0.647 \pm 0.059$ | $0.249 \pm 0.050$ |
| | CROSS DOMAIN 2: BREAST / COLORECTAL | $0.562$ | $0.133$ |

## H.2. Toward Pan-Cancer Representation Learning

The results in Section H.1 show that both SupGCL and GRACE suffer from performance degradation when the cancer type differs at the fine-tuning and inference stages. This suggests that pre-training on a single cancer type has inherent limitations in terms of generalizability and motivates learning strategies that incorporate more diverse biological backgrounds.

To explore the potential of more cancer-type–agnostic representations, we conducted a preliminary pan-cancer pre-training experiment in which GRNs from three cancer types (breast, lung, and colorectal) were jointly used during pre-training. In this setting, only the pre-training phase is multi-cancer (pan-cancer), while fine-tuning and downstream evaluation are restricted to breast cancer data. We then compared this pan-cancer pre-training with the single-cancer pre-training model that uses only breast GRNs.

The results are summarized in Table 24. We found that pan-cancer pre-training does not cause a substantial drop in performance compared with breast-only pre-training. This indicates that integrating multiple cancer types at the pre-training stage can retain, and potentially enrich, the learned representations without harming downstream performance on a specific cancer type. We expect that extending this approach to include a broader set of cancer types and combining it with domain adaptation techniques could further enhance cross-cancer generalization in future work.

*Table 24.* Pan-cancer pre-training vs. single-cancer pre-training (downstream task: breast cancer).

| MODEL SETTING | HAZARD (C-INDEX) | BP (SUBSET ACCURACY) |
|---|---|---|
| BREAST → BREAST | $0.650 \pm 0.059$ | $0.243 \pm 0.052$ |
| PAN-CANCER → BREAST | $0.649 \pm 0.048$ | $0.241 \pm 0.052$ |

# I. Robustness Analysis of SupGCL

In this section, we evaluate the robustness of the proposed SupGCL method. We analyze three aspects: 1. scaling with respect to the number of pretraining samples, 2. the impact of the sample size of supervision graphs, and 3. robustness to noise in the estimated GRNs.

## I.1. Scaling with Respect to Pretraining Data Size

To assess the effect of pretraining sample size, we progressively reduced the number of breast cancer patient GRNs and evaluated performance on downstream tasks. Specifically, we compared the original dataset with conditions using $1/2$ and $1/4$ of the samples, evaluating hazard prediction (graph-level task) and BP classification (node-level task) (Table 25).

The results show a consistent performance improvement in hazard prediction with increasing data size. In contrast, the impact of sample size was marginal for node-level BP classification, suggesting that node-level GCL methods, including SupGCL, can effectively learn node representations even from a limited number of graphs (in some cases a single graph).

*Table 25.* Performance as a function of patient sample size (Breast Cancer GRN).

| Sample Setting | Hazard (c-index) | BP (Subset Accuracy) |
|---|---|---|
| original (N=1092) | $0.650 \pm 0.059$ | $0.243 \pm 0.052$ |
| 1/2 sample (N=546) | $0.640 \pm 0.040$ | $0.243 \pm 0.026$ |
| 1/4 sample (N=273) | $0.631 \pm 0.045$ | $0.247 \pm 0.038$ |

## I.2. Effect of Supervision Graph Sample Size

Next, we examined the effect of reducing the number of knockdown samples used to construct supervision graphs. For breast cancer cell lines, we randomly subsampled the knockdown dataset from 8793 (original) $\rightarrow$ 4397 ($\simeq 1/2$ original) $\rightarrow$ 2199 ($\simeq 1/4$ original) and evaluated downstream task performance (Table 26).

Even with a reduction to one quarter of the original size, performance degradation was minimal. SupGCL targets GRNs with 975 genes, meaning that the 1/4 setting (2199 samples) corresponds to approximately two experiments per gene. This indicates that the method is robust to limited supervision and does not easily overfit, making it suitable for realistic data conditions.

*Table 26.* Performance under reduced supervision graph sample size (Breast Cancer GRN).

| Supervision Sample Setting | Hazard (c-index) | BP (Subset Accuracy) |
|---|---|---|
| original (N=8793) | $0.650 \pm 0.059$ | $0.243 \pm 0.052$ |
| 1/2 sample (N=4397) | $0.656 \pm 0.062$ | $0.239 \pm 0.034$ |
| 1/4 sample (N=2199) | $0.644 \pm 0.053$ | $0.245 \pm 0.043$ |

## I.3. Robustness to Noise in Estimated GRNs

In our framework, edge contribution scores derived from estimated GRNs are used as edge features for the GNN. To evaluate how robust the proposed method is to noise in these estimated GRNs, we conducted additional experiments.

We adopt Bayesian network inference for GRN estimation (see Appendix D for details), which is known to be sensitive to initialization and can introduce errors. To mitigate this, we estimate GRNs 1000 times and apply frequency-based filtering of edges to reduce noise. In the main setting, we apply a uniform cutoff of 5% for both the patient GRN $\mathcal{G}$ and the teacher GRN $\mathcal{H}$. For comparison, we additionally evaluate thresholds of 3% and 10% for both graphs, and also consider an asymmetric configuration in which we retain only higher-confidence edges in the patient GRN while allowing lower-confidence edges to remain in the teacher GRN, i.e., $(\mathcal{G} : 10\%, \mathcal{H} : 3\%)$. The results are summarized in Table 27, and the statistics of each GRN are presented in Table 28.

Overall, differences in downstream performance across cutoff values are small, indicating that SupGCL can pretrain robustly despite uncertainty inherent in GRN estimation. Moreover, even when the teacher GRN contains a non-negligible amount of noise (e.g., in the $(\mathcal{G} : 10\%, \mathcal{H} : 3\%)$ setting), the performance of SupGCL does not substantially degrade, suggesting that the method also exhibits robustness to noise in the perturbation graphs used as supervision.

*Table 27.* Downstream performance under different filtering thresholds in GRN estimation (breast cancer GRN).

| Setting | Hazard (C-index) | BP (Subset Accuracy) |
|---|---|---|
| SupGCL ($\mathcal{G}$:3%, $\mathcal{H}$:3%) | $0.665 \pm 0.059$ | $0.238 \pm 0.029$ |
| SupGCL ($\mathcal{G}$:5%, $\mathcal{H}$:5%, original) | $0.650 \pm 0.059$ | $0.243 \pm 0.052$ |
| SupGCL ($\mathcal{G}$:10%, $\mathcal{H}$:10%) | $0.646 \pm 0.039$ | $0.244 \pm 0.029$ |
| SupGCL ($\mathcal{G}$:10%, $\mathcal{H}$:3%) | $0.649 \pm 0.055$ | $0.242 \pm 0.045$ |

## J. Biological Enrichment Analysis Using Latent Representations

In this section, we investigate the biological interpretability and validity of the gene embeddings learned by GCL models through enrichment analyses based on Gene Ontology (GO) and KEGG. We focus on breast cancer, whose molecular

*Table 28.* Statistics of GRNs constructed with different edge cutoff thresholds

|  | 3% | | 5% | | 10% | |
| --- | --- | --- | --- | --- | --- | --- |
|  | TCGA | LINCS | TCGA | LINCS | TCGA | LINCS |
| number of nodes | 975 | 975 | 975 | 975 | 975 | 975 |
| number of edges | 15405 | 12057 | 13170 | 11209 | 10201 | 10061 |
| average degree | 15.8 | 12.366154 | 13.507692 | 11.49641 | 10.462564 | 10.318974 |

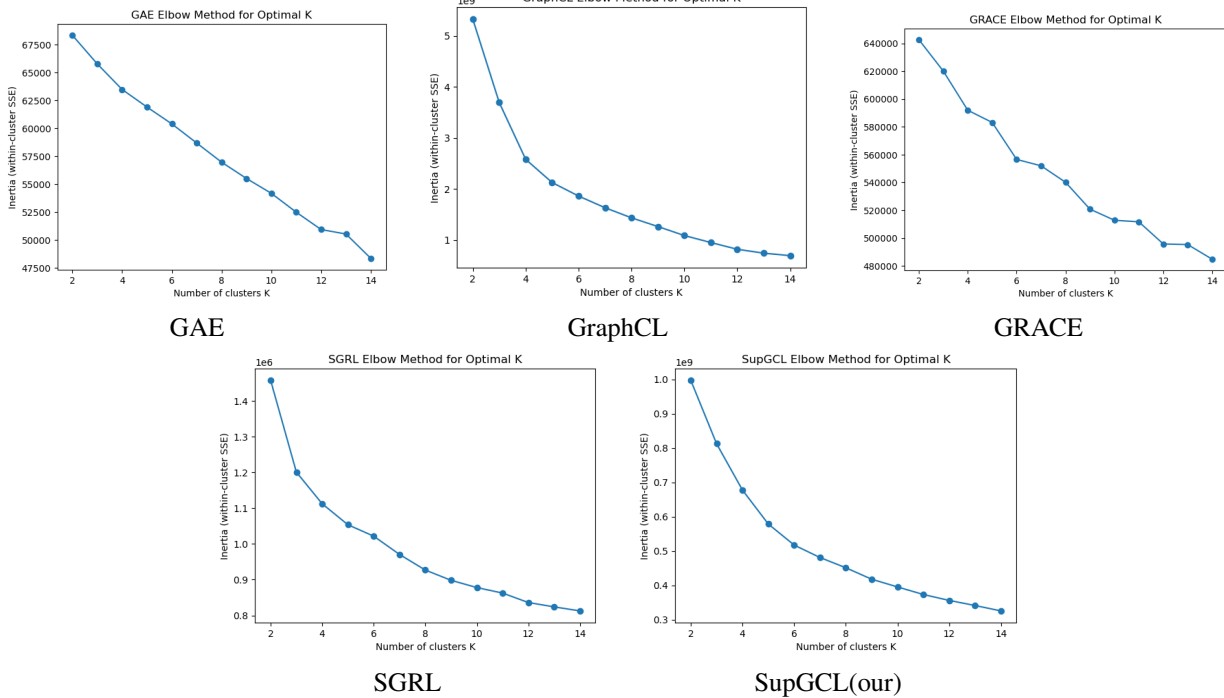

GAE  GraphCL  GRACE

SGRL  SupGCL(our)

*Figure 7.* Elbows of k-means clustering

mechanisms are well characterized, and examine the embeddings obtained by our proposed method and baseline approaches. The analysis consists of three steps: (a) clustering in the latent space, (b) GO enrichment analysis for each cluster, and (c) KEGG pathway enrichment analysis.

### J.1. Clustering in the Latent Space

We applied k-means clustering ($K = 7$) to the gene embeddings obtained from each method and performed GO:BP (Biological Process) and KEGG enrichment analyses using g:Profiler for each cluster. Since embeddings are computed for each node in each patient graph, we used patient-averaged gene embeddings for clustering. The number of clusters ($K = 7$) was determined based on the elbow criterion (see Fig. 7).

#### J.1.1. CLUSTERING RESULTS

The clustering results are shown in Table 29 and Fig. 8. SupGCL and GAE achieved lower Gini Index values compared to other methods, indicating more balanced cluster formation.

### J.2. Gene Ontology Enrichment Analysis

For enrichment analysis, we primarily focus on GAE, a reconstruction-based baseline, and GRACE, which corresponds to an ablation of our proposed method. Clusters containing fewer than five genes were excluded from the analysis.

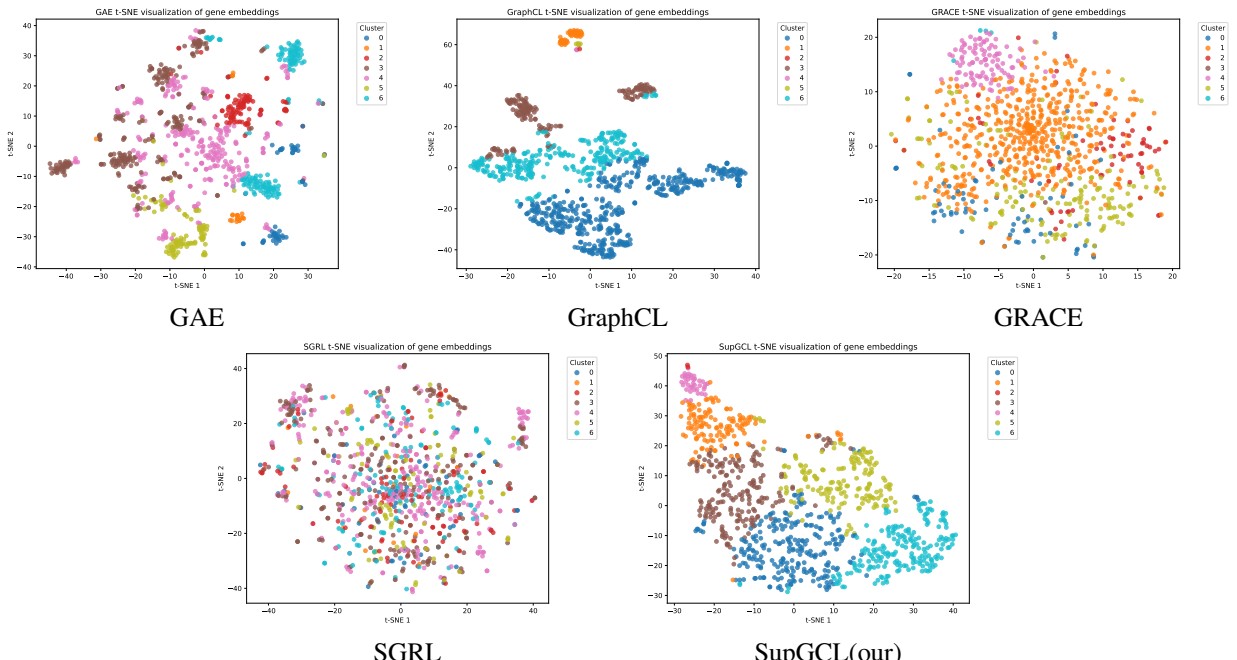

*Figure 8.* Gene-level embedding and the k-means clustering

*Table 29.* Clustering results (number of genes per cluster) for each method.

| Cluster | GAE | GraphCL | GRACE | SGRL | SupGCL |
|---|---|---|---|---|---|
| Cl 0 | 53 | 482 | 73 | 658 | 251 |
| Cl 1 | 22 | 44 | 556 | 11 | 130 |
| Cl 2 | 96 | 1 | 70 | 129 | 2 |
| Cl 3 | 262 | 151 | 1 | 1 | 192 |
| Cl 4 | 293 | 3 | 102 | 25 | 37 |
| Cl 5 | 105 | 6 | 169 | 1 | 160 |
| Cl 6 | 144 | 288 | 4 | 150 | 203 |
| Gini Index | 0.375 | 0.632 | 0.594 | 0.699 | **0.334** |

### J.2.1. GO RESULTS

Table 30 reports the top 5 significantly enriched GO terms for each cluster and method.

Overall, SupGCL and GAE recovered significant functional modules in 6 out of 7 clusters, whereas GRACE succeeded in only 4 clusters. Notably, SupGCL captured distinct separation of autophagy-related processes: regulatory terms (Cl 3) versus execution processes (Cl 6). Cl 0 represented a composite cluster including organelle organization, macromolecule localization, and stress/catabolism. GAE revealed clear boundaries between ROS response, mitophagy, and other biological processes. In contrast, GRACE detected only a limited number of significant terms in most clusters, indicating lower resolution.

*Table 30.* GO enrichment results per cluster (Top 5 terms). "**skip**" indicates clusters with fewer than five genes (excluded), and "—" indicates no significant enrichment.

| Cluster | GAE | GraphCL | GRACE | SGRL | SupGCL |
|---|---|---|---|---|---|
| Cl 0 | Negative regulation of reactive oxygen species metabolic process; Negative regulation of intrinsic apoptotic signaling pathway; Intrinsic apoptotic signaling pathway; Regulation of reactive oxygen species metabolic process; Lymphoid progenitor cell differentiation | Mitotic cell cycle process; Cell cycle process; Cell cycle; Cellular response to stress; Mitotic cell cycle | Intrinsic apoptotic signaling pathway; Regulation of programmed cell death; Intrinsic apoptotic signaling pathway in response to DNA damage; Apoptotic process; Programmed cell death | Positive regulation of cellular process; Positive regulation of biological process; Cellular response to stress; Regulation of cellular component organization; Catabolic process | Organelle organization; Macromolecule localization; Chromosome organization; Catabolic process; Cellular response to stress |
| Cl 1 | — | Antigen processing and presentation of exogenous peptide antigen via MHC class II; Catabolic process | Catabolic process; Cellular response to stress; Positive regulation of cellular process; Protein metabolic process; Regulation of metabolic process | Mitotic cell cycle process; Mitotic cell cycle; Cell cycle phase transition; Mitotic cell cycle phase transition; Cell cycle process | Cell cycle; Positive regulation of hydrolase activity; Cell cycle process; Regulation of cell cycle |
| Cl 2 | Catabolic process; Autophagy of mitochondrion | **skip** | — | Cell cycle process; Cellular response to stress; Chromosome organization; Mitotic cell cycle process; DNA metabolic process | **skip** |
| Cl 3 | Regulation of cellular component organization; Positive regulation of biological process; Cellular response to stress; Organelle organization; Positive regulation of cellular process | Programmed cell death; Cell death; Regulation of programmed cell death; Negative regulation of programmed cell death; Apoptotic process | **skip** | **skip** | Regulation of macroautophagy; Positive regulation of macroautophagy; Positive regulation of autophagy; Positive regulation of organelle assembly; Regulation of autophagy |
| Cl 4 | Positive regulation of cellular process; Cell cycle; Positive regulation of biological process; Regulation of cell cycle; Cell cycle process | **skip** | Mitotic cell cycle process; Mitotic cell cycle; Cell cycle process; Cell cycle; Sister chromatid segregation | Mitotic sister chromatid segregation; Sister chromatid segregation; Mitotic nuclear division; Nuclear chromosome segregation; Mitotic cell cycle | Positive regulation of biological process; System development; Positive regulation of cellular process; Nervous system development; Regulation of multicellular organismal development |
| Cl 5 | Cell migration; Organelle organization; Cellular response to stress; Positive regulation of cellular process | Regulation of calcium ion transport | Catabolic process; Establishment of protein localization; Protein transport | **skip** | Small molecule metabolic process; Small molecule biosynthetic process; Response to endogenous stimulus; Carboxylic acid metabolic process; Oxoacid metabolic process |
| Cl 6 | Macromolecule localization; Establishment of protein localization; Protein transport; Protein localization; Protein localization to organelle | Positive regulation of cellular process; Positive regulation of biological process; Positive regulation of metabolic process; Catabolic process; Regulation of metabolic process | **skip** | Amino acid metabolic process; Catabolic process; Small molecule metabolic process; Apoptotic mitochondrial changes; Sulfur compound metabolic process | Catabolic process; Process utilizing autophagic mechanism; Autophagy; Macroautophagy; Regulation of programmed cell death |

## J.3. KEGG Pathway Enrichment Analysis

### J.3.1. KEGG RESULTS

The KEGG enrichment results are shown in Table 31. SupGCL identified breast cancer–related modules, capturing endocrine resistance, ER stress, and estrogen signaling in Cl 6, and separating FoxO/p53/metabolic pathways into Cl 5. GAE distinguished DNA damage response, cell cycle, and metabolism, reflecting clear functional boundaries. In contrast, GRACE grouped infection-related and metabolic pathways into broad clusters, showing lower resolution.

*Table 31.* KEGG enrichment results per cluster (Top 5 terms). "**skip**" indicates clusters with fewer than five genes (excluded), and "—" indicates no significant enrichment.

| Cluster | GAE | GraphCL | GRACE | SGRL | SupGCL |
|---|---|---|---|---|---|
| Cl 0 | Amino sugar and nucleotide sugar metabolism | Cell cycle; Mismatch repair; Nucleotide excision repair; DNA replication; Terpenoid backbone biosynthesis | Kaposi sarcoma-associated herpesvirus infection; Epstein-Barr virus infection; Hepatitis B; Colorectal cancer; Human immunodeficiency virus 1 infection | MAPK signaling pathway; Epstein-Barr virus infection; Biosynthesis of nucleotide sugars; Valine, leucine and isoleucine degradation; Endocrine resistance | Mismatch repair |
| Cl 1 | — | Lysosome | Protein processing in endoplasmic reticulum; Lysosome; Vibrio cholerae infection | Cell cycle; Motor proteins | — |
| Cl 2 | — | **skip** | Protein processing in ER; Lysosome | — | **skip** |
| Cl 3 | Mismatch repair; Endometrial cancer; Nucleotide excision repair; Platinum drug resistance; DNA replication | Metabolic pathways | — | **skip** | — |
| Cl 4 | Cell cycle; p53 signaling pathway | **skip** | — | Cell cycle | — |
| Cl 5 | — | — | Fructose and mannose metabolism | **skip** | FoxO signaling pathway; p53 signaling pathway; Insulin resistance; Metabolic pathways |
| Cl 6 | Valine, leucine and isoleucine degradation; Arginine and proline metabolism | Colorectal cancer; Endometrial cancer; Pancreatic cancer; Breast cancer; Pathways in cancer | **skip** | Nucleotide excision repair | Protein processing in endoplasmic reticulum; Endocrine resistance; Estrogen signaling pathway; Vibrio cholerae infection |

## J.4. Summary of Enrichment Analysis

In breast cancer samples, SupGCL demonstrated superior biological interpretability by: (i) separating autophagy into regulatory (Cl 3) and execution (Cl 6) processes in GO analysis, (ii) distinguishing endocrine resistance/ER stress (Cl 6) and FoxO/p53/metabolic modules (Cl 5) in KEGG analysis, and (iii) isolating DNA repair (Cl 0) as an independent cluster.

While GAE also delineated major biological functions, it failed to detect breast cancer–specific endocrine pathways as distinct clusters. GRACE exhibited limited resolution, with insufficient reflection of cancer-specific pathway differentiation. These results suggest that SupGCL provides more biologically meaningful and interpretable latent representations in the context of breast cancer.

# K. Comparison with Other Models

*Table 32.* Comparison with additional models (breast cancer). Geneformer condition # 1: freeze all except last layer, # 2: freeze first 2 layers, # 3: full fine-tuning.

| MODEL | HAZARD (C-INDEX) | BP (SUBSET ACCURACY) |
|---|---|---|
| GCA | $0.620 \pm 0.039$ | $0.241 \pm 0.021$ |
| AFGRL | $0.616 \pm 0.037$ | $0.240 \pm 0.026$ |
| SIMGRACE | $0.638 \pm 0.032$ | $0.228 \pm 0.044$ |
| AD-GCL | $0.636 \pm 0.036$ | $0.228 \pm 0.023$ |
| AUTOGCL | $0.620 \pm 0.058$ | $0.214 \pm 0.021$ |
| ARIEL | $0.626 \pm 0.041$ | $0.230 \pm 0.039$ |
| GENEFORMER # 1 | $0.604 \pm 0.096$ | $0.241 \pm 0.076$ |
| GENEFORMER # 2 | $0.595 \pm 0.054$ | $0.230 \pm 0.057$ |
| GENEFORMER # 3 | $0.585 \pm 0.048$ | $0.226 \pm 0.066$ |
| MUSE-GNN (ADAPTED) | $0.639 \pm 0.046$ | $0.238 \pm 0.037$ |
| **SUPGCL (OURS)** | $\mathbf{0.650 \pm 0.059}$ | $\mathbf{0.243 \pm 0.052}$ |

## K.1. Comparison with other graph contrastive learning models

In graph contrastive learning (GCL), topological changes introduced by augmentations such as node or edge dropping are known to degrade performance. To address this issue, recent works have proposed frameworks that either correct or avoid the effects of augmentation. For instance, SGRL (He et al., 2024), the successor to BGRL (Thakoor et al., 2021), has established a state-of-the-art augmentation-free paradigm. Other representative methods include GCA (Zhu et al., 2021), which automatically adjusts and corrects augmentation effects; AFGRL (Lee et al., 2022), which integrates reconstruction and bootstrap learning; and SimGRACE (Xia et al., 2022a), which perturbs model parameters with simple Gaussian noise.

Beyond these, more advanced frameworks explicitly treat augmentations as learnable objects. AD-GCL (Suresh et al., 2021) adopts a min–max adversarial optimization scheme in which the encoder and a perturbation generator compete to search for destructive transformations. AutoGCL (Yin et al., 2022) introduces a learnable policy generator that selects node-wise drop/mask/keep operations, and ArieL (Feng et al., 2024) employs Projected Gradient Descent (PGD) to construct adversarial views together with an information regularization term. All of these methods share the common philosophy of not fixing the augmentation design in advance, but instead optimizing augmentations jointly with the encoder. However, the perturbations learned in these frameworks are not semantic operations grounded in real-world mechanisms; they remain generic regularization noise, so the amount of data that *explains* or constrains the augmentations themselves is not increased.

In contrast, SupGCL differs fundamentally from augmentation-free and augmentation-adjustment approaches. Rather than avoiding topological changes, SupGCL leverages them as informative signals by supervising augmentations with knockdown-derived GRNs. That is, while conventional approaches aim to circumvent or correct augmentation-induced topology shifts, SupGCL explicitly exploits them as positive supervision, transforming a traditionally negative factor into a beneficial learning signal.

Table 32 reports the comparison on breast cancer GRNs for hazard prediction (graph-level) and BP classification (node-level). As shown in the results, SupGCL consistently outperforms representative augmentation-correction and augmentation-free models (GCA, AFGRL, SimGRACE, AD-GCL, AutoGCL, ArieL). This suggests that actively learning from topology changes as supervised signals, grounded in knockdown GRNs, leads to improved representation learning performance.

## K.2. Comparison with biological foundation model (Geneformer)

Recent years have seen rapid progress in the development of biological foundation models. A representative example is Geneformer (Theodoris et al., 2023), a transformer-based foundation model pretrained on tens of millions of single-cell transcriptomes. Although the data modality and task setup differ from those in this work, both SupGCL and Geneformer ultimately aim to learn useful representations of genes. Therefore, comparing SupGCL against such a foundation model is important for positioning and validating our approach.

Geneformer converts each cell's gene expression vector into a sequence of gene tokens ordered by expression rank, and then uses a Transformer to learn context-dependent representations of genes and cells. In this study, we use the publicly available

default model Geneformer-316M (embedding dimension 1152, 18 transformer layers) and finetune it on our task settings. Specifically, we consider three finetuning regimes:

  (i) updating only the final transformer layer and the prediction head,

 (ii) partially finetuning by freezing only the first two encoder layers while updating the remaining layers and the prediction head

(iii) full finetuning, where all layers are updated.

For the node-level task, we feed each patient's bulk expression profile into Geneformer and use the resulting gene embeddings for multi-label GO-BP classification. For the graph-level tasks, we construct patient representations by pooling the CLS token and apply them to breast cancer subtype classification and hazard prediction.

Table 32 summarizes the comparison between SupGCL and Geneformer. On the node-level gene classification task, Geneformer achieves reasonable performance when updating only the last layer or under partial finetuning, confirming the representational power of a transformer-based foundation model. However, for the graph-level tasks of patient prognosis prediction and subtype classification, SupGCL consistently outperforms Geneformer under all finetuning settings, with particularly poor performance observed under full finetuning. This degradation is likely attributable to a distributional mismatch, as Geneformer is designed and pretrained for single-cell transcriptomes, as well as to its inability to directly exploit patient-specific GRN structure.

Overall, while Geneformer—a large-scale transformer foundation model trained on single-cell data—provides competitive performance for node-level functional label prediction, it is outperformed by SupGCL on the GRN-based graph-level tasks that are the focus of this work. In particular, SupGCL's direct use of knockdown GRNs as supervision appears crucial for achieving strong performance on patient-level outcomes.

### K.3. Comparison with MuSeGNN (adapted)

We also compared SupGCL with MuSe-GNN (Liu et al., 2023), a representative multi-omics graph framework. While MuSe-GNN typically integrates distinct omics modalities (e.g., scRNA-seq, scATAC-seq, spatial transcriptomics), we adapted it to our setting. Note that this experimental setup utilizes data types that differ from the specific multi-omics modalities originally assumed by MuSe-GNN, but maintains the core principle of multi-view representation learning.

Specifically, we employed the patient GRNs $\mathcal{G}$ derived from TCGA bulk RNA-seq and the teacher knockdown GRNs $\mathcal{H}^*$ derived from LINCS experiments as the dual input sources. To rigorously implement this comparison, we constructed a MuSe-GNN-adapted architecture with independent encoders $f_\mathcal{G}$ and $f_\mathcal{H}$ for each domain:

$$f_\mathcal{G} : \mathcal{G}_a \mapsto Z^a = \{z_i^a\}_i, \quad f_\mathcal{H} : \mathcal{H}_a \mapsto Y^a = \{y_i^a\}_i$$

In the original MuSe-GNN framework, the total loss is defined as a combination of three components:

$$\text{Loss}_{\text{MuSe}-\text{GNN}} \triangleq \text{Loss}_{\text{Recon}} + \text{Loss}_{\text{Common-Sim}} + \text{Loss}_{\text{Diff-Sim}}, \tag{43}$$

which correspond to (i) edge reconstruction loss, (ii) similarity learning for genes common to both modalities, and (iii) contrastive learning for genes specific to one modality.

In our experimental setting, the node sets for $\mathcal{G}$ and $\mathcal{H}^*$ are identical. Consequently, the modality-specific term $\text{Loss}_{\text{Diff-Sim}}$ is not applicable. We therefore optimized the model using $\text{Loss}_{\text{Recon}}$ and $\text{Loss}_{\text{Common-Sim}}$. For the reconstruction term, we applied the standard graph reconstruction loss to both domains. For the similarity term, we adopted a standard InfoNCE formulation to promote the alignment of the corresponding node embeddings $z_i^a$ and $y_i^a$ across the two domains.

The results are summarized in Table 32. Although the adapted MuSe-GNN demonstrated high performance in both graph-level and node-level tasks, it did not surpass SupGCL. These results suggest that our proposed setting—where the knockdown GRN $\mathcal{H}$ provides a teacher distribution for the learner GRN $\mathcal{G}$—allows SupGCL to formulate the structural relationship more directly and validly than treating them as multi-modal inputs.

# L. Augmentation in GCL

## L.1. Meaning of Augmentation Learning of GCL

In this study, we adopted a supervised approach where biological priors determine the graph perturbations. A natural counter-argument might suggest optimizing the graph augmentation strategy itself—learning to generate the "optimal" augmented graph rather than using fixed rules. Here, we theoretically analyze why learning the augmentation model within our contrastive framework leads to trivial solutions, justifying our choice of fixed biological supervision.

Our proposed SupGCL framework is rigorously grounded in the minimization of the KL divergence between the joint distributions of node pairs $(i, j)$ and augmentation pairs $(a, b)$. The loss function is defined as:

$$\text{Loss}_{\text{GCL}} = D_{\text{KL}}(p_\phi(i, j, a, b) \mid q_\phi(i, j, a, b)).$$

Hereafter, we define $a, b \in \mathcal{K}$ as the indices of the target genes subjected to knockdown experiments, rather than abstract augmentation indices. This allows us to formulate the augmentation generation process as follows.

To discuss "optimal graph augmentation" in this context, one must consider modeling a virtual graph augmentation generator $q_\psi(\mathcal{G}_a \mid \mathcal{G}, a)$, parameterized by $\psi$. Here, $\mathcal{G}_a$ denotes the generated graph augmentation corresponding to the knockdown index $a$, which aims to approximate the biological ground truth $\mathcal{H}_a$. The objective would be to learn this generator $q_\psi$ while simultaneously minimizing the KL divergence over the knockdown set $\mathcal{K}$ and node set $\mathcal{V}$.

However, if the graph augmentation model $q_\psi(\mathcal{G}_a \mid \mathcal{G}, a)$ is not fixed and is trained jointly with the representation parameters $\phi$, the optimization becomes ill-posed. Specifically, the reference distribution $p_\phi(i, j, a, b)$ and the target distribution $q_{\psi,\phi}(i, j, a, b)$ would be optimized via separate parameters without sufficient constraints. This formulation allows the model to converge to a trivial solution.

A representative example of such a trivial solution is the mapping $g_\psi(\mathcal{G}, a) = \mathcal{H}_a$, $\forall \mathcal{G}$. In this scenario, the generator learns to output the teacher graph $\mathcal{H}_a$ directly, completely ignoring the structural information of the input patient graph $\mathcal{G}$. While this minimizes the distributional distance, it results in a "data extension" that is independent of the input graph $\mathcal{G}$. Consequently, the learned representations would fail to capture the unique topological features of the patient-specific networks, rendering them useless for downstream tasks that require patient-specific biological insights.

Therefore, we argue that a mathematical formulation seeking "contrastively effective data augmentation" often results in a "contrastive formulation capable of trivial solutions." Such solutions may be mathematically valid in an unconstrained optimization landscape but are unsuitable for downstream tasks requiring biological fidelity.

While it is technically possible to prevent this collapse by introducing additional constraints—such as reconstruction losses via AutoEncoders or regularization on the weights of $q_\psi$—our goal in this work was to establish the simplest and most interpretable formulation of supervised contrastive learning. Thus, we deliberately excluded learned augmentations in favor of a probabilistic formulation that directly leverages biological experimental data as explicit supervision.

## L.2. Comparison of Data Augmentation Strategies

In this work, we adopt an artificial augmentation scheme in which the node features of a perturbed gene are set to zero and all its incident edges are deleted. The motivation is to mimic the effect of a gene knockdown, which reduces the expression level of a specific gene and weakens its interactions with other genes. Indeed, prior studies on virtual knockdown and in silico gene perturbation employ similar strategies: CellOracle (Yang et al., 2023) simulates genetic perturbations by replacing the expression of the target gene with zero, while GenKI (Kamimoto et al., 2023) and scTenifoldKnk (Osorio et al., 2022) model knockdown effects on GRNs by removing all edges adjacent to the perturbed gene. Our artificial augmentation design follows these precedents.

In addition to this default augmentation, we further evaluated several alternative perturbation modes on the breast cancer dataset: (i) **node copy**: replacing the node features of the target gene with those of another randomly chosen gene; (ii) **in-edge**: deleting only the incoming edges of the target gene; and (iii) **out-edge**: deleting only its outgoing edges. The results are summarized in Table 33.

Overall, the node-copy scheme (i) yields slightly worse performance than the others, but the differences across perturbation

modes are modest. One plausible interpretation is that, whereas knockdown corresponds to a decrease in expression, randomly copying features from another gene can sometimes increase the effective expression of the target gene, thereby inducing perturbations that are misaligned with the underlying biological process. These observations suggest that augmentations that more faithfully reflect the biology of knockdown-like perturbations tend to support better representation learning.

*Table 33.* Performance comparison under different perturbation modes (breast cancer GRN).

| METHOD | HAZARD (C-INDEX) | BP (SUBSET ACCURACY) |
|---|---|---|
| NODE COPY | $0.647 \pm 0.052$ | $0.236 \pm 0.026$ |
| IN-EDGE | $0.653 \pm 0.077$ | $0.242 \pm 0.039$ |
| OUT-EDGE | $0.667 \pm 0.059$ | $0.240 \pm 0.042$ |
| *original setting* | $0.650 \pm 0.059$ | $0.243 \pm 0.052$ |

## Appendix References

Ashburner, M., Ball, C. A., Blake, J. A., Botstein, D., Butler, H., Cherry, J. M., Davis, A. P., Dolinski, K., Dwight, S. S., Eppig, J. T., et al. Gene ontology: tool for the unification of biology. *Nature genetics*, 25(1):25–29, 2000.

Costa, P. R., Acencio, M. L., and Lemke, N. A machine learning approach for genome-wide prediction of morbid and druggable human genes based on systems-level data. In *BMC genomics*, volume 11, pp. 1–15. Springer, 2010.

Friedman, N., Linial, M., Nachman, I., and Pe'er, D. Using Bayesian networks to analyze expression data. In *Proceedings of the fourth annual international conference on Computational molecular biology*, RECOMB '00, pp. 127–135, New York, NY, USA, 2000. Association for Computing Machinery. ISBN 978-1-58113-186-4.

Imoto, S., Kim, S., Goto, T., Aburatani, S., Tashiro, K., Kuhara, S., and Miyano, S. Bayesian network and nonparametric heteroscedastic regression for nonlinear modeling of genetic network. *Journal of bioinformatics and computational biology*, 1(02):231–252, 2003.

Langfelder, P. and Horvath, S. WGCNA: an R package for weighted correlation network analysis. *BMC Bioinformatics*, 9 (1):559, December 2008. ISSN 1471-2105.

Margolin, A. A., Nemenman, I., Basso, K., Wiggins, C., Stolovitzky, G., Favera, R. D., and Califano, A. ARACNE: An Algorithm for the Reconstruction of Gene Regulatory Networks in a Mammalian Cellular Context. *BMC Bioinformatics*, 7(1):S7, March 2006. ISSN 1471-2105.

Paolacci, A. R., Tanzarella, O. A., Porceddu, E., and Ciaffi, M. Identification and validation of reference genes for quantitative rt-pcr normalization in wheat. *BMC molecular biology*, 10:1–27, 2009.

Shu, H., Zhou, J., Lian, Q., Li, H., Zhao, D., Zeng, J., and Ma, J. Modeling gene regulatory networks using neural network architectures. *Nature Computational Science*, 1(7):491–501, July 2021. ISSN 2662-8457. Publisher: Nature Publishing Group.

Subramanian. GEO: Gene Expression Omnibus, GSE92742: L1000 phase I landmark gene expression profiles, 2017. Accessed: 2025-05-15.

Tamada, Y., Shimamura, T., Yamaguchi, R., Imoto, S., Nagasaki, M., and Miyano, S. Sign: Large-Scale Gene Network Estimation Environment for High Performance Computing. *Genome Informatics*, 25(1):40–52, 2011.

Thakoor, S., Tallec, C., Azar, M. G., Azabou, M., Dyer, E. L., Munos, R., Veličković, P., and Valko, M. Large-scale representation learning on graphs via bootstrapping. *arXiv preprint arXiv:2102.06514*, 2021.

Theodoris, C. V., Xiao, L., Chopra, A., Chaffin, M. D., Al Sayed, Z. R., Hill, M. C., Mantineo, H., Brydon, E. M., Zeng, Z., Liu, X. S., et al. Transfer learning enables predictions in network biology. *Nature*, 618(7965):616–624, 2023.

Tomoto, M., Mineharu, Y., Sato, N., Tamada, Y., Nogami-Itoh, M., Kuroda, M., Adachi, J., Takeda, Y., Mizuguchi, K., Kumanogoh, A., Natsume-Kitatani, Y., and Okuno, Y. Idiopathic pulmonary fibrosis-specific Bayesian network integrating extracellular vesicle proteome and clinical information. *Scientific Reports*, 14(1):1315, January 2024. ISSN 2045-2322. Publisher: Nature Publishing Group.

UCSC Xena. TCGA TARGET GTEx study data. Available from UCSC Xena Platform, 2016. Accessed: 2025-05-15.

Xin, J., Mark, A., Afrasiabi, C., Tsueng, G., Juchler, M., Gopal, N., Stupp, G. S., Putman, T. E., Ainscough, B. J., Griffith, O. L., et al. Mygene. info and myvariant. info: gene and variant annotation query services. *bioRxiv*, pp. 035667, 2015.

Yahara, H., Yanamoto, S., Takahashi, M., Hamada, Y., Sakamoto, H., Asaka, T., Kitagawa, Y., Moridera, K., Noguchi, K., Sugiyama, M., Maruoka, Y., and Yahara, K. Whole blood transcriptome profiling identifies gene expression subnetworks and a key gene characteristic of the rare type of osteomyelitis. *Biochemistry and Biophysics Reports*, 32:101328, December 2022. ISSN 2405-5808.

Zhu, Y., Xu, Y., Yu, F., Liu, Q., Wu, S., and Wang, L. Graph contrastive learning with adaptive augmentation. In *Proceedings of the Web Conference 2021*, WWW '21, pp. 2069–2080. ACM, April 2021.

