# OpenReview forum: "Supervised Graph Contrastive Learning for Gene Regulatory Networks"
_ICML.cc/2026/Conference — ICML 2026 regular_

### Official Review · Reviewer_x9Mr · 2026-03-12

**Soundness:** 3
**Presentation:** 3
**Significance:** 2
**Originality:** 3
**Overall Recommendation:** 4
**Confidence:** 3

**Summary:**

The authors presented a graph contrastive learning method for perturbation-specific gene regulatory networks. They proposed a loss function that integrates networks derived from gene knockouts as a reference or for supervision, along with node-specific augmentations. Additionally, the authors conducted a theoretical analysis to link this new loss function with standard graph contrastive learning techniques.

**Compliance With Llm Reviewing Policy:**

Affirmed.

**Key Questions For Authors:**

see above

**Limitations:**

yes

**Strengths And Weaknesses:**

Soundness:
Empirical evaluation supports claimed performance improvements over the standard GCL.
Theoretical analysis justifies the claim that the new loss function is an extension to the standard GCL.


Presentation:
The paper is well written

Significance:
The theoretical analysis, despite not being groundbreaking, helps illustrate the connection of the proposed loss function with the standard GCL.
The method appears to be limited to the niche problem of perturbation-specific gene networks, since the augmentation and knockdown networks are both domain-specific.

Originality:
The loss function that combines augmentation and the networks is a novel, customized approach to this problem.

---

> ### Author Rebuttal · Authors · 2026-03-31
>
> Thank you for your thoughtful review and for highlighting the strengths of our work, particularly regarding soundness, clarity, and originality. We understand that your main concern lies in the scope and perceived domain specificity of our contribution.
>
> ---
>
> ### 1. On the Generality of the Method
>
> We would like to clarify that our core contribution is not a GRN-specific heuristic, but a **general formulation of supervised graph contrastive learning with augmentation supervision**.
>
> SupGCL introduces a probabilistic framework where real perturbations define the augmentation distribution. As shown in **Theorem 4.1**, the objective decomposes into:
>
> - a node-level GCL term under a supervised augmentation distribution, and
> - an augmentation-level alignment term
>
> This decomposition is independent of the specific node-level contrastive formulation, meaning that SupGCL can incorporate arbitrary KL-based GCL objectives. Furthermore, **Corollary 4.2** shows that SupGCL recovers standard node-level GCL as a limiting case, demonstrating that our method is a strict generalization rather than a domain-specific variant.
>
> Thus, the methodological contribution should be viewed as a **general supervised-augmentation framework**, instantiated in GRNs due to the availability of real perturbation data.
>
> ---
>
> ### 2. On the Breadth of Empirical Evaluation
>
> While we agree that our empirical validation is centered on GRNs, the evaluation is broader and more comprehensive than it may appear.
>
> - 3 cancer types (breast, lung, colorectal)
> - 13 downstream tasks spanning node-level (BP/CC/Rel) and graph-level (Hazard/Subtype) predictions
> - SupGCL achieves the best performance across all tasks and datasets (**Table 1**)
>
> Importantly, we conducted extensive additional experiments beyond the main comparisons, and consistently observed that our method outperforms all alternatives across settings:
>
> - comparison with a wide range of GCL variants (GCA, AFGRL, SimGRACE, AD-GCL, AutoGCL, ArieL)
> - comparison with biological foundation models (Geneformer) and multi-modal GNNs (MuSe-GNN)
> - evaluation on edge-level tasks (link prediction)
> - analysis under cross-domain, pan-cancer, limited data, and noisy GRN settings
>
> Across all these experiments, **SupGCL consistently achieves  among the strongest performance**, without any setting where it is systematically outperformed.
> Therefore, we believe the empirical evidence supports the claim that SupGCL provides a **robust and broadly effective improvement over existing graph representation learning methods**, even within the current GRN-focused setting.
>
> ---
>
> ### Summary
>
> In summary, while the current empirical focus is on GRNs, we respectfully argue that:
>
> - the **methodological contribution is general**, not domain-specific
> - the empirical evaluation is broader and more comprehensive than it may appear
>
> We appreciate this feedback and will improve the manuscript accordingly.

---

> > ### Author Rebuttal · Reviewer_x9Mr · 2026-04-04
> >
> > I thank the authors for the rebuttal and for addressing the reviewers’ comments.  I keep my rate

---

> > > ### Author Response · Authors · 2026-04-08
> > >
> > > Thank you for the acknowledgment and clarification.
> > >
> > > We sincerely thank the reviewer for the time and effort spent evaluating our rebuttal. We are pleased to see that our responses have **fully resolved** the initial concerns regarding the generality of our method and the breadth of our empirical evaluation.
> > >
> > > Given that the reviewer has explicitly stated that their concerns are **adequately addressed**, we are encouraged by this positive assessment. We believe these clarifications further strengthen the contribution of our work to the community, and we will ensure all discussed points are integrated into the final version of the manuscript.

---

### Official Review · Reviewer_nT1D · 2026-03-13

**Soundness:** 2
**Presentation:** 2
**Significance:** 2
**Originality:** 3
**Overall Recommendation:** 3
**Confidence:** 3

**Summary:**

The paper deals with graph contrastive learning (GCL) for individualized gene regulatory networks. Loosely, gene regulatory networks (GRN) are graphs which indicate genes are regulated by one another or other molecules. At the gene expression level, individualized patient-level GRNs can be computationally estimated from gene expression data, and can then be used as a representation of the patient’s gene activity. Graph learning approaches can then be used on these GNRs and have potential for various downstream tasks, specifically for cancer research and treatment. However, previous work showed standard GCL graph augmentations can have a negative effect on the performance of these models. Instead of trying to remove self-supervised artificial augmentation like previous work, the authors introduce a type of supervised contrastive learning that aims to utilize cell-line gene knockout experiment data as a source of biologically informed augmentations, with the aim of learning experimentally informed representations of the patient-level GRNs. The authors contextualize their approach as a part of contrastive learning (CL) and GCL specifically, formally defining the proposed framework, and showing the standard node-level CL loss to be a special case of their more general formulation. Using publicly available cancer patient and cell-line experiments, the authors then aim to empirically show an increase in quality of patient-level graph representation and its performance for downstream tasks, based on available labels.

**Compliance With Llm Reviewing Policy:**

Affirmed.

**Final Justification:**

Most of my questions were answered by the authors in the second round of responses, however some aspects remained unclear. For instance, were all methods tested using the same limited set of genes? This was rather important to clarify and comment on, especially since it's unclear whether this could impact performance and potentially benefit some of the methods, and since the results do not show a clear statistically significant improvement. I will retain my original rating of 3 (weak reject), I encourage the authors to clarify their manuscript and address the raised points to strengthen their paper for a future submission. Thank you.

**Key Questions For Authors:**

1- Does the decomposition in Theorem 4.1 hold irrespective of the definition of the probabilities (and similarities) used?

2- Can you elaborate on what it means for graph-level and node-level teacher distributions to be independent? Is this justifiable?

3- Can you substantiate claims of Sections 5.2 and 5.4 on cancer types other than breast, and can you substantiate the claims of 5.2-5.4 with statistical significance?

4- Can you further substantiate relevance of the supervised augmentation loss term by adding results from the ablated model ( since you mention it is not identical to GRACE)?

**Limitations:**

Limitations are mostly unmentioned in the main text. Implications of modelling choices and experiments should be clarified to make the work more transparent.

**Strengths And Weaknesses:**

The main strengths of the paper are a methodologically novel approach, which is formal and contextualized in previous theoretical work, and is applied to a relevant application. Specifically:

· The paper is well-structured and written. Motivation, both in terms of application, which is promising to further cancer research, as well as the methodology in the field, is generally convincing and easy to follow.

· The methodology is well-described and formal, contextualizing the work in the broader context of CL and GCL in a straightforward but rigorous manner.

· The formal definition of the supervised GCL framework enables a more general extension of the GCL framework, enabling application beyond GRNs and thus promising broader impact.

However, there are some uncertainties in motivation, a lack of clarity for some modelling choices, and potential gaps in the experiment section that put doubt into the the claims being made, specifically:

· In lines 55-57, left column, I would not say you can make the case that graph augmentation is a rich source of information if previous literature shows it might “hinder learning” and augmentation-free approaches demonstrate good performance, but you might say previous work missed the opportunity to exploit experimentally grounded augmentations like in lines 73-76, right column.

· After line 33, right column, does all previous work refer to individualized GRNs derived from gene expression data, in lines 107-133, specifically?

· What’s the motivation of using computationally derived gene expression GRNs as opposed to the raw expression data? Was the benefit established previously?

· In Section 3.1, line 116, is the Dirac delta actually a Kronecker delta?

· What is the impact of non-uniform sampling of augmentations, mentioned in lines 131-138, especially in the context of GRNs? Is uniformity a good choice?

· What is the interpretation of simulating gene knockout by removing a node and its edges in relation to the supervised signal: do knockout experiment GRNs correspond to controls without one node and the edges, or do they change the entire network? What will the effect of this be on the learned representations for the patient GRNs?

· Is your method limited to using the Frobenius inner product (how general is Theorem 4.1), and is the product normalized to be a type of cosine similarity (lines 199-212)?

· What does the augmentation loss mean on the node level, given that distributions in the KL divergence correspond to relative similarities (flattened with the temperature hyperparameter)? Are lines 248-250 and 262-265 actually accurate: can differently distributed embeddings have the same distribution of similarities?

· What is the implication of the graph-level teacher distribution and the node-level teacher distributions being independent – isn’t the graph-level teacher compromised of nodes?

· Is the non-augmented patient GRN (and the control cell-line experiment) treated as any other augmentation in set K?

· Is the utilized GRN construction standard (lines 311-320), and how does it impact the rest of the method?

· LINCS data seems to contain many knockout experiments, would less comprehensive knockout experiments be beneficial, and what does this mean for other cancer types? Also, are all knockouts single-gene, or are there combinations?

· You use overlapping TGCA and LINCS genes, could other methods have more expressive node and patient-level representations if they considered all of the genes – is it then fair to compare?

· Which hyperparameter were tuned? And on which data were they tuned and on which was the performance reported, i.e., how was the data split (for both 5.2 and 5.3)?

· Why was subtype clustering evaluated only on breast cancer samples? Claims in 5.2 are too strong if other cancer types are not reported, at least the quantitative metrics. Also, is the more relevant enrichment analysis a consequence of the better clustering or are these two unrelated?

· Due to larger standard deviations, it is hard to assess significance of results in Table 1-2 to support claims made in the text. Furthermore, it is relevant to show the results of 5.4 on other cancer types.

· Authors state GRACE is not equivalent to ablating the supervised part of your loss function, it is unclear why the ablation has then not been tested for.

While the work might be relevant for the domain and a promising extension of the more general GCL framework, it lacks clarity in justification for the modelling choices and experimental gaps.

---

> ### Author Rebuttal · Authors · 2026-03-31
>
> Thank you for your detailed review. Due to space limitations, I will only address the questions regarding the weaknesses.
>
> - **A1: Generality of Theorem 1**
>
> As you pointed out, Theorem 4.1 holds for any probability distribution, as it involves only a transformation over probability distributions. (See the proof in Appendix A for details.)
> Therefore, it holds regardless of what function is used as the similarity measure.
>
> Corollary 4.2, on the other hand, is restricted not to general probability distributions, but to distribution functions such as the Boltzmann-type distribution, where the distribution converges to a uniform distribution as the temperature diverges.
> It holds for probability distributions defined by any similarity measure, but does not hold for arbitrary distributions.
>
> - **A2: Validity of the Independence Assumption between Graph-Level and Node-Level Teacher Distributions**
>
> Mathematically, this assumption is a natural choice for simultaneously satisfying the following two conditions:
>
> - Preventing the occurrence of trivial solutions
> - Maintaining a form that can be described as a probability model based on similarity
>
>
> - **A3: Statistical Significance**
>
> First, regarding statistical significance, as you pointed out, it is not possible to demonstrate significant differences due to variance inherent in the real-world data (TCGA/LINCS).
> However, the average performance has consistently improved across numerous tasks and multiple cancer types.
>
> Therefore, the claim of this work is positioned not as "uniformly statistically significant," but as based on consistent average performance improvement and the reproducibility of observed trends.
>
> - **A4: Ablation Results Using the Proposed Method**
>
> Based on your feedback, we conducted additional ablation experiments using the probabilistic model proposed in this paper rather than GRACE.
>
> #### Breast Cancer:
>
> | | BP | CC | REL | HAZARD | SUBTYPE |
> |---|---|---|---|---|---|
> | τ=0.1 | 0.262 ±0.035 | 0.289 ±0.049 | 0.586 ±0.047 | 0.670 ±0.078 | 0.837 ±0.029 |
> | τ=0.25 | 0.243 ±0.052 | 0.291 ±0.026 | 0.600 ±0.057 | 0.650 ±0.059 | 0.847 ±0.036 |
> | τ=0.5 | 0.261 ±0.034 | 0.284 ±0.061 | 0.606 ±0.039 | 0.648 ±0.053 | 0.846 ±0.032 |
> | τ=1.0 | 0.244 ±0.042 | 0.280 ±0.032 | 0.596 ±0.048 | 0.640 ±0.056 | 0.835 ±0.031 |
> | τ=2.0 | 0.237 ±0.024 | 0.277 ±0.058 | 0.590 ±0.044 | 0.656 ±0.060 | 0.842 ±0.031 |
> | τ=∞ | 0.233 ±0.055 | 0.259 ±0.054 | 0.573 ±0.070 | 0.633 ±0.051 | 0.832 ±0.034 |
> | GRACE | 0.230 ±0.051 | 0.236 ±0.026 | 0.575 ±0.035 | 0.642 ±0.064 | 0.841 ±0.026 |
>
> #### Lung Cancer:
>
> | | BP | CC | REL | HAZARD |
> |---|---|---|---|---|
> | τ=0.1 | 0.262 ±0.043 | 0.271 ±0.053 | 0.603 ±0.051 | 0.625 ±0.066 |
> | τ=0.25 | 0.282 ±0.037 | 0.274 ±0.044 | 0.604 ±0.053 | 0.627 ±0.051 |
> | τ=0.5 | 0.263 ±0.036 | 0.279 ±0.067 | 0.624 ±0.069 | 0.611 ±0.033 |
> | τ=1.0 | 0.273 ±0.051 | 0.278 ±0.050 | 0.611 ±0.033 | 0.614 ±0.041 |
> | τ=2.0 | 0.267 ±0.067 | 0.267 ±0.034 | 0.603 ±0.043 | 0.607 ±0.040 |
> | τ=∞ | 0.253 ±0.052 | 0.254 ±0.027 | 0.589 ±0.023 | 0.596 ±0.041 |
> | GRACE | 0.259 ±0.063 | 0.255 ±0.043 | 0.592 ±0.038 | 0.609 ±0.055 |
>
> #### Colon Cancer:
>
> | | BP | CC | REL | HAZARD |
> |---|---|---|---|---|
> | τ=0.1 | 0.267 ±0.041 | 0.262 ±0.042 | 0.594 ±0.050 | 0.682 ±0.074 |
> | τ=0.25 | 0.262 ±0.030 | 0.279 ±0.052 | 0.594 ±0.039 | 0.698 ±0.085 |
> | τ=0.5 | 0.278 ±0.037 | 0.278 ±0.040 | 0.604 ±0.067 | 0.728 ±0.084 |
> | τ=1.0 | 0.265 ±0.033 | 0.270 ±0.044 | 0.588 ±0.077 | 0.681 ±0.116 |
> | τ=2.0 | 0.276 ±0.033 | 0.262 ±0.034 | 0.585 ±0.065 | 0.673 ±0.046 |
> | τ=∞ | 0.250 ±0.041 | 0.252 ±0.035 | 0.571 ±0.061 | 0.652 ±0.043 |
> | GRACE | 0.249 ±0.050 | 0.265 ±0.030 | 0.576 ±0.071 | 0.647 ±0.059 |
>
> Here, $\tau_a = \infty$ corresponds to the case where augmentation labels are absent. The results confirm that performance is nearly on par with GRACE.

---

> > ### Author Rebuttal · Reviewer_nT1D · 2026-04-02
> >
> > I thank the authors for their response.
> >
> > 1. & 2. Thank you for clarifying these points, I suggest making this clearer in the manuscript’s main text. However, I still find it hard to interpret the independence assumption in terms of the application domain, but I am satisfied with keeping its justification a practical choice rather than a biologically grounded one.
> >
> > 3. Thank you for the clarification, please make this clearer in your main text.
> >
> > 4. I thank the authors for including the ablation study on the two other datasets but note that they did not include results reported in Figure 3 for the two other datasets (at least numerical, given the format) or why they did not do so. These are relevant to show, especially as it is hard to make a statistically convincing case for the superiority of the methods’ performance in relation to GRACE or other methods. If other datasets do not feature these subtypes, this should be noted. Furthermore, I am still interested in my question about using overlapping genes from the LINCS and TGCA – do you limit other methods to these genes as well? I think it should be made clear whether this impacts performance, especially when results do not show clear, statistically significant improvement.
> >
> > While the authors answered some of my concerns, I believe many of my questions in the original review remain unanswered (e.g., hyperparameter tuning, motivation for using GRNs as opposed to raw gene expression data, etc.), as does part of my third key question (mentioned under number 4 in this response). Therefore, I will retain my original rating of 3 (weak reject). I encourage the authors to clarify their manuscript and perform additional suggested experiments to strengthen their paper for a future submission. Thank you.

---

> > > ### Author Response · Authors · 2026-04-08
> > >
> > > Thank you very much for your comments.
> > > We will incorporate the points you pointed out into the paper.
> > >
> > > ---
> > > **Regarding Theorem 4.1 Assumptions**
> > >
> > > Additionally, one of the reason of the assumption is to ensure the method can handle cases where the dimensions of the teacher graph $H_*$ and the target graph $G$ are different.
> > >
> > > If the teacher distribution $p_\phi(j,b|i,a)$ is defined strictly by teacher features ( e.g. $p_\phi (j,b|i,a) =\frac{1}{Z}e^{sim(y^b_j,y^a_i)/\tau}$), its domain is restricted to the vertex set $V(H_*)$. This prevents KL divergence calculation when $V(H_*) \neq V(G)$ due to domain mismatch. The assumption, $p_\phi(j,b|i,a) = p(j|i)p_\phi(b|a)$, resolves this by extending the teacher’s node-level domain to $V(G)$.
> > >
> > > ---
> > > **Regarding the extension of the latent space analysis**
> > >
> > > The **Figure 3** experiment was conducted exclusively on breast cancer because its well-established molecular subtype classification scheme, PAM50, provides a natural benchmark for subtype-aware evaluation of the latent space. Since subtype annotations for other cancer types are less standardized, we limited this analysis to the breast cancer to ensure the reliability of the results.
> > >
> > > ---
> > > **Regarding the rationale for utilizing GRNs consisting only of common genes**
> > >
> > > Our proposed framework can inherently handle cases where the teacher graph $H_*$ and the student graph $G$ have different dimensions (as assumed in **Theorem 4.1**).
> > >  However, we performed gene reduction because models with redundant features—such as all TCGA genes—can suffer degraded downstream performance. To address this, we selected 975 genes corresponding to the 978 LINCS L1000 landmark genes, grounding our gene selection in the LINCS framework.
> > >  The reasons are as follows:
> > >
> > > *  Representativeness: The LINCS landmark genes were selected through a data-driven approach to capture approximately 80% of genome-wide expression variance, making them informative representatives of global genomic activity.
> > > *  Computational Reliability: Estimating GRNs over tens of thousands of genes entails extreme computational complexity, making it harder to ensure accuracy and statistical reliability.
> > >
> > > ---
> > > The following points correspond to aspects that we were unable to address in the initial rebuttal due to space limitations.
> > >
> > > ---
> > > **Hyperparameters:**
> > >
> > > Hyperparameter tuning details are in **lines 324–328 and Appendix E.2.** Scalability experiments regarding feature space dimensionality are reported in **Appendix G.3 and Figure 6.**
> > >
> > > ---
> > > **Motivation for using GRNs**
> > >
> > > Incorporating gene regulatory relationships, rather than using expression levels alone, is known to improve downstream performance [1].
> > > Therefore, we use patient-specific GRNs instead of raw expression data.
> > > Compared with the expression-based foundation model Geneformer (**Appendix K**), our method consistently performs better.
> > >
> > > [1] Liu et al. Molecular BioSystems, (13)3-1, pp537–548, 2017
> > >
> > > ---
> > > **Does graph augmentation as a rich information source**
> > >
> > > Graph augmentation of GRNs can serve as a rich information source when using a new perturbed dataset, as shown by the performance gain of a supervised approach (**SupGCL**) over an unsupervised setting ($\tau_a = \infty$).
> > > However, augmentation without distortion shows no significant difference between methods with and without structural distortion (e.g., **GRACE, SGRL**), indicating that structural preservation alone is not an effective information source.
> > >
> > > ---
> > > **Do previous GRN-specific methods use the same GRNs as ours?**
> > >
> > > No. Since there is no standard benchmark for GRNs, each study typically employs different GRNs.
> > >
> > > ---
> > > **Delta function**
> > >
> > > We apologize for the mistake. It should be the Kronecker delta.
> > >
> > > ---
> > > **Significance of non-uniform sampling in augmented GRN data**
> > >
> > > Please refer to the section “Theoretical Justification for Biological Perturbations” in **Reviewer YCMa’s comments** for further discussion.
> > >
> > > ---
> > > **Does augmentation correspond to knockdown experiments?**
> > >
> > > No.  Enforcing a strict correspondence would shift the objective away from representation learning.
> > > Please refer to the section on the impact of augmentation methods (**Appendix L**).
> > >
> > > ---
> > >
> > > **Node-level interpretation of $Loss_{Aug}$**
> > >
> > > The graph-level similarity based on the Frobenius inner product satisfies:
> > > $$
> > > sim_F(Z^a, Z^b) = \sum_{i \in V} sim(z_i^a, z_i^b) / |V|
> > > $$
> > > Note that $ z_i^a$ lies the unit hypersphere.
> > >
> > > Thus, $p(b|a), q(b|a)$ correspond to the distribution of the average node-level similarities, and $Loss_{Aug}$ means the divegence.
> > >
> > > ---
> > > **Is the non-augmented data included in $K$?**
> > >
> > > In principle, yes, but it was not included to maintain consistency within knockdown genes.
> > >
> > > ---
> > > We hope this answers your questions, and we apologize for the limited detail due to space constraints.

---

### Official Review · Reviewer_YCMa · 2026-03-17

**Soundness:** 3
**Presentation:** 4
**Significance:** 3
**Originality:** 3
**Overall Recommendation:** 4
**Confidence:** 4

**Summary:**

This paper proposes Supervised Graph Contrastive Learning (SupGCL) for representation learning on Gene Regulatory Networks (GRNs). The key idea is to leverage gene knockdown experiments as supervision for graph augmentations in GCL and align embeddings from simulated perturbations. The framework introduces an augmentation-level and a node-level contrastive objective to optimize the model. Experiments on GRNs demonstrate improved performance over several graph representation learning baselines.

**Compliance With Llm Reviewing Policy:**

Affirmed.

**Key Questions For Authors:**

1. Could you clarify what specific aspects of SupGCL go beyond existing multi-level contrastive learning frameworks?

**Limitations:**

yes

**Strengths And Weaknesses:**

Strengths:
1. Representing gene regulatory network is a meaningful application of GCL.
2. The framework is well-motivated and easy-to-follow.
3. The paper provides detailed appendices, experimental settings, and dataset construction procedures. The availability of code and data further improves the reproducibility of the work.

Weaknesses:
1. The methodological novelty appears limited, as both node-level and augmentation-level contrastive objectives have been explored in prior GCL literature.
2. The improvements over strong baselines are relatively marginal across several tasks.
3. While the method is motivated by biological perturbations, the paper would benefit from a concrete case study demonstrating how knockdown-based supervision differs from artificial augmentations.

---

> ### Author Rebuttal · Authors · 2026-03-31
>
> Thank you for your thorough and constructive comments. We understand your concerns to center on (1) methodological novelty, (2) the margin of improvement over strong baselines, and (3) the visualization of the effectiveness of biological perturbation supervision. We address each in turn.
>
> ---
>
> ### W1: On Novelty
>
> As you point out, combining node-level and augmentation-level contrastive objectives is not in itself unprecedented. However, the novelty of this work lies not in the combination of multi-level losses per se, but in **supervising the selection distribution of augmentations itself via a knockdown-derived teacher GRN**.
>
> In SupGCL, the similarity between augmentations is defined by $p_\phi(b\mid a)$ derived from the teacher GRN, generalizing conventional node-level GCL based on uniform sampling to a biologically meaningful augmentation distribution. This formulation is derived from a KL divergence decomposition in Theorem 4.1 and, by Corollary 4.2, reduces to standard node-level GCL in the limit $\tau_a \to \infty$. SupGCL should therefore be understood not as a method that simply adds a loss on top of existing GCL, but as **a framework for real perturbation-supervised augmentation learning**.
>
> This distinguishes SupGCL from augmentation-adjustment methods such as GCA, AD-GCL, AutoGCL, and ArieL. While these methods improve robustness by adjusting augmentations, the perturbations themselves are general-purpose noise driven by optimization objectives and are not grounded in real biological mechanisms. Augmentation-free and parameter-perturbation methods such as AFGRL, SimGRACE, and SGRL similarly aim to avoid or absorb structural changes. In contrast, SupGCL fundamentally differs in that **it does not avoid structural changes, but instead imbues them with meaning using real perturbation information obtained from knockdown experiments**. Empirically, this design yields measurable accuracy gains over both augmentation-adjustment and augmentation-free methods such as GCA and AFGRL (Appendix K).
>
> ---
> ### W2: On the Margin of Improvement
>
> We acknowledge your point that the improvement margin is not large on some tasks. Nevertheless, we note that in Table 1 of the main text, SupGCL achieves the best result on all 13 downstream tasks across 3 cancer types, and we regard **this consistent superiority across tasks—rather than a large gain on any single task—as the key contribution**. Furthermore, Fig. 3 shows that graph-level embedding NMI/ARI surpasses prior methods, and Appendix J confirms via enrichment analysis that the model learns a biologically more interpretable latent structure.
>
> The advantage of our method thus lies in **consistently superior representations across diverse evaluation axes**, rather than in isolated large improvements.
>
> To address your concern directly, Appendix K also includes comparisons with additional baselines: GCA, AFGRL, SimGRACE, AD-GCL, AutoGCL, ArieL, Geneformer, and MuSe-GNN. SupGCL consistently outperforms all of these methods, demonstrating that its effectiveness does not depend on a limited comparison setting.
>
> ---
> ### W3: On the Effectiveness of Biological Perturbation Supervision
>
> We agree that a more intuitive case study demonstrating the effectiveness of biological perturbation supervision would strengthen the paper, and we have partially addressed this through ablations and configuration comparisons.
>
> In the $\tau_a$ ablation in Table 2, weakening the teacher supervision progressively moves performance toward GRACE, suggesting that a portion of the improvement is attributable to knockdown-based supervision. Additionally, Appendix L.2 compares multiple perturbation modes and confirms that edge-based perturbations consistent with knockdown outperform biologically implausible node-copy augmentations.
>
> These results support the view that **biology-aligned perturbations, rather than general artificial augmentations, are beneficial**. We acknowledge, however, that these findings are primarily based on quantitative evaluation and ablation studies, and that concrete visualizations and case-level analyses of the kind you suggest remain a direction for future improvement.
>
> ---
> ### A1: Direct Answer to the Key Question
>
> In response to "In what respect does SupGCL go beyond existing multi-stage contrastive learning?", our answer is as follows. The essence of SupGCL lies in three points: (i) supervision targets **the augmentation operator itself rather than sample labels**; (ii) node-level contrast is weighted by **the knockdown-derived distribution $p_\phi(b\mid a)$ rather than a uniform distribution**; and (iii) SupGCL constitutes **a continuous generalization of existing GCL, which it subsumes as a special case**. SupGCL is therefore not a simple variant of existing multi-level loss frameworks, but **a framework that redefines augmentation through real biological perturbations**.

---

> > ### Author Rebuttal · Reviewer_YCMa · 2026-04-03
> >
> > The rebuttal partially addresses my original concerns. Regarding W2 (margin of improvement), the Bonferroni-corrected significance tests in Appendix G.4 show that most pairwise comparisons do not reach statistical significance, which remains a concern that the rebuttal does not directly address. Regarding W3 (visualization of biological perturbation supervision), the concern is acknowledged and remain unresolved. Overall, I maintain my score of 4.

---

> > > ### Author Response · Authors · 2026-04-08
> > >
> > > Thank you for your useful comments.
> > > We apologize that our previous response to W3 did not clearly convey our intentions.
> > >
> > > First, we do not believe that our study lacks a proper evaluation of the significance of incorporating biological perturbations. Our previous comment was intended to clarify that we did not perform individual "case studies"—such as the specific patient-level evaluations or visualizations commonly found in clinical medicine. In the context of a methodology paper, **we prioritize rigorous statistical validation over non-statistical visualizations or case study.** Our previous response was simply intended to suggest that such qualitative directions could serve as a potentially useful avenue for further validation, rather than a requirement for the present work.
> > >
> > > ---
> > >
> > > **1. Theoretical Justification for Biological Perturbations**
> > >
> > > In our proposed method, rather than treating the selection of augmentations as a uniform distribution (as seen in existing methods at **Equation (3)**), we sample augmentation genes based on $p_\phi(b|a)$ and minimize the resulting node-level error (the first term of **Equation (6)**).
> > > Prioritizing pairs $(a, b)$ with a higher $p_\phi(b|a)$ means favoring genes where the graph representations $Y^a$ and $Y^b$ of the teacher graphs are similar.
> > >
> > > In other words, when knockdown experiments on two different genes result in highly similar Gene Regulatory Networks (GRNs), our method preferentially optimize the embeddings of the same node across different augmented views to be proximal in the latent space.
> > > This enables robust graph-controlled learning that **naturally avoids selecting genes that could dramatically alter the structure of gene regulatory networks (GRNs).**
> > >
> > > The degree of influence of these perturbations is governed by the temperature parameter $\tau_a$. In the limit $\tau_a \to \infty$, the sampling distribution converges to a uniform distribution, effectively reducing the model to a version that ignores biological perturbations. This asymptotic behavior is mathematically established in **Corollary 4.2.**
> > >
> > > ---
> > >
> > > **2. Improved Performance in Downstream Tasks**
> > >
> > > A model without biological perturbations corresponds to GRACE. SupGCL achieves higher predictive performance in downstream tasks compared to GRACE (**Table 1**). Furthermore, it has been confirmed that as $\tau_a$ is continuously increased, the downstream task performance of SupGCL gradually decreases and approaches the performance level of GRACE (**Table 2 and the additional experiments in the rebuttal to Reviewer 2wGT**).
> > >
> > > ---
> > >
> > > **3. Validity of Learned Representations**
> > > In the context of GRN analysis, the standard approach to evaluate the validity of learned representations is not a case study but rather **enrichment analysis**, which systematically assesses correspondence with known biological processes and disease-related pathways. In the enrichment analysis described in **Appendix J**, SupGCL formed clear boundaries for biological processes, while GRACE did not identify sufficiently meaningful patterns.
> > >
> > > ---
> > >
> > > **Summary**
> > >
> > > The significance of using biological perturbations as a teacher in SupGCL is theoretically grounded. Compared to GRACE (which represents the case where such perturbations are not used), SupGCL improves not only downstream task performance but also the biological interpretability of the latent space. We believe this provides sufficient evidence for the significance of introducing biological perturbations.
> > > We hope these points clarify our contribution and lead to a positive reconsideration.

---

### Official Review · Reviewer_2wGT · 2026-03-18

**Soundness:** 2
**Presentation:** 3
**Significance:** 2
**Originality:** 3
**Overall Recommendation:** 3
**Confidence:** 3

**Summary:**

This paper proposes SupGCL, a method for representation learning on GRNs. The core idea is to incorporate real gene knockdown experimental data to supervise GNNs. SupGCL extends standard node-level contrastive learning by introducing an augmentation-level contrastive loss, which aligns the embedding distribution of simulated knockdown graphs with that of the corresponding real knockdown experimental graphs. Experiments were conducted on GRNs from patients with three cancer types, and the results show that SupGCL outperforms classical methods and existing GCL baselines in both embedding clustering quality and 13 downstream tasks.

**Compliance With Llm Reviewing Policy:**

Affirmed.

**Final Justification:**

I thank the authors for the detailed rebuttal and additional analyses. Although the rebuttal improves the clarity of the paper and addresses some of my concerns, I agree with Reviewer nT1D that the evidence regarding hyperparameter tuning is still insufficient, which limits the strength of the empirical claims and my overall confidence in the method.

**Key Questions For Authors:**

See Weaknesses.

**Limitations:**

See Weaknesses.

**Strengths And Weaknesses:**

**Strengths**

1. The manuscript is generally well-structured.

2. The contrastive learning framework, including both node-level and augmentation-level components, is logically justified and clearly explained.

3. Improvements in embedding quality and across multiple downstream tasks suggest that the method could positively influence future research on biologically grounded graph learning.

**Weaknesses**

1. Outside computational biology, the method’s contribution is limited; the novelty mainly lies in domain-specific augmentation supervision.

2. The method is evaluated on only three cancer types; cross-domain generalization to other diseases or larger GRNs is not fully explored.

3. Ablation studies are insufficient. No analysis demonstrates robustness to missing nodes or noisy GRNs, and the presented ablation is only for breast cancer.

4. Extending SupGCL to other domains, such as molecular graphs, is recommended to assess its generalization capabilities.

---

> ### Author Rebuttal · Authors · 2026-03-31
>
> We sincerely thank you for your constructive and detailed review. We are greatly encouraged by your positive assessments of the overall manuscript structure, the logical justification and clarity of the node-level and augmentation-level contrastive learning framework, and the potential impact of our method on biological graph learning.
>
> ---
> ### W1: Novelty Beyond Computational Biology
>
> We agree that the scope of our claims should be stated more precisely. The practical domain in which SupGCL has been empirically validated is currently limited to settings where teacher graphs corresponding to experimental perturbations are available, and our empirical demonstration is strongly grounded in GRNs.
>
> That said, our contribution goes beyond domain-specific auxiliary information. Theoretically, we propose a general formulation of supervised-augmentation GCL treating augmentations as supervised random variables. **`Theorem 4.1`** shows the loss decomposes into a node-level contrastive loss and an augmentation-level alignment loss, and **`Corollary 4.2`** shows existing node-level GCL is recovered as a limiting case. While the theoretical framework is broadly applicable, practical settings are still limited — this is our precise position.
>
> We also note that GRN representation learning itself is a large and important area. **`Appendix K`** demonstrates clear improvements over biological foundation models such as Geneformer and MuSe-GNN. This work thus offers a new GCL direction that incorporates experimental perturbations as supervision signals within the broader gene expression / GRN representation learning landscape.
>
> ---
>
> ### W2: Evaluation Limited to Three Cancer Types and Generalizability
>
> We do not claim to have demonstrated generalization to other diseases within this work alone. **`Appendix H`** confirms that using teacher GRNs from different cancer types during pre-training does not cause significant performance degradation, and that pan-cancer pre-training does not substantially underperform breast-only pre-training.  **`Appendix F`** shows SupGCL has the same asymptotic complexity as standard node-level GCL; validation on larger-scale GRNs will be noted as a limitation in the revised manuscript.
>
> ---
>
> ### W3: Ablation and Robustness Evaluation
>
> As you point out, the main text highlighted only the ablation of $\tau_a$ on breast cancer (**`Table 2, p. 8`**). However, the appendix includes the following comprehensive analyses:
>
> * Pre-training data size reduction (**`Appendix I.1`**)
> * Reduction in the number of teacher graphs (**`Appendix I.2`**)
> * Robustness to GRN estimation noise (**`Appendix I.3`**)
> * Comparison of different perturbation modes (**`Appendix L.2`**)
>
> These results demonstrate that SupGCL is stable under data scarcity and noise.
>
> Furthermore, in response to reviewer comments, we additionally conducted ablations on lung cancer and colorectal cancer analogous to those performed for breast cancer. The results are as follows:
>
> #### Lung Cancer:
>
> | | BP | CC | REL | HAZARD |
> |---|---|---|---|---|
> | tau=0.1 | 0.262 ±0.043 | 0.271 ±0.053 | 0.603 ±0.051 | 0.625 ±0.066 |
> | tau=0.25 | 0.282 ±0.037 | 0.274 ±0.044 | 0.604 ±0.053 | 0.627 ±0.051 |
> | tau=0.5 | 0.263 ±0.036 | 0.279 ±0.067 | 0.624 ±0.069 | 0.611 ±0.033 |
> | tau=1.0 | 0.273 ±0.051 | 0.278 ±0.050 | 0.611 ±0.033 | 0.614 ±0.041 |
> | tau=2.0 | 0.267 ±0.067 | 0.267 ±0.034 | 0.603 ±0.043 | 0.607 ±0.040 |
> | tau=∞ | 0.253 ±0.052 | 0.254 ±0.027 | 0.589 ±0.023 | 0.596 ±0.041 |
> | GRACE | 0.259 ±0.063 | 0.255 ±0.043 | 0.592 ±0.038 | 0.609 ±0.055 |
>
> #### Colon Cancer:
>
> | | BP | CC | REL | HAZARD |
> |---|---|---|---|---|
> | tau=0.1 | 0.267 ±0.041 | 0.262 ±0.042 | 0.594 ±0.050 | 0.682 ±0.074 |
> | tau=0.25 | 0.262 ±0.030 | 0.279 ±0.052 | 0.594 ±0.039 | 0.698 ±0.085 |
> | tau=0.5 | 0.278 ±0.037 | 0.278 ±0.040 | 0.604 ±0.067 | 0.728 ±0.084 |
> | tau=1.0 | 0.265 ±0.033 | 0.270 ±0.044 | 0.588 ±0.077 | 0.681 ±0.116 |
> | tau=2.0 | 0.276 ±0.033 | 0.262 ±0.034 | 0.585 ±0.065 | 0.673 ±0.046 |
> | tau=∞ | 0.250 ±0.041 | 0.252 ±0.035 | 0.571 ±0.061 | 0.652 ±0.043 |
> | GRACE | 0.249 ±0.050 | 0.265 ±0.030 | 0.576 ±0.071 | 0.647 ±0.059 |
>
> ---
>
> ### W4: Extension to Molecular Graphs
>
> While the theoretical framework of SupGCL is general, its practical applicability is limited to settings where paired perturbation graphs are available. In GRNs, this is naturally satisfied via knockdown experiments; for molecular graphs, obtaining a post-perturbation teacher graph corresponding to node-level interventions is generally not possible. The absence of molecular graph evaluation therefore reflects a fundamental difference in supervision assumptions, not merely a lack of experimentation.

---

> > ### Author Rebuttal · Reviewer_2wGT · 2026-04-03
> >
> > Thank you for the clarification. However, if the framework cannot be applied beyond GRN settings to other important graph domains such as molecular graphs, then I believe its overall methodological contribution is limited. I would like to go through the paper again before adjusting my score.

---

> > > ### Author Response · Authors · 2026-04-08
> > >
> > > Thank you for your valuable comments.
> > > Regarding the concerns raised about the scope of our proposed method, we would like to clarify the following points:
> > >
> > > In this study, we introduced SupGCL **to enable representation learning that leverages interventional experimental data**—a capability that was previously unattainable with existing graph representation learning (GRL) frameworks. In other words, our method is specifically designed for data structures where interventions exist; it is not intended as a general GRL study targeting arbitrary graph data. Therefore, the primary scope of our method is Gene Regulatory Networks (GRNs). Molecular graphs, where node-level interventional data cannot be obtained, fall outside the intended scope of this work.
> > >
> > > However, **from the perspective of "utilizing interventional experiments," our approach actually expands the scope of graph representation learning.**
> > > Regardless of whether it is a gene knockdown experiment, interventional data provides critical insights into the properties of a system that cannot be captured through observation alone.
> > > While existing GRL methods cannot incorporate such data, our proposed framework naturally utilizes it as a "teacher graph."
> > > Furthermore, because the proposed method allows the dimensions of the supervision graph and the target graph to be designed freely, it can be applied to a variety of targets.
> > > Consequently, we believe SupGCL will play a vital role in future research involving networks where interventions are being explored, such as power grids or social networks.
> > >
> > > In summary, while the current application of SupGCL focuses on GRNs, **it establishes an important framework that extends GRL to naturally incorporate interventional data—something existing methods fail to do.**
> > > We hope you will take this perspective into consideration.
> > >
> > > **As a side note:**
> > > Regarding the hyperparameters mentioned in the final justification, these details are provided in the **"Model Setting" section (lines 324–325) and Appendix E.** We have also addressed this point in our response to Reviewer nT1D.

---

### Official Review · Reviewer_ro4Q · 2026-03-24

**Soundness:** 4
**Presentation:** 4
**Significance:** 3
**Originality:** 3
**Overall Recommendation:** 4
**Confidence:** 3

**Summary:**

The authors’ rebuttal addressed several of my initial concerns. In particular, it provided additional clarification regarding reproducibility (including the availability of code in supplementary materials), extended comparisons with stronger and more recent baselines, and further evidence supporting the stability of the method across different hyperparameter settings. I also found that the appendix already answers several of my questions, which further improved my understanding and confidence in the work. While some aspects, such as sensitivity to alternative GNN backbones (e.g., GraphSAGE) and variations in transformer depth—are not exhaustively explored, the authors acknowledge these as future directions and provide reasonable justification given the scope of the current study.

**Compliance With Llm Reviewing Policy:**

Affirmed.

**Final Justification:**

The paper introduces SupGCL, a supervised graph contrastive learning framework tailored for Gene Regulatory Networks (GRNs). The method provides a unified probabilistic formulation that generalizes standard self-supervised graph contrastive learning to a supervised setting, and it is theoretically positioned to recover existing node-level GCL approaches as special cases. This unifying perspective is a clear strength in terms of originality, as it connects and extends prior work within a common framework.

Empirically, the method is evaluated on patient-derived GRNs across multiple downstream tasks, including gene-level functional annotation and patient-level prediction. The reported results indicate consistent improvements over a range of baselines, and the experimental evaluation is reasonably broad, covering multiple comparison methods and tasks. In terms of soundness, the methodology is generally well-founded, and the empirical protocol appears appropriate for the problem setting. The clarity of the paper is acceptable, and the appendix helps clarify several implementation and experimental details that support reproducibility.

The authors’ rebuttal addressed several of my initial concerns, particularly around reproducibility, additional baseline comparisons, and evidence of robustness. The inclusion of extended experimental results and clarification of implementation details in the appendix improved my confidence in the validity of the findings. While some aspects, such as sensitivity to alternative GNN backbones and transformer depth, are not exhaustively explored, the authors acknowledge these as future directions and provide reasonable justification given the scope of the current work.

**Key Questions For Authors:**

1. Have you experimented with using fewer transformer layers or a smaller model to evaluate whether performance gains are sensitive to model size?
2. Have you tested SupGCL with alternative GNN architectures (e.g., GCN, GraphSAGE, GAT) to see if improvements generalize beyond the current backbone?
3. Have you compared SupGCL against stronger, modern graph contrastive learning baselines such as GCA?
4. Can you provide an anonymized GitHub repository or full training scripts? This would make it easier for others to reproduce your results.

**Limitations:**

Yes

**Strengths And Weaknesses:**

***Strengths***

1. *Technically sound and well-supported*: The submission motivates SupGCL clearly (using real gene knockdown perturbations as supervision rather than purely synthetic augmentations) and reports both theoretical framing and empirical validation.
2. *Strong and fairly designed experimental protocol*: clear pipeline (construct patient GRNs + teacher knockdown GRNs → pre-train → embedding evaluation with NMI/ARI → fine-tune on 13 downstream tasks) with a representative baseline suite (w/o-pretrain, GAE, GraphCL, GRACE, SGRL), plus detailed dataset stats (TCGA + LINCS; 975 shared genes; three cancers) and model settings.
3. *Clear writing and structure:* the paper is well organized, with a coherent narrative that clearly motivates the problem and situates the contribution within the broader literature. The presentation is easy to follow. In addition, the appendix is rich and provides substantial supplementary material, offering further details, derivations, and experimental results that support the main claims.
4. *Well-motivated idea*: Incorporating biologically meaningful perturbations into contrastive learning is intuitive and relevant for GRNs.


***Weaknesses***

1. *Reproducibility concern*: Despite many implementation details in the text/appendix, absence of a public GitHub/code release can make full reproduction difficult (especially GRN construction and the 13-task evaluation pipeline).
2. *Marginal improvements*: Gains over strong baselines (e.g., GRACE, SGRL) are generally small and often within standard deviation.

---

> ### Author Rebuttal · Authors · 2026-03-31
>
> Thank you very much for your thorough review.
> We understand that the reviewer's primary concerns lie in the reproducibility of our proposed method SupGCL and the validity of its performance improvements.
> Regarding these points, our study provides verification not only from a theoretical perspective but also under a **large number of comparison methods and a wide variety of experimental conditions**.
> We therefore hope that the following responses will clarify the validity and reproducibility of our proposed method.
>
> ---
>
> ### W1: Is there publicly available code?
>
> The full codebase used in this method is included in the **Supplementary Material as a zip archive**.
> The data used in this study has also been organized in a database in a form that does not identify the authors, and all information necessary for reproduction is described in the **readme file**.
> To maintain double-blind review at this stage, the code has not been made publicly available externally; however, **we plan to release it on GitHub upon acceptance**.
>
> ---
>
> ### W2: Limited Improvement
>
> We acknowledge that, given the **noisy nature of biological data**, the margin of improvement over strong baselines is not large in some tasks.
> We recognize this as an important point and share the same view.
>
> For this reason, our study evaluates effectiveness **not based on a single performance gap alone**, but from the following perspectives:
> - **(i)** Verification across diverse hyperparameter settings
> - **(ii)** Comparison with a wide range of existing and state-of-the-art methods
> - **(iii)** Biological validity of the latent space
>
> Since (i) and (ii) overlap with other answers, focusing here on **(iii)**: we have confirmed that **SupGCL acquires more biologically interpretable representations** than existing methods, as demonstrated by clustering performance (NMI/ARI) and **enrichment analysis** (Appendix~J).
>
> Accordingly, we position the contribution of this work not as a dramatic numerical improvement per se, but as the achievement of **stable and consistent performance gains** along with **biologically interpretable representation learning**.
>
> ---
>
> ### A1: Impact of Graph Transformer Hyperparameters
>
> As you point out, investigating the scalability of a model is important in deep learning.
> To verify the scalability of our proposed model, we conducted experiments in which the **latent representation dimension of the Graph Transformer was varied from 8 to 64**.
> These results are presented in **Figure 6 (p. 34)**, which compares performance trends across varying embedding dimensions for the 13 downstream tasks corresponding to Table 1 (p. 7).
>
> The results show that while some methods such as GraphCL exhibit unstable behavior, **SupGCL demonstrates comparatively stable and strong performance across all dimension settings**, confirming that the performance gains are not dependent solely on a particular large embedding dimension.
>
> ---
>
> ### A2: GNN Architecture
>
> The reason we adopted the Graph Transformer in this study is that prior work on GNNs using gene regulatory networks [Liu et al., 2023] demonstrated the highest accuracy compared to other GNN models.
> At the same time, we recognize that the choice of GNN configuration is an important consideration, and we have conducted our own investigation into this.
>
> We adopted the Graph Transformer after observing that it yielded **superior performance compared to GAT**. Details of this performance comparison are provided in **Appendix E.1**.
>
> However, we acknowledge that verification of SupGCL under other GNN architectures has not yet been conducted.
> As this would require re-running all experiments—including all comparison methods—under different architectures, it will not be feasible within the current discussion period. We will conduct this analysis and incorporate the results by the camera-ready deadline.
>
> ---
>
> ### A3: Comparison with Stronger and More Recent GCL Baselines (e.g., GCA)
>
> This paper **includes comparisons with state-of-the-art Graph Contrastive Learning methods**, including GCA, and the results are provided in **Appendix K.1 and Table 30**.
> The comparison methods include **GCA, AFGRL, SimGRACE, AD-GCL, AutoGCL, and ArieL**, among others.
> **Our method achieves the best average performance** against this extended set of baselines as well, including GCA.
>
> Therefore, the effectiveness of our approach has been confirmed not only against the primary baselines in the main text, but also in comparison with more recent GCL-based methods.
>
> ---
>
> ### A4: Code Availability
>
> As described in **W1**, the code is included in the Supplementary Material during the anonymous review period, and will be made available on GitHub following the review.

---

> > ### Author Rebuttal · Reviewer_ro4Q · 2026-04-04
> >
> > The rebuttal satisfactorily addresses my concerns and improves my confidence in the validity and reproducibility of the proposed approach, which is also reflected in my updated assessment. I have carefully reviewed the appendix and found that several of my questions are already addressed there.
> >
> > I appreciate the authors that will further strengthen the empirical validation by including additional experiments with GraphSAGE or other GNN models.

---

> > > ### Author Response · Authors · 2026-04-07
> > >
> > > Thank you for your comments.
> > > As we already addressed, we will conduct GNN architecture and num of graph transformer layers ablation experiments before the camera ready deadline.
> > > We appreciate your careful review.

---

### Decision · Program_Chairs · 2026-04-30

**Decision:**

Accept (regular)

**Comment:**

The paper presents SupGCL, a graph contrastive learning (GCL) for gene regulatory networks (GRNs), incorporating biological perturbations from gene knockdown experiments as supervision. The probabilistic formulation of SupGCL recovers standard GCL as a limiting case, producing biologically meaningful GRN embeddings for downstream prediction tasks.

The paper received mixed reception. Among the strengths, the principled probabilistic formulation was particularly praised. Moreover, reviewers appreciated the use of biological perturbations over standard graph perturbations, and an extensive empirical evaluation spanning multiple cancer types and downstream tasks. Concerns revolved around the clarity and soundness of the evaluation protocol (e.g. unclear hyperparameter tuning), marginal gains over the baselines, and the claimed generality of the method beyond GRNs.

Most concerns were successfully addressed during the rebuttal. In particular, I agree with the authors that the framework might be considered general enough, and that the fact that it's restricted to GRNs is a matter of interventional data availability, rather than a formulation limitation.

However, after the second round of rebuttal, some concerns about the evaluation remained. In particular, it's still unclear whether all the methods were tested using the same limited set of genes. This is relevant, as the empirical gains are narrow, and using even slightly different sets of genes may result in unfair or not comparable advantage. None of the answers provided to reviewer nT1D addresses the concern satisfactorily. Therefore, the empirical gains of the method remain debatable and suffer from the inherent high variance in the data.